# CE-SSL: Computation-Efficient Semi-Supervised Learning for ECG-based Cardiovascular Diseases Detection

## Abstract

The label scarcity problem is the main challenge that hinders the wide application of deep learning systems in automatic cardiovascular diseases (CVDs) detection using electrocardiography (ECG). Tuning pre-trained models alleviates this problem by transferring knowledge learned from large datasets to downstream small datasets. However, bottlenecks in computational efficiency and detection performance limit its clinical applications. It is difficult to improve the detection performance without significantly sacrificing the computational efficiency during model training. Here, we propose a computation-efficient semi-supervised learning paradigm (CE-SSL) for robust and computation-efficient CVDs detection using ECG. It enables a robust adaptation of pre-trained models on downstream datasets with limited supervision and high computational efficiency. First, a random-deactivation technique is developed to achieve robust and fast low-rank adaptation of pre-trained weights. Subsequently, we propose a one-shot rank allocation module to determine the optimal ranks for the update matrices of the pre-trained weights. Finally, a lightweight semi-supervised learning pipeline is introduced to enhance model performance by leveraging labeled and unlabeled data with high computational efficiency. Extensive experiments on four downstream datasets demonstrate that CE-SSL not only outperforms the state-of-the-art methods in multi-label CVDs detection but also consumes fewer GPU footprints, training time, and parameter storage space. As such, this paradigm provides an effective solution for achieving high computational efficiency and robust detection performance in the clinical applications of pre-trained models under limited supervision.

## 1 Introduction

Cardiovascular diseases have become the deadliest 'killer' of human health in recent years (Kelly et al., 2010). As a non-invasive and low-cost tool, ECG provides a visual representation of the electrical activity of the heart and is widely used in the detection of various CVDs (Kiyasseh et al., 2021a; Lai et al., 2023). Benefiting from recent progress in computing hardware, ECG-based deep learning systems have achieved notable success in automatic CVDs detection (Hannun et al., 2019; Ribeiro et al., 2020; Al-Zaiti et al., 2023; Lu et al., 2024b). However, previous deep learning models required sufficient labeled samples to achieve satisfactory performance when trained on new application scenarios with unseen CVDs (Berthelot et al., 2019; Sohn et al., 2020). Unfortunately, collecting well-labeled ECG recordings requires physicians' expertise and their laborious manual annotation, and therefore is expensive and time-consuming in clinical practice (Zhang et al., 2022; Zhou et al., 2023). Recent advancements in pre-trained models have enhanced the performance of deep learning models on the downstream datasets without large-scale labeled data (Vaswani et al., 2017; Radford et al., 2019; He et al., 2022). A commonly used pipeline consists of pre-training over-parameterized backbone models on large-scale datasets and then fine-tuning them on small downstream datasets in a supervised manner. However, two bottlenecks still greatly limit the clinical application of CVDs detection systems based on pre-trained models under limited supervision.

**(1) The bottleneck in CVDs detection performance.** Fine-tuning of pre-trained models is currently conducted in a purely supervised manner. When the labeled data is very scarce in the down-

stream datasets, model performance may drop due to over-fitting (Wang et al., 2021). Fortunately, a large amount of unlabeled data in the medical domain is relatively easy to collect. Semi-supervised learning (SSL) is able to extract sufficient information from the unlabeled data and outperform the supervised models trained with the same amount of labeled data (Zhou et al., 2018; Sohn et al., 2020; Li et al., 2021; Zhang et al., 2021; Peiris et al., 2023). For example, self-tuning integrates the exploration of unlabeled data and the knowledge transfer of pre-trained models into a united framework, which significantly outperforms supervised fine-tuning on five downstream tasks (Wang et al., 2021). Despite their robust performance, existing SSL methods are mainly built on pseudo-label techniques and the weak-strong consistency training on unlabeled samples (Berthelot et al., 2019; 2020; Sohn et al., 2020; Zhang et al., 2021; Chen et al., 2023a), which greatly increases the GPU memory footprint and computation time during model training. This drawback results in a bottleneck of computational efficiency during the performance enhancement of pre-trained models using semi-supervised learning.

**(2) The bottleneck in computational efficiency for parameter optimization.** Nowadays, many studies have introduced large-scale foundation models to achieve better CVDs detection performance using ECG (Vaid et al., 2023; Han & Ding, 2024; Mathew et al., 2024; McKeen et al., 2024; Pham et al., 2024; Jin et al., 2025), greatly increasing the computation costs of modifying them for downstream applications. SSL methods and fine-tuning both update all the model parameters. Despite their effectiveness, both methods have a main drawback that they require saving the gradients of all the model parameters and even the momentum parameters, resulting in large GPU memory footprints when tuning large pre-trained models (Hu et al., 2022). Additionally, each tuned model can be regarded as a full copy of the original models, therefore leading to high storage consumption when simultaneously tuned on multiple datasets (Zhang et al., 2023b). To address this, parameter-efficient fine-tuning (PEFT) methods have been introduced to reduce the trainable parameters during model training and thus decrease the computational costs during model training (Houlsby et al., 2019; Zaken et al., 2021; Chen et al., 2023b). For example, Low-rank adaptation (LoRA) achieves this goal by updating the pre-trained weights with low-rank decomposition matrices. AdaLoRA and IncreLoRA overcome the performance bottleneck of LoRA by allocating different ranks to different pre-trained weights based on their importance (Zhang et al., 2023b;a). However, the above improvement is achieved at the cost of increased training time for iterative importance estimation.

Therefore, a dilemma is encountered: model performance improvement often comes at the expense of a large sacrifice of computational efficiency during model training. Specifically, semi-supervised learning enhances CVDs detection performance under limited supervision but at significantly increased computational costs. Conversely, methods that prioritize computational efficiency may compromise model performance (Ding et al., 2023). Consequently, achieving a superior detection performance with high computation efficiency poses a great challenge to the clinical application of pre-trained models in ECG-based CVDs detection. To the best of our knowledge, no prior study has designed and evaluated a framework to escape the dilemma.

Here, we propose a united paradigm capable of addressing the above two bottlenecks simultaneously. It is a computation-efficient semi-supervised learning paradigm (CE-SSL) for adapting pre-trained models on downstream datasets with high computational efficiency under limited supervision. Our method enables robust and low-cost detection of CVDs in clinical practice using ECG recordings. As shown in Figure 1, first, a base backbone is pre-trained on a large-scale 12-lead ECG dataset in a supervised manner. We also provide medium and large backbones for performance enhancement by increasing the backbone's depth and width. Second, a random-deactivation low-rank adaptation (RD-LoRA) method formulates a low-cost and robust pipeline for updating the pre-trained backbone on downstream datasets. Specifically, it stochastically activates or deactivates low-rank adaptation in each trainable layer of the backbone with a probability $p$. To reduce GPU memory footprints, the pre-trained weights in each layer are always frozen. Theoretical analysis indicates that the random deactivation operation integrates various sub-networks generated during model training, thus overcoming the performance bottleneck in tuning pre-trained models. Additionally, deactivating low-rank adaptation in some layers reduces computation costs and speeds up the training process, especially when the backbone model size is large. Third, a one-shot rank allocation module determines the optimal ranks for the low-rank matrices in each layer. In contrast to AdaLoRA (Zhang et al., 2023b) and IncreLoRA (Zhang et al., 2023a), the proposed method can determine the optimal ranks using only one gradient backward iteration, improving the adaptation performance at low computational costs. Additionally, a lightweight semi-supervised learning

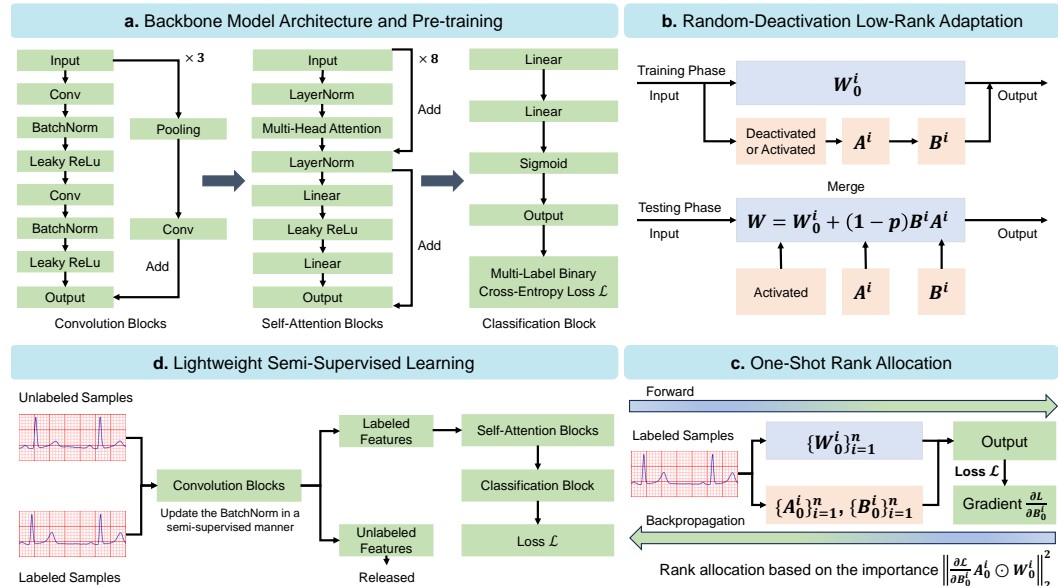

Figure 1: Overview of CE-SSL. **a** Backbones are pre-trained on a public 12-lead ECG dataset using the supervised multi-label binary cross-entropy loss. **b.** On the downstream datasets, the pre-trained weights in the backbones are updated by the random-deactivation low-rank adaptation. All the low-rank matrices are activated and merged into the pre-trained weights in the testing stage, which generates an ensemble network combining all the sub-networks produced by the random deactivation operation. **c.** The ranks of the low-rank matrices are determined by the proposed one-shot rank allocation method using only one gradient backward on the labeled samples. **d.** This lightweight semi-supervised pipeline improves the model performance in a computationally efficient way.

module is utilized to leverage the abundant information within unlabeled data. This module uses unlabeled data to stabilize the statistics estimation process in batch normalization layers, enhancing their generalization performance on unseen data distributions. Compared to the pseudo-labeling and the weak-strong consistency training methods (Sohn et al., 2020; Chen et al., 2023a), the module can alleviate the label scarcity problem with significantly higher computational efficiency.

Finally, extensive experiments on four downstream datasets demonstrate the superior CVDs detection performance of the proposed CE-SSL against various state-of-the-art models under very limited supervision. Most importantly, our method only requires 66.5% training time, 70.7% GPU memory footprint, and 1.8%-5.8% trainable parameters of the state-of-the-art SSL methods. Furthermore, its computational costs can be minimized to adapt to resource-limited environments without a significant accuracy loss. In conclusion, our proposed computation-efficient semi-supervised learning paradigm provides an effective solution to overcome the two bottlenecks that limit the clinical applications of pre-trained models in ECG-based CVDs detection. We summarize the major contributions as follows:

- A random deactivation low-rank adaptation method is proposed to update the backbones with high computational efficiency and robust performance.

- A one-shot rank allocation module is present to determine the optimal rank distribution during low-rank adaptation at minimal costs.

- A lightweight semi-supervised method is utilized to leverage large-scale unlabeled ECG data without greatly sacrificing computational efficiency.

- A computation-efficient semi-supervised framework for low-cost and accurate CVDs detection is proposed, which is the first one to escape the dilemma between model performance and computational efficiency.

## 2 METHODOLOGY

### 2.1 RANDOM-DEACTIVATION LOW-RANK ADAPTATION

Recent studies have demonstrated that low-rank adaptation (LoRA) can drastically decrease computation and storage costs in large-scale neural network fine-tuning while achieving promising performance on downstream tasks (Hu et al., 2022; Zhang et al., 2023b; Ding et al., 2023). The LoRA method models the incremental update of the pre-trained weights by the matrix multiplication of two low-rank matrices. For a hidden layer output $h = WX$, the LoRA forward process is defined as,

$$h = (W_0 + \triangle W)X = (W_0 + BA)X, \tag{1}$$

where $W_0, \triangle W \in \mathbb{R}^{d_1 \times d_2}$, $B \in \mathbb{R}^{d_1 \times r}$ and $A \in \mathbb{R}^{r \times d_2}$, and the rank $r \ll \min(d_1, d_2)$. The LoRA freezes the pre-trained weight $W_0$ during model training and only optimizes the low-rank matrices $A$ and $B$, which greatly reduces the number of trainable parameters during model training (Hu et al., 2022). However, the incremental updates of low-rank matrices are inadequate for achieving optimal performance on downstream datasets (Zi et al., 2023; Zhang et al., 2023a). To bridge the performance gap efficiently, we propose a novel random-deactivation low-rank adaptation (RD-LoRA) method, which randomly activates or deactivates the low-rank matrices in each trainable layer with a given probability $p$. To be specific, the forward process of the proposed RD-LoRA can be defined as,

$$h = (W_0 + \delta BA)X, \delta = \begin{cases} 1, & z \geq p \\ 0, & z < p \end{cases}, \tag{2}$$

where $\delta \sim Ber(\delta, 1 - p)$ can be regarded as a binary gate controlled by a random variable $z$ following a uniform distribution $U(0, 1)$, where $Ber$ indicates the Bernoulli distribution. $p$ is set to 0.2 in our experiments by default. In the training stage, the multi-label binary cross-entropy loss is employed for parameter optimization. In the testing stage, for input data $X_{test}$ and the pre-trained weight $W_0$, the expectation of $h_{test}$ given by RD-LoRA over $\delta$ can be calculated as,

$$\mathbb{E}_{\delta \sim Ber(\delta, 1-p)}[h_{test}] = \mathbb{E}_{\delta \sim Ber(\delta, 1-p)}[(W_0 + \delta BA)X_{test}]. \tag{3}$$

Considering that the low-rank matrices $\{A, B\}$ are fixed during the testing stage and $\delta$ is the only one random variable, thus,

$$\mathbb{E}_{\delta \sim Ber(\delta, 1-p)}[h_{test}] = (W_0 + \mathbb{E}_{\delta \sim Ber(\delta, 1-p)}[\delta] BA)X_{test} = (W_0 + (1-p)BA)X_{test}. \tag{4}$$

$$\mathbb{E}_{\delta \sim Ber(\delta, 1-p)}[h] \approx (W_0 + \mathbb{E}_{\delta \sim Ber(\delta, 1-p)}[\delta] BA)X = (W_0 + (1-p)BA)X. \tag{5}$$

Note that Eq.5 can only approximated the expectation of $h$ during the training stage, because $B$ and $A$ become variables and are not fully independent of $\delta$. This approximation is commonly used and works empirically (Huang et al., 2016; Srivastava et al., 2014). Similar to LoRA, the low-rank matrices are merged into the pre-trained weight $W_0$ in the testing stage to avoid extra inference costs, and the random-drop operation is deactivated. According to Eq.4, to ensure the expected output will be the same as the output with RD-LoRA, the merged matrix should be computed as,

$$W = W_0 + (1-p)BA. \tag{6}$$

After merging the low-rank matrices into the pre-trained weights of different layers, the final network can be viewed as an ensemble of all possible sub-networks during model training. In Appendix C.1, we provide a brief explanation to discuss how the random-deactivation module improves the model's generalization performance on unseen data during the test stage. Additionally, deactivating some low-rank matrices avoids the computation of update matrices in some layers, increasing training speed in the low-rank adaptation of large-scale models. The impact of $p$ on model performance and training speed is discussed in Appendix D.7.

### 2.2 EFFICIENT ONE-SHOT RANK ALLOCATION

Another limitation of LoRA is that it prespecifies the same rank for all low-rank incremental matrices, neglecting that their importance in model training varies across layers. In response to this limitation, AdaLoRA (Zhang et al., 2023a) and IncreLoRA (Zhang et al., 2023b) proposed to dynamically adjust the ranks of different incremental matrices during model training based on their

importance, which improved the low-rank adaptation performance. However, these dynamic methods require continuous calculation of the importance of all low-rank matrices in each iteration, significantly increasing the computation time. Additionally, their rank allocation processes are based on the singular value decomposition (SVD) theory and thus require an extra regularization loss to force the orthogonality of the low-rank matrices. This property introduces extra hyperparameters and computation costs. Here, we propose an efficient one-shot rank allocation method to overcome the computational inefficiency of the existing dynamic methods. Based on the first-order Taylor expansion, the importance of a weight matrix can be computed by the error induced by removing it from the network (Molchanov et al., 2019),

$$I(W^i) = \frac{1}{N_e} \sum_{j=1}^{N_e} (\mathcal{L}(Y, M(X)) - \mathcal{L}_{W^i(j)=0}(Y, M(X)))^2 \approx \left\| \frac{\partial \mathcal{L}(Y, M(X))}{\partial W^i} \odot W^i \right\|_2^2, \quad (7)$$

where $W^i(j)$ is the $j$-th element in the weight matrix $W^i$, $N_e$ is the number of elements in $W^i$ and $\odot$ is the Hadamard product. However, the gradient matrix $\frac{\partial \mathcal{L}(Y, M(X))}{\partial W^i}$ can not be obtained because $W^i$ is frozen during the low-rank training process. Here, we approximate it using its incremental update $\triangle W^i$, which can be computed by low-rank matrices $A^i$ and $B^i$ using Eq.1.

$$\frac{\partial \mathcal{L}(Y, M(X))}{\partial W^i} \propto -\frac{1}{\eta}\triangle W^i = \frac{1}{\eta}(B_t^i A_t^i - B_{t+1}^i A_{t+1}^i)$$

$$(8)$$

$$= B_t^i \frac{\partial \mathcal{L}(Y, M(X))}{\partial A_t^i} + \frac{\partial \mathcal{L}(Y, M(X))}{\partial B_t^i} A_n^i - \eta \frac{\partial \mathcal{L}(Y, M(X))}{\partial B_t^i} \frac{\partial \mathcal{L}(Y, M(X))}{\partial A_t^i},$$

where $A_t^i$ and $B_t^i$ are the low-rank matrices at training round $t$, constant $\eta$ is the learning rate and $W_0$ is the pre-trained weight. Although Eq.8 enables importance score estimation during model training, iterative matrix multiplication induces a heavy computation burden. Hence, we propose to simplify the estimation function Eq.8 and compute the importance score in a 'one-shot' manner. Specifically, we only use the first gradient-backpropagation process to achieve the entire rank allocation process and fix the ranks of different low-rank matrices during the remaining training iterations. In the first backpropagation process, the low-rank matrices $\{A^i\}_{i=1}^n$ are initialized from a normal distribution $N(0, \sigma^2)$ and $\{B^i\}_{i=1}^n$ are initialized to zero. Consequently, the gradient of $\{A^i\}_{i=1}^n$ at the 0-th (first) iteration is zero according to Eq.1. Based on the above initialization conditions, Eq.8 at the 0-th iteration can be rewritten as,

$$\frac{\partial \mathcal{L}(Y, M(X))}{\partial W_0^i} \propto -\frac{1}{\eta}\triangle W_0^i = \frac{\partial \mathcal{L}(Y, M(X))}{\partial B_0^i} A_0^i, \frac{\partial \mathcal{L}(Y, M(X))}{\partial A_0^i} = 0, B_0^i = 0, \quad (9)$$

where $\{W_0^i\}_{i=1}^n$ are the pre-trained weight matrices in the backbone model $M(X)$. Then, the importance score of the pre-trained weight $W_0^i$ can be approximated as,

$$I(W_0^i) \approx \hat{I}(W_0^i) = \left\| \left( \frac{\partial \mathcal{L}(Y, M(X))}{\partial B_0^i} A_0^i \right) \odot W_0^i \right\|_2^2. \quad (10)$$

Eq.10 is computed using the labeled samples from the downstream dataset, which estimates the importance of $W_0^i$ during fine-tuning. Then, we sort the importance $\hat{I}(W_0^i)$ of all pre-trained matrices in descending order and allocate different ranks for their low-rank matrices. Here, we assume the ranks of the incremental matrices corresponding to the important weights should be higher than those of the incremental matrices associated with the unimportant weights. The allocated rank $r^i$ of the incremental matrices of the pre-trained weight $W_0^i$ is defined as,

$$r^i = \begin{cases} r, & \hat{I}(W_0^i) \text{ in the top-}k \text{ of } \{\hat{I}(W_0^i)\}_{i=1}^n \\ \frac{1}{2}r, & \text{otherwise} \end{cases}, k = nc, \quad (11)$$

where $r$ is an initial rank, and $c$ is a hyper-parameter that controls the number of important weight matrices. The impacts of $r$ and $c$ on model performance are discussed in Appendix D.8 and D.10, respectively. Note that the allocated ranks $\{r^i\}_{i=1}^n$ are fixed during the remaining iterations, and the low-rank matrices ($\{B^i\}_{i=1}^n$, $\{A^i\}_{i=1}^n$) are reset based on their allocated ranks. Eq.10 is only computed at the 0-th iteration, which avoids numerous matrix multiplications. In addition, the proposed rank allocation process does not require constraints on the orthogonality of low-rank matrices. In summary, the above advantages allow the proposed method to have a faster training speed compared to AdaLoRA (Zhang et al., 2023a) and IncreLoRA (Zhang et al., 2023b).

## 2.3 LIGHTWEIGHT SEMI-SUPERVISED LEARNING

Semi-supervised learning (SSL) is an efficient tool for model performance enhancement when large-scale unlabeled data is available (Chapelle et al., 2006; Berthelot et al., 2019). Recently, many studies utilized label guessing and consistency regularization to further improve the model performance in SSL tasks, such as FixMatch (Sohn et al., 2020), FlexMatch (Zhang et al., 2021) and SoftMatch (Chen et al., 2023a). However, the above two techniques require the output predictions of the weak and strong-augmented unlabeled samples, which induces extra computation costs. Consequently, traditional SSL methods usually exhibit much higher memory costs and longer training time than naive supervised models. Here, we utilize a lightweight but effective SSL method without extensive consistency training and pseudo-label guessing. Motivated by Koçyigit et al. (2020), we can update the batch normalization (BN) layers in a semi-supervised manner using both labeled and unlabeled data. Subsequently, the unlabeled data is released, and only the labeled data is forwarded to the self-attention and classification blocks for loss computation. Different from Koçyigit et al. (2020), we integrate normalization and parameter optimization into one forward-backward step to avoid extra computational costs. For labeled inputs $\{x_b^i\}_{i=1}^{N_B}$ and unlabeled inputs $\{x_u^i\}_{i=1}^{N_U}$, the mean value $\mu$ and the variance $\sigma$ of the semi-supervised BN layers in the convolution blocks can be updated as,

$$\mu = \frac{\gamma}{N_B} \sum_{i=1}^{N_B} x_b^i + \frac{1-\gamma}{N_U} \sum_{i=1}^{N_U} x_u^i, \sigma = \frac{\gamma}{N_B} \sum_{i=1}^{N_B} (x_b^i - \mu)^2 + \frac{1-\gamma}{N_U} \sum_{i=1}^{N_U} (x_u^i - \mu)^2, \quad (12)$$

where $N_B$ and $N_U$ are the numbers of labeled and unlabeled samples in the current mini-batch, and $\gamma = \frac{N_B}{N_B+N_U}$. Note that $N_B$ equals $N_U$ in this study, thus $\gamma = 0.5$. The impact of $N_B : N_U$ on model performance is discussed in Appendix D.11. With only limited labeled data $x_b$, the estimated mean $\mu_B = \frac{1}{N_B} \sum_{i=1}^{N_B} x_b^i$ and variance $\sigma_B = \frac{1}{N_B} \sum_{i=1}^{N_B} (x_b^i - \mu_B)^2$ in traditional BN are prone to be influenced by the over-fitting problem according to the law of large numbers. On the contrary, semi-supervised BN can alleviate the problem by utilizing large-scale unlabeled data $x_u$ for parameter estimation, which improves the model performance on unseen distributions. Since the BN layers do not exist in the self-attention and classification blocks, we only forward the labeled features to them to reduce memory cost and training time. Compared with the SOTA methods in semi-supervised learning, the proposed CE-SSL discards the label guessing and the consistency regularization modules. However, the results demonstrate that it achieves comparable CVDS detection performance to the SOTA methods on four downstream ECG datasets, while achieving less memory consumption and faster training speed. Finally, we present the pseudo-code of CE-SSL in Appendix Algorithm 1.

## 3 EXPERIMENTS AND DATASETS

As shown in Appendix C.2, the base, medium, and large backbones are pre-trained on a public and large-scale dataset collected by Ribeiro et al. (2019; 2020), which have 9.505 million, 50.494 million, and 113.490 million parameters, respectively. Subsequently, we use four downstream datasets for model fine-tuning and evaluation: the Georgia 12-lead ECG Challenge (G12EC) database (Alday et al., 2020), the Chapman-Shaoxing database (Zheng et al., 2020b), the Ningbo database (Zheng et al., 2020a), and the Physikalisch-Technische Bundesanstalt (PTB-XL) database (Wagner et al., 2020). Specifically, the G12EC database contains 10344 ECG recordings from 10,344 people, and the PTB-XL database comprises 21837 recordings from 18885 patients. The Chapman database contains 10,646 recordings from 10646 patients, and the Ningbo database encompasses 40258 recordings from 40258 patients. Only 34,905 recordings in the Ningbo database are publicly available (Alday et al., 2020). The recordings from the four downstream databases are around 10 seconds, and the sampling rate is 500 Hz. Additionally, each database contains over 17 different CVDs, and multiple CVDs can be identified from one ECG segment simultaneously.

The pre-trained backbones are fine-tuned on the four downstream datasets using different methods under limited supervision. Taking the G12EC database as an example, the ECG recordings are split into a training set and a held-out test set in a ratio of 0.9: 0.1. Then, the training set is divided into a labeled training set and an unlabeled training set in a ratio of 0.05: 0.95. How the ratio of training labeled data impacts model performance is discussed in Appendix D.12. A validation set is randomly sampled from the labeled training set and accounts for 20% of it, which is used for selecting the best-performing model during training. For model comparisons, we reproduce several baseline models in

Table 1: Performance comparisons of CE-SSL and semi-supervised baselines on the base backbone. The average performance and the standard deviation of different metrics are shown across six seeds.

| Methods | G12EC Dataset | | PTB-XL Dataset | | Ningbo Dataset | | Chapman Dataset | |
|---|---|---|---|---|---|---|---|---|
| | Macro $G_{\beta=2}$ | Macro $F_{\beta=2}$ | Macro $G_{\beta=2}$ | Macro $F_{\beta=2}$ | Macro $G_{\beta=2}$ | Macro $F_{\beta=2}$ | Macro $G_{\beta=2}$ | Macro $F_{\beta=2}$ |
| MixedTeacher | 0.275±0.016 | 0.507±0.025 | 0.316±0.007 | 0.542±0.014 | 0.324±0.018 | 0.549±0.028 | 0.327±0.019 | 0.510±0.024 |
| FixMatch | 0.280±0.010 | 0.510±0.016 | 0.322±0.007 | 0.541±0.007 | 0.321±0.014 | 0.545±0.020 | 0.339±0.012 | 0.518±0.025 |
| FlexMatch | 0.274±0.019 | 0.497±0.035 | 0.316±0.008 | 0.536±0.007 | 0.318±0.012 | 0.544±0.019 | 0.325±0.010 | 0.495±0.019 |
| SoftMatch | 0.276±0.017 | 0.504±0.021 | 0.317±0.009 | 0.540±0.011 | 0.321±0.014 | 0.552±0.020 | 0.335±0.011 | 0.511±0.021 |
| Adsh | 0.268±0.009 | 0.489±0.013 | 0.322±0.008 | 0.543±0.015 | 0.318±0.010 | 0.545±0.012 | 0.335±0.013 | 0.517±0.020 |
| SAW | 0.269±0.018 | 0.494±0.024 | 0.323±0.019 | 0.548±0.017 | 0.314±0.010 | 0.536±0.016 | 0.333±0.012 | 0.510±0.020 |
| **CE-SSL$_{r=16}$** | **0.307±0.016** | **0.551±0.017** | **0.346±0.006** | **0.578±0.006** | **0.334±0.011** | **0.569±0.014** | **0.355±0.005** | **0.530±0.008** |
| **CE-SSL$_{r=4}$** | **0.304±0.013** | **0.553±0.020** | **0.346±0.005** | **0.580±0.006** | **0.327±0.010** | **0.567±0.011** | **0.352±0.009** | **0.530±0.012** |

semi-supervised learning: FixMatch (Sohn et al., 2020), FlexMatch (Zhang et al., 2021), SoftMatch (Chen et al., 2023a), MixedTeacher (Zhang et al., 2022), Adsh (Guo & Li, 2022), SAW (Lai et al., 2022). Additionally, we integrate the state-of-the-art parameter-efficient methods (LoRA (Hu et al., 2022), DyLoRA (Valipour et al., 2023), AdaLoRA (Zhang et al., 2023b), IncreLoRA (Zhang et al., 2023a)) with FixMatch for comparisons.

## 4  RESULTS AND DISCUSSION

### 4.1  ANALYSIS OF THE CVDs DETECTION RESULTS

We comprehensively evaluate the model performance of various methods using multiple metrics and training costs. Since multiple CVDs can be detected from one recording simultaneously, we used metrics on multi-label classification, such as macro $G_{\beta=2}$ score, and macro $F_{\beta=2}$ score. In Appendix D, we also include ranking loss, coverage, mean average precision (MAP), and macro AUC for comprehensive comparisons. Additionally, we report the training costs of different methods, including the peak GPU memory footprint during model training (Mem), the number of trainable parameters (Params), and the average training time for each optimization iteration (Time/iter). The higher the number of trainable parameters, the higher the parameter storage consumption. Note that the number of trainable parameters of CE-SSL can be adjusted by the initial rank $r$. Lower ranks indicate fewer trainable parameters. The AdamW optimizer (Loshchilov & Hutter, 2017) is used under a learning rate of 1e-3. By default, the batch sizes for labeled and unlabeled data are both 64 for all the compared methods. All the experiments are conducted in a single NVIDIA A6000 graphics processing unit using the Pytorch library.

Table 1 and Table 2 show that CE-SSL achieves superior detection performance on four downstream datasets with the lowest computational costs compared with the SOTA methods. In Table 13, we present a detailed model comparison using more metrics. In the G12EC dataset, CE-SSL with $r = 16$ achieves a macro $F_{\beta=2}$ of 0.551±0.017, which is 4.1% larger than the second-best model's (FixMatch) performance. In Appendix D.1, we present the detection performance of different models on each CVD. The results demonstrate that CE-SSL ranks the best in some CVDs, such as atrial fibrillation and first-degree AV block. It also achieves comparable performance to the compared methods in the remaining CVDs. Regarding computational costs, it requires 33.5% less training time than MixedTeacher, occupies 29.3% less GPU memory than Adsh, and has only 5.8% of the trainable parameters found in them. When the initial rank $r$ decreases to 4, CE-SSL shows a slight performance drop in four datasets, but the number of trainable parameters further decreases to 1.8% of the baseline models. This observation indicates the stability and robustness of the CE-SSL under extremely low parameter budgets. As shown in Appendix D.2, we demonstrate that the superiority of CE-SSL persists on the medium and the large backbones. Additionally, its robustness under resource-limited environments is also illustrated in Appendix D.4.

We further compare the proposed CE-SSL with the parameter-efficient methods, which are integrated with FixMatch for parameter-efficient semi-supervised learning. For example, FixMatch with AdaLoRA is denoted as 'FixMatch+AdaLoRA'. Similar to CE-SSL, their budgets for the number of trainable parameters are controlled by the initial rank $r$. At the same time, the lightweight semi-

Table 2: Computational efficiency of CE-SSL and semi-supervised baselines on the base backbone.

| Methods | G12EC Dataset | | | PTB-XL Dataset | | | Ningbo Dataset | | | Chapman Dataset | | |
|---|---|---|---|---|---|---|---|---|---|---|---|---|
| | Params | Mem | Time/iter | Params | Mem | Time/iter | Params | Mem | Time/iter | Params | Mem | Time/iter |
| MixedTeacher | 9.505 M | 3.941 GB | 147 ms | 9.505 M | 3.941 GB | 164 ms | 9.506 M | 3.941 GB | 173 ms | 9.504 M | 3.941 GB | 148 ms |
| FixMatch | 9.505 M | 5.784 GB | 187 ms | 9.505 M | 5.784 GB | 208 ms | 9.506 M | 5.784 GB | 217 ms | 9.504 M | 5.784 GB | 186 ms |
| FlexMatch | 9.505 M | 5.784 GB | 187 ms | 9.505 M | 5.784 GB | 209 ms | 9.506 M | 5.784 GB | 217 ms | 9.504 M | 5.784 GB | 185 ms |
| SoftMatch | 9.505 M | 5.784 GB | 187 ms | 9.505 M | 5.784 GB | 209 ms | 9.506 M | 5.784 GB | 217 ms | 9.504 M | 5.784 GB | 187 ms |
| Adsh | 9.505 M | 3.887 GB | 207 ms | 9.505 M | 3.887 GB | 316 ms | 9.506 M | 3.887 GB | 423 ms | 9.504 M | 3.887 GB | 207 ms |
| SAW | 9.505 M | 5.784 GB | 188 ms | 9.505 M | 5.784 GB | 208 ms | 9.506 M | 5.784 GB | 215 ms | 9.504 M | 5.784 GB | 185 ms |
| **CE-SSL$_{r=16}$** | **0.510 M** | **2.747 GB** | **98 ms** | **0.582 M** | **2.748 GB** | **110 ms** | **0.550 M** | **2.748 GB** | **115 ms** | **0.581 M** | **2.748 GB** | **97 ms** |
| **CE-SSL$_{r=4}$** | **0.183 M** | **2.743 GB** | **98 ms** | **0.159 M** | **2.744 GB** | **109 ms** | **0.168 M** | **2.744 GB** | **114 ms** | **0.180 M** | **2.743 GB** | **97 ms** |

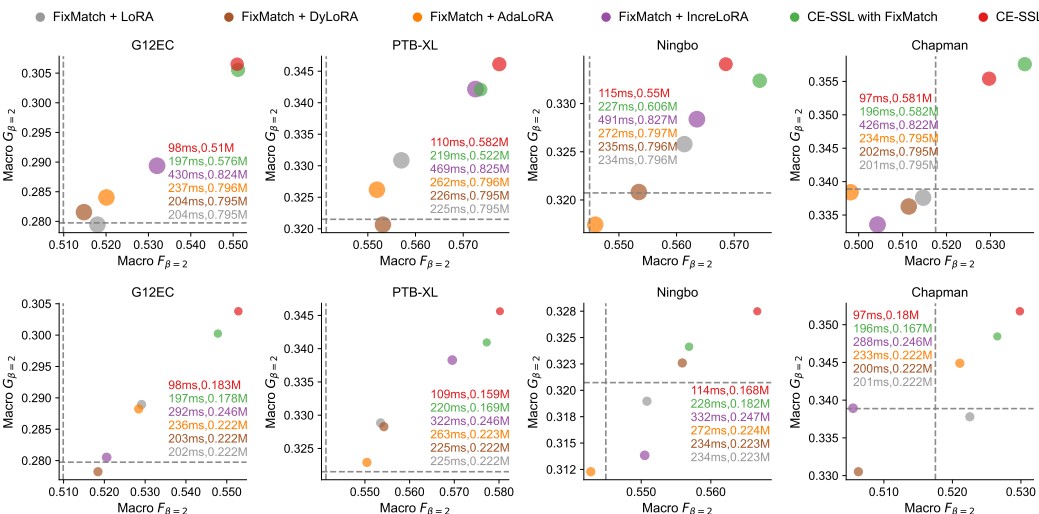

Figure 2: Comparison between CE-SSL and parameter-efficient semi-supervised methods on the base backbone. Circles with various colors denote different models, and their size represents the number of trainable parameters. The training time for each optimization iteration (Time/iter) of different methods is also reported. The gray dotted lines represent the performance of the FixMatch baseline without parameter-efficient training (approximately 9.505M trainable parameters). The first row of the figure presents the performance of different models with sufficient parameter budgets ($r = 16$), while the second row reports their performance under limited parameter budgets ($r = 4$).

supervised learning module within CE-SSL is replaced with FixMatch for comparison, denoted as 'CE-SSL with FixMatch'. As illustrated in Figure 2, we report their macro-$F_{\beta=2}$ scores, macro-$G_{\beta=2}$ scores, and Time/iter on four datasets at sufficient ($r = 16$) and limited ($r = 4$) budget levels. The macro-$F_{\beta=2}$ and macro-$G_{\beta=2}$ scores of the FixMatch without parameter-efficient training are denoted as gray dotted lines. The experiment results indicate that CE-SSL consistently outperforms the other methods on four datasets at different budget levels. Under a sufficient parameter budget ($r = 16$), CE-SSL achieves a macro-$G_{\beta=2}$ score of 0.307±0.016 on the G12EC dataset, which is 2.8% higher than the FixMatch with LoRA. When the parameter budget is limited ($r = 4$), CE-SSL still outperforms it by 1.5%. In Table 16, we present detailed comparison results on more evaluation metrics, which provide supplementary evidence on the efficiency of the proposed CE-SSL in CVDs detection. Paired t-tests are conducted to evaluate the significance levels of the performance difference between CE-SSL and the aforementioned SOTA methods (Figure 7). Based on the calculated two-sided $p$-value, it can be observed that CE-SSL outperforms the baselines at a 0.05 significance level in most datasets and evaluation metrics, which indicates a significant superiority for the proposed CE-SSL framework.

At the same time, with one-shot rank allocation, the proposed RD-LoRA is generally better than other low-rank adaptation methods when integrated with FixMatch. Under a sufficient parameter

Table 3: Ablation study using the base backbone.'RA: One-Shot Rank Allocation', 'RD: Random Deactivation', 'SSBN: Semi-Supervised BN', 'FixM: FixMatch'.

| Methods | G12EC Dataset | | | PTB-XL Dataset | | | Ningbo Dataset | | | Chapman Dataset | | |
|---|---|---|---|---|---|---|---|---|---|---|---|---|
| | Params | Time/iter | Macro $F_{\beta=2}$ | Params | Time/iter | Macro $F_{\beta=2}$ | Params | Time/iter | Macro $F_{\beta=2}$ | Params | Time/iter | Macro $F_{\beta=2}$ |
| LoRA | 0.795M | 78ms | 0.520±0.011 | 0.795M | 87ms | 0.537±0.008 | 0.796M | 92ms | 0.546±0.019 | 0.795M | 77ms | 0.499±0.014 |
| +RA | 0.510M | 81ms | 0.522±0.030 | 0.582M | 91ms | 0.554±0.008 | 0.550M | 96ms | 0.549±0.007 | 0.581M | 81ms | 0.521±0.013 |
| +RD | 0.795M | 76ms | 0.530±0.024 | 0.795M | 83ms | 0.558±0.010 | 0.796M | 88ms | 0.566±0.018 | 0.795M | 74ms | 0.515±0.013 |
| +SSBN | 0.795M | 99ms | 0.530±0.013 | 0.795M | 111ms | 0.558±0.012 | 0.796M | 116ms | 0.557±0.021 | 0.795M | 100ms | 0.512±0.011 |
| +SSBN+RA | 0.510M | 104ms | 0.536±0.021 | 0.582M | 115ms | 0.554±0.011 | 0.550M | 121ms | 0.553±0.017 | 0.581M | 102ms | 0.514±0.015 |
| +SSBN+RD | 0.795M | 97ms | 0.537±0.019 | 0.795M | 108ms | 0.560±0.014 | 0.796M | 114ms | 0.563±0.014 | 0.795M | 96ms | 0.514±0.018 |
| +RA+RD | 0.510M | 78ms | 0.536±0.029 | 0.582M | 87ms | 0.565±0.007 | 0.550M | 92ms | 0.559±0.018 | 0.581M | 77ms | 0.527±0.026 |
| +RA+RD+FixM | 0.576M | 197ms | 0.551±0.016 | 0.522M | 219ms | 0.574±0.006 | 0.606M | 227ms | 0.574±0.008 | 0.582M | 196ms | 0.538±0.014 |
| **CE-SSL** | 0.510M | 98ms | 0.551±0.017 | 0.582M | 110ms | 0.578±0.006 | 0.550M | 115ms | 0.569±0.014 | 0.581M | 97ms | 0.530±0.008 |

budget ($r = 16$), 'CE-SSL with FixMatch' achieves an average macro-$G_{\beta=2}$ score of 0.334 and average macro-$F_{\beta=2}$ score of 0.559 across four datasets, outperforming 'FixMatch + LoRA' by 1.6% and 2.2%. Under a tight parameter budget ($r = 4$), 'CE-SSL with FixMatch' achieves an average macro-$G_{\beta=2}$ score of 0.325 and average macro-$F_{\beta=2}$ score of 0.546 across four datasets, outperforming 'FixMatch + DyLoRA' by 1.3% and 1.8%. Additionally, it achieves the highest training speed and the best performance with the least trainable parameters compared to other parameter-efficient semi-supervised learning frameworks. In summary, the experiments demonstrate the robustness and computational efficiency of the CE-SSL in cardiovascular disease detection under limited supervision. In other words, CE-SSL can enhance the detection performance of ECG-based CVDs detection models without introducing heavy computation burdens.

## 4.2 Ablation Study

As shown in Table 3, we conducted an ablation study to evaluate the contribution of the modules in CE-SSL. Specifically, we add the one-shot rank allocation (RA), random deactivation (RD), and semi-supervised BN (SSBN) to LoRA and record the corresponding model performance. Note that the initial rank $r$ is set to 16 for LoRA. **(1) The random-deactivation low-rank adaptation improves model performance and computational efficiency.** In the G12EC dataset, removing it from CE-SSL decreases the macro $F_{\beta=2}$ from 0.551±0.017 (CE-SSL) to 0.536±0.021 (+SSBN+RA). Its effectiveness can also be supported by directly adding it to LoRA, where macro $F_{\beta=2}$ increases from 0.537±0.008 (LoRA) to 0.558±0.010 (+RD) in the PTB-XL dataset. Additionally, the Time/iter is slightly reduced compared with LoRA when the deactivation probability $p$ is set to 0.2. As shown in Appendix D.7, the Time/iter can be significantly reduced by increasing $p$. As illustrated in Appendix D.5, the improvements in training speed become more significant on larger backbones. **(2) The one-shot rank allocation improves fine-tuning performance.** In the PTB-XL dataset, removing it from CE-SSL decreases the macro $F_{\beta=2}$ from 0.578±0.006 (CE-SSL) to 0.560±0.014 (+SSBN+RD). It can be observed that it does not introduce heavy computational burdens (Time/iter only increases by 1-2ms) while further reducing the number of trainable parameters, demonstrating its high computational efficiency. Its effectiveness can also be supported by directly adding it to full fine-tuning, where macro $F_{\beta=2}$ increases from 0.499±0.014 (LoRA) to 0.521±0.013 (+RA) in the Chapman dataset. (3) **The proposed lightweight semi-supervised learning benefits model performance without greatly increasing the training time.** It utilizes the unlabeled data to stabilize the statistics within the BN layers in the convolution blocks, preventing them from over-fitting to small amounts of labeled data. Removing it from CE-SSL decreases its detection performance on all the datasets. Compared to RA+RD+FixMatch, such as FixMatch, the extra computational costs caused by SSBN are significantly lower, while their detection performance is comparable.

## 4.3 External Validation

In this section, an external validation is conducted using the medium backbone, where four downstream datasets (G12EC, Chapman, PTB-XL, Ningbo) are used for fine-tuning, and one held-out dataset provided by (Lai et al., 2023) is used for evaluation. Additional results for the base and large backbones are presented in Appendix D.6. We integrate the proposed one-shot rank allocation

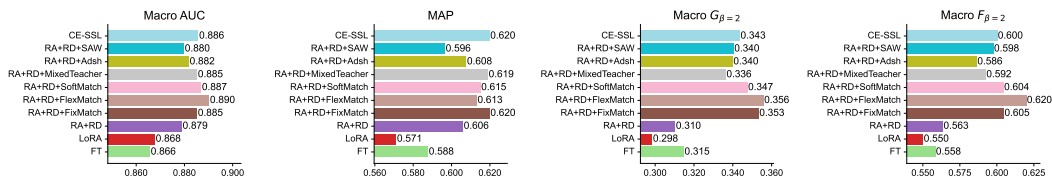

Figure 3: External validation results. 'RA: One-Shot Rank Allocation', 'RD: Random Deactivation'

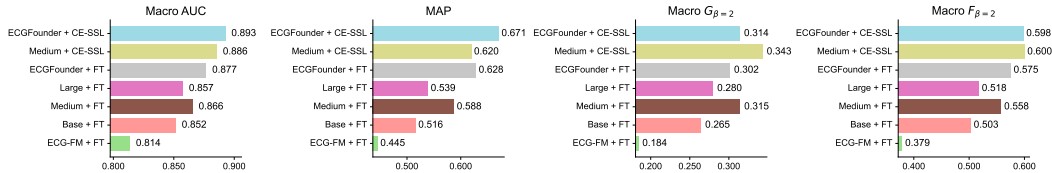

Figure 4: Effect of the pretrained backbones on the model generalization performance.

(RA) and random-deactivation low-rank adaptation modules (RD) as a powerful supervised baseline (RA+RD). Subsequently, we combine the baseline with various semi-supervised methods, including FixMatch, FlexMatch, SoftMatch, Adsh, SAW, MixedTeacher, as well as the proposed semi-supervised BN (CE-SSL). In addition, LoRA and full fine-tuning (FT) are also used for benchmarking. We calculate the average detection performance across six random seeds. More implementation details are provided in Appendix D.6. The experiment results provide two critical insights. First, the CVDs detection system powered by CE-SSL can generalize well to unseen data collected by different devices and medical centers. As shown in Figure 3, CE-SSL achieves a macro AUC of 0.886 on the external dataset and demonstrates better cross-distribution robustness than LoRA (0.866) and FT (0.868). Second, the semi-supervised BN enhances the model's generalization performance on unseen data without introducing heavy computational burdens. Specifically, CE-SSL outperforms the powerful supervised baseline (RA+RD) by 3.67% on macro $F_{\beta=2}$ score. It achieves comparable performance to the baseline with SOTA semi-supervised methods and demonstrates significantly lower computational costs (Table 7).

Pretrained backbones are critical factors that determine the generalization performance of the fine-tuned models. We fine-tune different pretrained backbones using full fine-tuning (FT) on four downstream datasets and evaluate them on the external dataset. We include five backbones for benchmarking, including the backbones (base, medium, large) provided in our study and two external backbones: ECG-FM (McKeen et al., 2024) and ECGFounder (Li et al., 2025). As shown in Figure 4, ECGFounder and our medium backbone demonstrate the best and the second-best generalization performance on the external dataset. More importantly, we can observe that CE-SSL can consistently improve their performance, which indicates its effectiveness across various backbones.

## 5 CONCLUSION

Bottlenecks in model performance and computational efficiency have become great challenges in the clinical application of CVDs detection systems based on pre-trained models, especially when the supervised information is scarce in the downstream ECG datasets. In this paper, we propose a computationally efficient semi-supervised learning paradigm (CE-SSL) for adapting pre-trained models on downstream datasets with limited supervision and high computational efficiency. Experiment results on four downstream ECG datasets and three backbone settings indicate that CE-SSL achieves superior CVDs detection performance and computational efficiency compared to state-of-the-art methods. In conclusion, our study offers a fast and robust semi-supervised learning paradigm for ECG-based CVDs detection under limited supervision. It provides a feasible solution for efficiently adapting pre-trained models to downstream ECG datasets. We hope this learning paradigm will pave the way for the application of automatic CVDs detection systems and broaden their applicability to various ECG-based tasks.

## REPRODUCIBILITY STATEMENT

To ensure reproducibility, we give a derivation of the ensemble optimization properties of the RD-LoRA in Appendix C.1. All datasets used in our experiments are publicly available and clearly specified in Section 3. The evaluation metrics used in our experiments are defined in Appendix C.4. We provide the algorithm of the proposed CE-SSL in Algorithm 1 and the anonymous source codes in a supplementary .zip file. The pretrained backbones with various sizes will be released after publication.

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

## A  THE USE OF LARGE LANGUAGE MODELS (LLMS)

In this paper, LLMs are only used as tools for spell checking and grammar suggestions. The authors take full responsibility for the contents within the paper.

## B  APPENDIX: RELATED WORK

### B.1  AI-ENABLED CVDS PREDICTION USING ECG

Benefiting from the development of deep learning, AI-enabled systems have shed light on automatic ECG screening and cardiovascular disease diagnosis (Pourbabaee et al., 2018; Hannun et al., 2019; Ribeiro et al., 2020; Strodthoff et al., 2020; Kiyasseh et al., 2021b; Huang et al., 2022; Vaid et al., 2023; Han & Ding, 2024; Mathew et al., 2024). Tracing the development of the systems, it can be observed that the prediction models they used are continuously scaling up. In the first stage, small-scale models demonstrated promising diagnosis performance in ECG analysis and CVDs detection. For example, Pourbabaee et al. (2018) designed a deep convolutional neural network to extract features from ECG signals and utilize standard classifiers for screening paroxysmal atrial fibrillation. Hannun et al. (2019) proposed an end-to-end deep convolutional neural network to achieve automatic single-lead ECG screening. The results demonstrated that the network achieved similar diagnosis performance compared with common cardiologists. In the second stage, pre-trained models with a prohibitive number of parameters were introduced, which demonstrated better transferability than previous networks. This advantage reduces their requirement for supervision information on downstream datasets. For instance, Vaid et al. (2023) pre-trained a large-scale vision transformer (HeartBEiT) on a huge ECG dataset and fine-tuned it on downstream datasets. The experiment results demonstrated the superiority of HeartBEiT in CVDs detection compared with traditional CNN architectures. In the current stage, many studies have proposed various kinds of foundation models for more advanced ECG screening and cardiac healthcare, inspired by their success in natural language processing (Han & Ding, 2024; Mathew et al., 2024). However, pre-trained models might experience a performance drop on downstream datasets when the labeled samples are very scarce there. Additionally, the computational costs of adapting them to various tasks significantly increase as their sizes scale up.

### B.2  SEMI-SUPERVISED LEARNING FOR PERFORMANCE ENHANCEMENT UNDER LIMITED SUPERVISION.

Semi-supervised learning offers an effective solution to address the label scarcity problem by leveraging unlabeled samples (Wang et al., 2021; Berthelot et al., 2019; Sohn et al., 2020; Zhang et al., 2021; Chen et al., 2023a). For example, Sohn et al. (2020) combined consistency regularization and pseudo-labeling to formulate a powerful algorithm (FixMatch). Extensive experiments demonstrate the superiority of FixMatch against the supervised baselines under the label scarcity condition. Subsequently, Zhang et al. (2021) proposed curriculum pseudo labeling (CPL) to flexibly adjust the thresholds for pseudo label selection, aiming at utilizing unlabeled data based on the model's training progress. Using a truncated Gaussian function, Chen et al. (2023a) designed a soft threshold to weight unlabeled samples according to their prediction confidence, which achieved a balance between pseudo-label quality and quantity. Compared with FixMatch, FlexMatch, and SoftMatch demonstrate better performance in various datasets. However, Wang et al. (2021) pointed out that the performance of semi-supervised models will be influenced by inaccurate pseudo labels, especially in a large label space. Hence, they proposed a self-tuning technique to explore the potential of the transfer of pre-trained models and a pseudo-label group contrast mechanism to increase the model's tolerance to inaccurate labels. Experiments on five tasks demonstrated the superiority of the proposed framework against previous semi-supervised and supervised methods. In summary, massive unlabeled data and powerful pre-trained models led to the success of semi-supervised methods. However, high computation burdens are the side effects of leveraging them, greatly limiting their applications in resource-limited settings.

### B.3 PARAMETER-EFFICIENT METHODS FOR HIGHER COMPUTATIONAL EFFICIENCY.

Parameter-efficient training has demonstrated great potential in decreasing the computational costs of fine-tuning pre-trained models (Zaken et al., 2021; Hu et al., 2022; Valipour et al., 2023; Zhang et al., 2023b). For example, Zaken et al. (2021) proposed BitFit to fine-tune the bias terms of the pre-trained models and freeze the other parameters, greatly reducing the computational costs. However, BitFit sacrifices the performance of the fine-tuned models because most of their parameters are not well adapted to downstream tasks. Hu et al. (2022) designed a low-rank adaptation method (LoRA) to inject trainable low-rank matrices into the transformer architecture, decreasing the performance gap between parameter-efficient methods and full fine-tuning. However, Zhang et al. (2023b) pointed out that LoRA ignored the varying importance of different pre-trained weights and allocated the same rank for all the trainable matrices, which led to suboptimal fine-tuning performance. Consequently, they designed AdaLoRA to address this problem, which dynamically allocates different ranks to the low-rank matrices according to their importance during fine-tuning. During this process, the trainable parameters of the matrices with low importance are pruned. Different from AdaLoRA, IncreLoRA adaptively adds trainable parameters to the low-rank matrices with high importance (Zhang et al., 2023a). As a non-pruning method, its performance is not limited by the preset parameter budget. Although IncreLoRA and AdaLoRA surpass LoRA in some scenarios, they result in high computation costs for weight importance estimation. Consequently, advancing fine-tuning performance without sacrificing computational efficiency remains challenging when designing parameter-efficient methods.

# C APPENDIX: METHODOLOGY DETAILS

## C.1 ENSEMBLE OPTIMIZATION PROPERTIES OF THE RD-LoRA

In this section, we briefly analyze the ensemble properties of the proposed RD-LoRA. Here, we simply consider a network $M$ with $n$ fully-connected layers, defined as $M(X) = \prod_{i=1}^{n} W_0^i X$, where $X$ is the input data and $W_0^i \in \mathbb{R}^{c_{out} \times c_{in}}$ is the pre-trained weight matrix at the $i$-th layer. During model training, a convex loss function $\mathcal{L}(Y, M(X))$ is employed for parameter optimization. When the RD-LoRA is activated, the expectation of the loss function $\mathbb{E}_{\delta \sim Ber(\delta, 1-p)}\left[\mathcal{L}(Y, M(X))\right]$ at the iteration $t$ can be given as,

$$
\begin{aligned}
\mathbb{E}_{\delta \sim B(\delta, 1-p)}^{t}\left[\mathcal{L}(Y, M(X))\right] = {} & (1-p)^n \mathcal{L}(Y, \prod_{i=1}^{n}(W_0^i + B_t^i A_t^i)X) \\
& + \sum_{j=1}^{n}\left[p(1-p)^{n-1}\mathcal{L}(Y, \prod_{i=1, i\neq j}^{n}(W_0^i + B_t^i A_t^i)W_0^j)X\right] \\
& + \cdots + p^n \mathcal{L}(Y, \prod_{i=1}^{n} W_0^i X),
\end{aligned}
\tag{13}
$$

where the low-rank matrices $\{A_t^i\}_{i=1}^{n}$ and $\{B_t^i\}_{i=1}^{n}$ are trainable while the pre-trained weights $\{W_0^i\}_{i=1}^{n}$ are frozen. Eq.13 can be regarded as a weighted mean of the losses of $2^n$ sub-networks, which are minimized during model training. The number of activated low-rank matrices $np$ of the sub-networks is lower than the entire network $n$. Consequently, the training costs of the sub-networks are lower than those of the entire network. In the testing stage, all the low-rank matrices are merged into the pre-trained weights, which generates an ensemble model combining all the possible sub-networks. After that, the low-rank matrices $\{A, B\}$ are fixed and only $\delta$ is a random variable. Hence, given the testing data $X_{test}$ and the ground truth $Y_{test}$, the testing loss can be estimated as

$$
\mathcal{L}(Y_{test}, \mathbb{E}_{\delta \sim Ber(\delta, 1-p)}\left[M(X_{test})\right]) = \mathcal{L}(Y_{test}, \prod_{i=1}^{n}(W_0^i + (1-p)B^i A^i)X_{test}).
\tag{14}
$$

In this paper, the multi-label binary cross-entropy loss with sigmoid activation $\sigma(M(X)) = [\sigma(M(X))_1, \sigma(M(X))_2, \cdots \sigma(M(X))_C]$ is convex according to the second-order condition of convexity, where $C$ is the number of categories. Specifically, the Hessian matrix of $\mathcal{L}(Y, \sigma(M(X)))$ is diagonal and the $c$-th element of the main diagonal can be given as,

$$
\frac{\partial^2 \mathcal{L}(Y, \sigma(M(X)))}{\partial M(X)_c^2} = \sigma(M(X))_c(1 - \sigma(M(X))_c) \geq 0,
\tag{15}
$$

where $Y = [y_1, y_2, \cdots y_C]$, $y_c \in \{0, 1\}$ and $\sigma(M(X)) = (1 + e^{-M(X)})^{-1}$. According to Eq 15, the Hessian matrix of $\mathcal{L}(Y, \sigma(M(X)))$ is positive semidefinite, demonstrating the convexity of the loss function. Based on Jensen's inequality, the loss of any ensemble average is smaller than the average loss of the ensemble components,

$$
\mathcal{L}(Y_{test}, \mathbb{E}_{\delta \sim Ber(\delta, 1-p)}\left[M(X_{test})\right]) \leq \mathbb{E}_{\delta \sim Ber(\delta, 1-p)}\left[\mathcal{L}(Y_{test}, M(X_{test}))\right].
\tag{16}
$$

In the training stage, the proposed RD-LoRA optimizes the parameters of multiple sub-networks and generates an ensemble network in the testing stage, improving the model performance on the testing data.

## C.2 BACKBONE MODEL PRE-TRAINING

The base backbone model is pre-trained on a public 12-lead ECG dataset (CODE-15% (Ribeiro et al., 2019; 2020)), where 345779 ECG recordings from 233770 patients are provided. The medium and large backbones are pre-trained on a restricted dataset with 2,322,513 ECG recordings from 1,558,772 patients (CODE-full (Ribeiro et al., 2019; Lu et al., 2024a)). The specific settings of the backbone models with different sizes are shown in Table 4. Note that multiple abnormalities could be identified from one ECG recording simultaneously, which indicates that a multi-label classification

---

**Algorithm 1** CE-SSL algorithm

---

**Require:**
- Labeled dataset $D_B = \{X_b, Y_b\}$ and unlabeled dataset $D_U = \{X_u\}$;
- Pre-trained model $M_0 = \{W_0^i\}_{i=1}^n$; Initial rank $r$; The ratio of important weights $c$; The random-deactivation probability $p$; Batch sizes of the labeled samples ($N_B = 64$) and the unlabeled samples ($N_U = 64$).

**Ensure:** Adapted model $M$ with the updated parameters $\{W^i = W_0^i + (1-p)A^iB^i\}_{i=1}^n$;
1: One-shot rank allocation
2: Compute the importance of each pre-trained weight using the Eq.10 and the labeled dataset $D_B$;
3: Based on the initial rank $r$ and the ratio $c$, allocate the final rank $r^i$ of the incremental matrices $(A^i, B^i)$ of the pre-trained weight $W_0^i$ using Eq.11.
4: **for** 1 to $iteration$ **do**
5:     sample labeled data $\{x_b, y_b\}$ from $D_B$;
6:     sample unlabeled data $\{x_u\}$ from $D_U$;
7:     apply data augmentation to $x_b$ and $x_u$;
8:     Lightweight semi-supervised learning
9:     Based on Eq.12, update the semi-supervised batch-normalization layers in the convolution blocks using the labeled data $x_b$ and the unlabeled data $x_u$.
10:    release the unlabeled data $x_u$ in the GPU memory
11:    Random-deactivation low-rank adaptation
12:    initialize $h_0 = x_b$
13:    **for** $i = 1, 2, ...n$ **do**
14:        sample $\delta_i$ from the Bernoulli distribution $B(\delta, 1-p)$
15:        $h_i = (W_0^i + \delta_i B^i A^i)h_{i-1}$
16:    **end for**
17:    Based on the model output $h_n$ and the ground-truth $y_b$, compute the supervised multi-label binary cross-entropy loss using Eq.17. Apply an early-stop strategy to avoid overfitting.
18: **end for**
19: Merge the incremental matrices into the pre-trained weights, as $\{W^i = W_0^i + (1-p)B^iA^i\}_{i=1}^n$;

---

model should be implemented for ECG-based CVDs detection. As shown in Figure 1, the backbone model $M(X)$ consists of three parts: (1) Convolution blocks, (2) Self-attention blocks, and (3) Classification blocks. Specifically, the convolution blocks comprise multiple convolution layers (Conv) and batch normalization layers. The Leaky-Relu function is used as the activation function and skip-connection is implemented (Nejedly et al., 2021). In addition, a simple but efficient self-attention pipeline is employed in the self-attention blocks (Radford et al., 2019) and two successive fully-connected layers with sigmoid activation are used for label prediction in the classification block. A multi-label binary cross-entropy function is employed for model training, defined as,

$$\mathcal{L}(Y, M(X)) = -\frac{1}{BC}\sum_{i=1}^{B}\sum_{c=1}^{C}(1 - y_{i,c})\log(1 - p_{i,c}) + y_{i,c}\log p_{i,c}, \tag{17}$$

where $X = \{x_i\}_{i=1}^B, x_i \in \mathbb{R}^{12 \times L}$ are the ECG recordings in the current mini-batch, $L$ is the signal length and $Y = \{y_i\}_{i=1}^B$ is the corresponding ground truths. $p_{i,c}$ is the model prediction on class $c$ and $C$ is the number of categories. During model training, a held-out validation set is used for early-stop model validation. The best-performing model on the validation set is used for downstream tasks on small-scale datasets.

## C.3 SIGNAL PRE-PROCESSING AND DATA AUGMENTATION

Artifact removal and data augmentation are two factors that play important roles in model performance. Firstly, we introduce the signal pre-processing pipeline employed in the proposed framework. The ECG recordings from the CODE-15% and CODE-full databases are first resampled to a 400Hz sampling rate following the configuration of the dataset provider (Ribeiro et al., 2020). The sampling rate of the recordings from the four downstream databases remains unchanged. Firstly, the length of all recordings is normalized into 6144 samples by zero-padding. Subsequently, a band-pass filter (1-47Hz) is applied to remove the power-line interference and baseline drift. Then, the

Table 4: Backbone model specifications. $N_{conv}$ indicates the number of convolution blocks, $N_{att}$ indicates the number of self-attention blocks, and $N_{cls}$ indicates the number of classification blocks. $C$ is the number of convolution channels. Hidden size is the hidden layer dimension of the self-attention blocks. Head Num is the number of heads in multi-head self-attention. Params is the total number of parameters in the backbone.

| Backbone Size | $N_{conv}$ | $N_{att}$ | $N_{cls}$ | $C$ | Hidden size | Head Num | Params |
|---|---|---|---|---|---|---|---|
| Base | 3 | 8 | 1 | 256 | 256 | 16 | 9.505M |
| Medium | 3 | 12 | 1 | 512 | 512 | 16 | 50.494M |
| Large | 3 | 12 | 1 | 768 | 768 | 16 | 113.490M |

Table 5: Description of the cardiovascular diseases analyzed in our study. The abbreviations (Abb) and the total number of instances (Nums) of a certain class are denoted as 'Abb (Nums)'.

| Original annotation | Abb (Nums) | Original annotation | Abb (Nums) |
|---|---|---|---|
| **G12EC Dataset** | | | |
| atrial fibrillation | AF (570) | 1st degree av block | IAVB (769) |
| incomplete right bundle branch block | IRBBB (407) | left axis deviation | LAD (940) |
| left bundle branch block | LBBB (231) | low qrs voltages | LQRSV (374) |
| nonspecific intraventricular conduction disorder | NSIVCB (203) | sinus rhythm | NSR (1752) |
| premature atrial contraction | PAC (639) | prolonged qt interval | LQT (1391) |
| qwave abnormal | QAb (464) | right bundle branch block | RBBB (542) |
| sinus arrhythmia | SA (455) | sinus bradycardia | SB (1677) |
| sinus tachycardia | STach (1261) | t wave abnormal | TAb (2306) |
| t wave inversion | TInv (812) | ventricular premature beats | VPB (357) |
| **PTB-XL Dataset** | | | |
| atrial fibrillation | AF (1514) | complete right bundle branch block | CRBBB (542) |
| 1st degree av block | IAVB (797) | incomplete right bundle branch block | IRBBB (1118) |
| left axis deviation | LAD (5146) | left anterior fascicular block | LAnFB (1626) |
| left bundle branch block | LBBB (536) | nonspecific intraventricular conduction disorder | NSIVCB (789) |
| sinus rhythm | NSR (18092) | premature atrial contraction | PAC (398) |
| pacing rhythm | PR (296) | prolonged pr interval | LPR (340) |
| qwave abnormal | QAb (548) | right axis deviation | RAD (343) |
| sinus arrhythmia | SA (772) | sinus bradycardia | SB (637) |
| sinus tachycardia | STach (826) | t wave abnormal | TAb (2345) |
| t wave inversion | TInv (294) | | |
| **Ningbo Dataset** | | | |
| atrial flutter | AFL (7615) | bundle branch block | BBB (385) |
| complete left bundle branch block | CLBBB (213) | complete right bundle branch block | CRBBB (1096) |
| 1st degree av block | IAVB (893) | incomplete right bundle branch block | IRBBB (246) |
| left axis deviation | LAD (1163) | left anterior fascicular block | LAnFB (380) |
| low qrs voltages | LQRSV (794) | nonspecific intraventricular conduction disorder | NSIVCB (536) |
| sinus rhythm | NSR (6299) | premature atrial contraction | PAC (1054) |
| pacing rhythm | PR (1182) | poor R wave Progression | PRWP (638) |
| premature ventricular contractions | PVC (1091) | prolonged qt interval | LQT (337) |
| qwave abnormal | QAb (828) | right axis deviation | RAD (638) |
| sinus arrhythmia | SA (2550) | sinus bradycardia | SB (12670) |
| sinus tachycardia | STach (5687) | t wave abnormal | TAb (5167) |
| t wave inversion | TInv (2720) | | |
| **Chapman Dataset** | | | |
| atrial fibrillation | AF (1780) | atrial flutter | AFL (445) |
| 1st degree av block | IAVB (247) | left axis deviation | LAD (382) |
| left bundle branch block | LBBB (205) | low qrs voltages | LQRSV (249) |
| nonspecific intraventricular conduction disorder | NSIVCB (235) | sinus rhythm | NSR (1826) |
| premature atrial contraction | PAC (258) | qwave abnormal | QAb (235) |
| right axis deviation | RAD (215) | right bundle branch block | RBBB (454) |
| sinus bradycardia | SB (3889) | sinus tachycardia | STach (1568) |
| t wave abnormal | TAb (1876) | ventricular premature beats | VPB (294) |

pre-processed signals are normalized using z-score normalization. Secondly, CutMix (Yun et al., 2019) is employed for labeled data augmentation. Since the sample generation process of CutMix requires true labels that are absent in the unlabeled data, we employed the ECGAugment (Zhou et al., 2023) for unlabeled data augmentation, which generates new samples by randomly selecting a

transformation to perturb the pre-processed signals. Note that only the weak-augmentation module in the ECGAugment is employed.

## C.4 Evaluation Metrics

In the model evaluation section, we evaluate the CVDs detection performance of different models using six metrics: ranking loss, coverage, mean average precision (MAP), macro AUC, macro $G_{beta}$, and macro $F_{beta}$. Here, we provide detailed descriptions of how to compute the metrics based on the model predictions $P = M(X), P \in \mathbb{R}^{N \times C}$ and the multi-label ground truths $Y \in \mathbb{R}^{N \times C}$. $N$ is the sample size and $C$ is the number of categories. Each row $y_n = \left[y_n^1, y_n^2, \cdots, y_n^C\right], y_n^C \in \{0, 1\}$ in $Y$ indicates the multi-label ground-truth of sample $n$. Specifically, if $y_n^1 == 1, y_n^2 == 1, y_n^3 == 0$, sample $n$ belongs to class 1 and class 2 simultaneously, but it does not belong to class 3. Each row $p_n = \left[p_n^1, p_n^2, \cdots, p_n^C\right], p_n^C \in [0, 1]$ in $P$ indicates the multi-label CVDs predictions of sample $n$.

(1) The Ranking Loss calculates the average count of label pairs that are reversely ordered (Zhang & Zhou, 2013; Tsoumakas et al., 2010). For given predictions $P$ and ground-truth $Y$, it is weighted by the size of the label set and the number of labels not in the label set. The best performance is achieved with a ranking loss of zero. The computation process of the ranking loss can be found in Zhang & Zhou (2013).

(2) The coverage evaluates the steps needed to go through the ranked label list to cover all the ground-truth labelsZhang & Zhou (2013); Tsoumakas et al. (2010). The smaller the coverage is, the better the performance. The best value is the average number of positive labels in $Y$ per sample. The computation process of the coverage can also be found in Zhang & Zhou (2013).

(3) Macro AUC calculates the average Area Under Curve (AUC) across all the CVDs categories, defined as

$$\text{Macro AUC} = \frac{1}{C} \sum_{c=1}^{C} \text{AUC}_c, \tag{18}$$

where $\text{AUC}_c$ is AUC on CVD class $c$. The higher the Macro AUC is, the better the performance. The best performance is achieved with a ranking loss of one.

(4) MAP indicates the mean average precision across all CVDs. The computation process of the average precision on a given class can also be found in Zhang & Zhou (2013). The higher the MAP is, the better the performance. The best performance is achieved with a ranking loss of one.

(5) Macro $F_{\beta=2}$ calculates the average $F_{\beta=2}$ score across all the CVDs categories, defined as

$$\text{Macro } F_{\beta=2} = \frac{1}{C} \sum_{c=1}^{C} F_{\beta=2}^c, \tag{19}$$

$$F_\beta = \frac{\left(1 + \beta^2\right) \text{TP}}{\left(1 + \beta^2\right) \text{TP} + \text{TP} + \beta^2 \text{FN}} \tag{20}$$

where $F_{\beta=2}^c$ is $F_{\beta=2}$ score on CVD class $c$. TP represents the number of true positive predictions, while FN represents the number of false negative predictions. The $\beta$ value is set to 2 for all the corresponding experiments following the configurations provided in Strodthoff et al. (2020). The higher the macro $F_{\beta=2}$ is, the better the performance. The best performance is achieved with a macro $F_{\beta=2}$ of one.

(6) Macro $G_{\beta=2}$ calculates the average $G_{\beta=2}$ score across all the CVDs categories, defined as

$$\text{Macro } G_{\beta=2} = \frac{1}{C} \sum_{c=1}^{C} G_{\beta=2}^c, \tag{21}$$

$$G_\beta = \frac{\text{TP}}{\text{TP} + \text{FP} + \beta \text{FN}} \tag{22}$$

where $G_{\beta=2}^c$ is $G_{\beta=2}$ score on CVD class $c$. FP represents the number of false positive predictions. The $\beta$ value is set to 2 for all the corresponding experiments following the configurations provided in Strodthoff et al. (2020). The higher the macro $G_{\beta=2}$ is, the better the performance. The best performance is achieved with a macro $G_{\beta=2}$ of one.

## D APPENDIX: EXTENDED EXPERIMENTS

### D.1 DETAILED MODEL PERFORMANCE FOR EACH CVD

Here, we provide the detailed model performance for each CVD using the base backbone. The CVDs analyzed in our study can be found in Table 5. Note that different datasets contain various CVD classes, and there is a class imbalance issue with all datasets. Then, we report the $F_{\beta=2}$ score of each compared model on each CVD class. We also present the macro $F_{\beta=2}$ score, which is an average of the $F_{\beta=2}$ score across all CVDs. In this section, state-of-the-art methods in semi-supervised learning are used for comparisons, including FixMatch (Sohn et al., 2020), FlexMatch (Zhang et al., 2021), SoftMatch (Chen et al., 2023a), MixedTeacher (Zhang et al., 2022), Adsh (Guo & Li, 2022), SAW (Lai et al., 2022). The experiment results on four datasets are shown in Table 9, Table 10, Table 11 and Table 12. Compared with other semi-supervised models, CE-SSL demonstrates the best detection performance in some CVDs and achieves on-par performance in the remaining CVDs.

### D.2 PERFORMANCE COMPARISONS UNDER VARIOUS BACKBONE SIZES

In the previous sections, we have already proved the robustness and computation efficiency of the proposed CE-SSL under a base backbone with 9.505 million parameters. Here, we compare its performance with other baseline models under medium and large backbones, which share the same architecture as the base backbone but have more parameters (Table 4). Specifically, the medium backbone has 50.494 million parameters, and the large backbone has 113.490 million parameters. They are pre-trained on the CODE-full dataset, a huge but restricted ECG dataset with 2,322,513 ECG recordings from 1,558,772 patients (Ribeiro et al., 2019; 2020). In Table 14 and 15, we report the performance of CE-SSL and semi-supervised baselines on the medium and the large backbones, respectively. The results demonstrate that CE-SSL achieves similar and even better CVDs detection performance than the semi-supervised baselines and exhibits the lowest computation costs. For example, using the medium backbone, CE-SSL achieves a macro $F_{\beta=2}$ of 0.599±0.010, which is 3.7% larger than the second-best model's (SAW) performance in the PTB-XL dataset. Using the large backbone, CE-SSL achieves a macro $F_{\beta=2}$ of 0.565±0.010 in the G12EC dataset, outperforming SAW by 3.1%. Regarding the computational costs, the number of trainable parameters of CE-SSL is 0.9% to 3.1% of the other baselines on the medium backbone and 0.6% to 2.1% on the large backbone. In addition, CE-SSL demonstrates the lowest GPU memory consumption and the highest training speed compared to the other semi-supervised baselines. For the memory footprint, CE-SSL achieves an average GPU memory usage of 6.16 GB using the medium backbone and 9.22 GB using the large backbone, 3.09 GB and 4.59 GB less than the second-best model (Adsh). Furthermore, CE-SSL achieves an average training time per iteration of 259.25 ms using the medium backbone and 485.5 ms using the large backbone, 162.5 ms and 289.75 ms faster than the second-best model (MixedTeacher). These phenomena demonstrate that as the number of model parameters increases, the computational efficiency advantage of CE-SSL over other models becomes increasingly apparent. In Table 17 and Table 18, we present the performance of CE-SSL and parameter-efficient semi-supervised methods on the medium and large backbones, respectively. It can be observed that CE-SSL outperforms the other models in CVDs detection on both medium and large backbones. Additionally, CE-SSL demonstrates the fastest training speed across four datasets compared with other parameter-efficient methods. In Figure 8 and Figure 9, we provide the paired t-test results of the model performance on the two backbones. The statistical results indicate that CE-SSL outperforms the above baselines in ECG-based CVDs detection at a 0.05 significance level in most conditions.

### D.3 DETAILED RESULTS ON STATISTICAL ANALYSIS

In this section, we provide detailed statistical analysis results to evaluate the significance levels of the performance difference between CE-SSL and the aforementioned baselines using different backbones. Applying paired t-tests, we compare their performance on four datasets and present the two-sided $p$-value in Figure 7, Figure 8 and Figure 9. For each dataset, the model performance under six random seeds is used for the paired t-tests. Note that the initial ranks for LoRA, Dy-LoRA, AdaLoRA, IncreLoRA, and CE-SSL are set to 16. Based on the calculated $p$-value, it can

be observed that CE-SSL outperforms the baselines at a 0.05 significance level in most datasets and evaluation metrics, which indicates a significant superiority for the proposed CE-SSL framework.

## D.4 TOWARD HIGHER COMPUTATIONAL EFFICIENCY IN CLINICAL PRACTICES

Although deploying the CE-SSL paradigm with the base backbone on low-level devices (4-6 GB GPU memory) is easy, implementing the paradigm with the medium and large backbones is still challenging. To overcome this limitation, we adopt a simple but effective approach to boost the computational efficiency of the CE-SSL. Specifically, we freeze the first two convolution blocks in the backbones during the CE-SSL training process. The new paradigm is denoted as 'CE-SSL-F' in the following analysis. We present the CVDs detection performance and the computational efficiency of CE-SSL-F, CE-SSL, and the SOTA methods in semi-supervised learning in Figure 10. Note that the batch sizes for all the compared methods are set to 64. The initial rank for CE-SSL and CE-SSL-F is set to 16 and 4, respectively.

First, freezing the convolution blocks greatly reduces the cached activation during the forward pass, significantly decreasing the GPU memory footprints. As shown in Figure 10a, it can be observed that the CE-SSL-F requires nearly 50% less GPU memory footprints compared to the CE-SSL, generalizing its applications in low-level devices (NVIDIA RTX 3050 laptops and RTX 4060 GPU cards). Specifically, CE-SSL-F is deployable on RTX 3050 laptops with both base and medium backbones, and it is the only method that can be implemented on the RTX 4060 GPU cards with a large backbone. In contrast, deploying the CE-SSL with a large backbone requires medium-level devices (NVIDIA RTX 4070 GPU cards), while other semi-supervised methods require high-level devices with GPU memory larger than 12 GB. Second, the parameters of the frozen blocks are not updated during the backward pass, which increases the training speed of CE-SSL-F. The larger the backbone is, the more parameters are frozen, and thus the more gradient backward time is saved. As shown in Figure 10b, CE-SSL-F demonstrates the fastest training speed compared with other models, and its advantages become more significant along with the increase in backbone sizes. Third, CE-SSL-F only sacrifices 1-2% CVDs detection performance compared with CE-SSL. More importantly, it consistently outperforms the other semi-supervised methods across different backbones (Figure 10c), demonstrating its effectiveness in CVDs detection. This phenomenon can be explained by the strong transferability of the pre-trained convolution blocks located in the first few layers of the backbone (Sharif Razavian et al., 2014; Tajbakhsh et al., 2016). Specifically, they mainly contain domain-invariant knowledge for CVDs detection, and their parameters will not be changed significantly during the fine-tuning process. Therefore, freezing them does not greatly decrease the model performance. In summary, the experiment results illustrate that the computational efficiency of the CE-SSL can be increased to adapt to low-level devices without losing its superior CVDs detection performance compared to other semi-supervised methods. This advantage demonstrates CE-SSL's flexibility in different clinical application scenarios with various computational resources.

## D.5 EXTENDED RESULTS ON ABLATION STUDY

In this section, we provide the ablation study of CE-SSL using medium and large backbones in Table 20 and Table 21. Note that the initial rank $r$ is 16 for all the compared models. (1) It can be observed that removing the random-deactivation technique from CE-SSL increases the Time/iter and decreases the CVDs detection performance on the four datasets. For example, with the medium backbone, the Time/iter increases from 243ms to 259ms and the macro $F_{beta}$ decreases from $0.561\pm0.024$ to $0.540\pm0.022$ on the G12EC database. With the large backbone, the Time/iter increases from 451ms to 480ms and the macro $F_{beta}$ decreases from $0.552\pm0.018$ to $0.529\pm0.021$ on the Chapman database. (2) It is demonstrated that the one-shot rank allocation increases the detection performance with high computation efficiency. For instance, with the medium backbone, the macro $F_{\beta=2}$ increases from $0.515\pm0.022$ to $0.540\pm0.019$, and the MAP increases from $0.537\pm0.010$ to $0.553\pm0.013$ on the Chapman dataset. With the large backbone, the macro $F_{\beta=2}$ increases from $0.562\pm0.019$ to $0.587\pm0.008$, and the macro $G_{\beta=2}$ increases from $0.340\pm0.016$ to $0.358\pm0.005$ on the PTB-XL database. More importantly, the proposed method completes the rank allocation process without introducing high computational costs (Time/iter only increases by 1-7ms). (3) Removing the lightweight semi-supervised learning module from CE-SSL decreases the CVDs diagnostic performance on different backbone sizes. With the medium backbone, the macro $F_{\beta=2}$ score decreases from $0.588\pm0.021$ to $0.576\pm0.024$ and macro $G_{\beta=2}$ decreases from

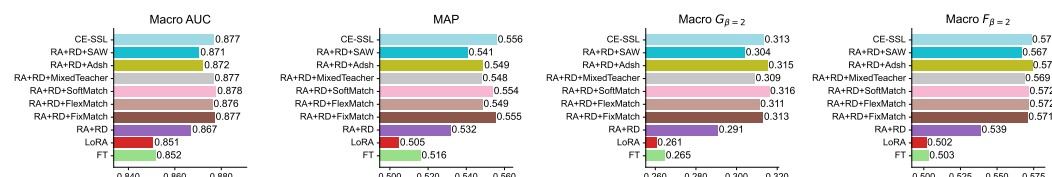

Figure 5: External validation results (base backbone). 'RA: One-Shot Rank Allocation', 'RD: Random Deactivation'

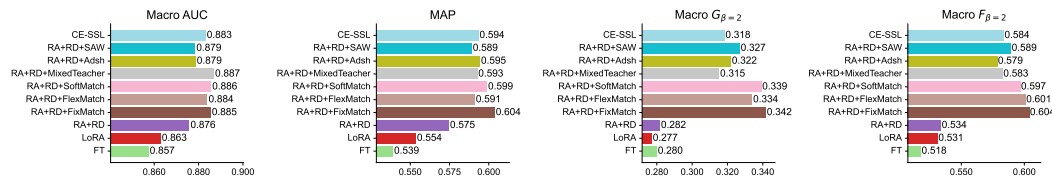

Figure 6: External validation results (large backbone). 'RA: One-Shot Rank Allocation', 'RD: Random Deactivation'

$0.356\pm0.013$ to $0.346\pm0.018$ on the Ningbo dataset. With the large backbone, the macro $F_{\beta=2}$ score decreases from $0.565\pm0.010$ to $0.552\pm0.018$ and macro $G_{\beta=2}$ decreases from $0.322\pm0.009$ to $0.314\pm0.014$ on the G12EC dataset.

### D.6 DETAILS ABOUT EXTERNAL VALIDATION

A main advantage of semi-supervised learning is increasing the model's generalization performance on unseen samples, especially when the labeled data is expensive to collect. Here, we conduct an external validation on the model trained by various methods to highlight the contribution of introducing semi-supervised BN for improving the model's generalization performance. Specifically, we combine the G12EC, PTB-XL, Ningbo, and Chapman datasets as a joint dataset for model training. It is divided into a labeled training set and an unlabeled training set in a ratio of 0.05: 0.95. An internal validation set is randomly sampled from the labeled training set and accounts for 20% of it, which is used for selecting the best-performing model during training. Then, an external validation set provided by (Lai et al., 2023) is used to evaluate the model's generalization performance on unseen samples, which contains 7000 wearable 12-lead ECG recordings. The CVDs that co-exist in the external dataset and the joint dataset are used for evaluation, including NSR, QAb, TAb, IAVB, BBB, CRBBB, IRBBB, CLBBB, SB, SA, PAC, AF, AFL, PVC, and PR. For all the compared methods, the batch sizes of labeled and unlabeled data for CE-SSL and all compared SOTA SSL methods are set to 64 ($N_B : N_U = 1 : 1$). The deactivation probability for random deactivation low-rank adaptation is set to $p = 0.2$ and the initial rank $r$ for one-shot rank allocation is set to 16. All the compared methods are equipped with a medium backbone for training. In terms of fine-tuning costs before validation (Table 6), top semi-supervised methods (FixMatch, FlexMatch) increase the training time per iteration and GPU memory consumption by 2.27 times and 2.88 times, respectively. In contrast, semi-supervised BN only increases the training time per iteration and GPU memory consumption by 1.18 times and 1.41 times, demonstrating higher computation efficiency.

### D.7 EFFECT OF THE DEACTIVATION PROBABILITY

For each pre-trained weight $W_0^i$ in the CE-SSL, the proposed RD-LoRA deactivates its low-rank matrices ($A^i, B^i$) in the current iteration at a probability of $p$, which produces multiple sub-networks during model training. All the low-rank matrices are activated in the testing stage, generating an ensemble network that combines all the sub-networks. Consequently, the probability $p$ is an important parameter that controls the training time and the final performance of the proposed CE-SSL. In Figure 11, we adjust $p$ from 0.1 to 0.5 and present the averaged model performance across four datasets, including the training time for each iteration. Note that the labeled ratio is set to 5%, and the initial ranks for all the low-rank matrices are set to 16. The results show that the CE-SSL with

Table 6: Fine-tuning efficiency of different methods before external validation (base backbone).

| Method | RA+RD | +FixMatch | +FlexMatch | +SoftMatch | +MixedTeacher | +Adsh | +SAW | CE-SSL |
|---|---|---|---|---|---|---|---|---|
| Time/iter | 110 ms | 250 ms | 250 ms | 250 ms | 200 ms | 180 ms | 240 ms | 130 ms |
| Memory | 1.95 GB | 5.62 GB | 5.62 GB | 5.62 GB | 3.82 GB | 3.77 GB | 5.62 GB | 2.75 GB |

Table 7: Fine-tuning efficiency of different methods before external validation (medium backbone).

| Method | RA+RD | +FixMatch | +FlexMatch | +SoftMatch | +MixedTeacher | +Adsh | +SAW | CE-SSL |
|---|---|---|---|---|---|---|---|---|
| Time/iter | 240 ms | 640 ms | 630 ms | 630 ms | 520 ms | 440 ms | 630 ms | 310 ms |
| Memory | 4.71 GB | 12.96 GB | 12.96 GB | 12.96 GB | 8.97 GB | 8.76 GB | 12.96 GB | 6.16 GB |

$p = 0.2$ demonstrates the best detection performance compared with the model with other settings. In addition, it can be observed that the training time of the CE-SSL decreases as $p$ increases. The reason is that the larger the $p$ is, the more low-rank matrices are deactivated during model training, which speeds up the forward-backward propagation.

### D.8    RANK INITIALIZATION IN THE ONE-SHOT RANK ALLOCATION

Rank initialization is an important component in low-rank adaptation, which controls the number of trainable parameters during model training. In this section, we adjust the initial rank from 4 to 32 and present the averaged model performance on the four datasets in Figure 12. Note that the labeled ratio is set to 5%. The results indicate that CE-SSL with high initial ranks ($r = 16, 32$) achieves better performance than that with low initial ranks ($r = 4, 8$). This is because the model with higher ranks has more trainable parameters and thus demonstrates a larger capacity during training.

### D.9    EFFECT OF WARM-UP EPOCHS FOR RANK ALLOCATION

Once the initial rank is determined, the proposed one-shot rank allocation module will determine the optimal ranks for the update matrices of the pre-trained weights using Eq.10. The allocation process only utilizes the gradient information at the 0-th (first) iteration. It is worth discussing whether determining the optimal ranks before fine-tuning would hinder the model's performance or not. Here, we first fine-tune the pre-trained model for $T$ warm-up epochs using LoRA with the initial rank $r = 16$. Then, we determine the optimal ranks using the fine-tuned parameters and the one-shot rank allocation. As shown in Figure 13, we adjust $T$ from 0 to 3 and compute the averaged model performance on four downstream datasets. Note that the labeled ratio is set to 5%. The results demonstrate that increasing the number of warmup epochs has a limited impact on the performance of the proposed CE-SSL. It can be observed that the fluctuations of macro $F_{\beta=2}$ score and macro $G_{\beta=2}$ score are within 0.4% when $T$ increases from 0 to 3. It indicates that determining the optimal ranks before fine-tuning would not hinder the model's performance. Specifically, during the pre-training and fine-tuning stages, the models' training objectives are correlated. Hence, the pre-trained models carry rigorous information for the downstream tasks. Additionally, the importance of each low-rank matrix is calculated using the labeled samples from the downstream datasets, which provide sufficient information for effective rank allocation without extra warm-up epochs.

### D.10    EFFECT OF THE NUMBER OF IMPORTANT WEIGHT MATRICES

Based on the proposed one-shot rank allocation, CE-SSL allocates a rank $r$ to the incremental matrices with high importance and a rank $r/2$ to the matrices with low importance. The ratio of the important matrices to the total number of pre-trained matrices is defined as the coefficient $c$. The higher the coefficient is, the higher the ratio of the important matrices. In Figure 14, we adjust the coefficient from 0.2 to 0.8 and report the averaged model performance across four datasets. Note that the labeled ratio is set to 5%, and the initial ranks $r$ for all the low-rank matrices are set to 16. It can be observed that the performance of the proposed model is relatively insensitive to the changes in the $c$. In Figure 18, we visualize the rank distribution generated by the proposed method under various coefficients $c$. When the ratio of important matrices decreases from 0.8 to 0.2, the proposed

Table 8: Fine-tuning efficiency of different methods before external validation (large backbone).

| Method | RA+RD | +FixMatch | +FlexMatch | +SoftMatch | +MixedTeacher | +Adsh | +SAW | CE-SSL |
|---|---|---|---|---|---|---|---|---|
| Time/iter | 430 ms | 1200 ms | 1200 ms | 1200 ms | 960 ms | 820 ms | 1200 ms | 580 ms |
| Memory | 7.1 GB | 18.94 GB | 18.96 GB | 18.95 GB | 13.37 GB | 12.92 GB | 18.95 GB | 9.23 GB |

method allocates more ranks to the self-attention and classification blocks than to the convolution blocks. This phenomenon indicates that the deep modules exhibit higher importance than the shallow modules during model training, which aligns with the conclusions made by previous studies (Li & Liang, 2021; Zhang et al., 2023b).

### D.11 EFFECT OF THE BATCH SIZE OF UNLABELED DATA

In this section, we investigate the effect of the batch size of unlabeled data during semi-supervised learning. By default, the batch sizes of labeled and unlabeled data for CE-SSL and all compared SOTA SSL methods are set to 64 ($N_B : N_U = 1 : 1$) in our experiments, aiming at reducing the GPU memory consumption during model training. According to previous studies (Sohn et al., 2020; Chen et al., 2023a; Guo & Li, 2022), 1:2 and 1:7 are also two common ratios for implementing the SOTA semi-supervised methods. To investigate their effects on model performance, we adjust the ratio from 1:1 to 1:2 and 1:7 and present the performance of different SSL methods in Figure 15. It can be observed that the CVDs detection performance of different semi-supervised methods is insensitive to the ratio between the batch sizes of labeled and unlabeled data.

### D.12 EFFECT OF THE RATIO OF LABELED SAMPLES

Here, we compare the proposed CE-SSL and baseline models under various ratios of labeled samples in the datasets. Specifically, we adjust the ratio of the labeled samples in the dataset from 5% to 15% and present the averaged performance of different models on the four datasets in Figure 16. The experiment results demonstrate the superiority of the proposed CE-SSL compared with FixMatch and FixMatch with LoRA under various ratios of the labeled data, especially when the ratio is low. As the ratio decreases from 15% to 5%, the performance advantage of CE-SSL over other models becomes more significant. When using 15% labeled data, CE-SSL achieves improvements of 1.3% on the macro $F_{\beta=2}$ compared to FixMatch with LoRA. In contrast, CE-SSL outperforms it by 1.9% on the macro $F_{\beta=2}$ using 5% labeled data. In Figure 17, we also compare CE-SSL with other baseline models, where CE-SSL consistently outperforms them in CVDs detection under various labeled ratios.

Table 9: Detailed model performance for each CVD within the G12EC dataset using the base backbone. For each CVD, the averaged $F_{\beta=2}$ and standard deviations are shown across six seeds. The model with the best performance is denoted in **bold**.

| Methods | MixedTeacher | FixMatch | FlexMatch | SoftMatch | Adsh | SAW | CE-SSL$_{r=4}$ | CE-SSL$_{r=32}$ |
|---|---|---|---|---|---|---|---|---|
| AF | 0.508±0.078 | 0.523±0.083 | 0.529±0.072 | 0.521±0.059 | 0.443±0.133 | 0.566±0.067 | 0.659±0.075 | **0.668±0.036** |
| IAVB | 0.729±0.030 | 0.670±0.066 | 0.597±0.129 | 0.679±0.044 | 0.589±0.194 | 0.654±0.077 | **0.747±0.022** | 0.719±0.081 |
| IRBBB | 0.467±0.067 | 0.435±0.071 | 0.425±0.058 | 0.410±0.092 | 0.381±0.126 | 0.436±0.090 | **0.536±0.022** | 0.533±0.040 |
| LAD | **0.659±0.065** | 0.627±0.094 | 0.642±0.031 | 0.604±0.084 | 0.601±0.077 | 0.608±0.070 | 0.633±0.045 | 0.636±0.043 |
| LBBB | 0.581±0.236 | 0.624±0.193 | 0.557±0.255 | 0.544±0.201 | 0.588±0.126 | 0.598±0.221 | 0.706±0.121 | **0.713±0.191** |
| LQRSV | 0.208±0.069 | **0.212±0.025** | 0.167±0.068 | 0.202±0.051 | 0.160±0.065 | 0.205±0.030 | 0.184±0.062 | 0.197±0.064 |
| NSIVCB | 0.119±0.090 | 0.058±0.044 | 0.080±0.070 | 0.077±0.075 | 0.030±0.035 | 0.051±0.059 | **0.260±0.030** | 0.208±0.026 |
| NSR | 0.759±0.020 | 0.754±0.024 | 0.764±0.029 | **0.771±0.018** | 0.755±0.009 | 0.738±0.031 | 0.748±0.020 | 0.766±0.014 |
| PAC | 0.313±0.027 | 0.310±0.031 | 0.299±0.025 | 0.324±0.046 | 0.329±0.035 | 0.292±0.056 | **0.388±0.043** | 0.376±0.033 |
| LQT | 0.548±0.055 | 0.578±0.013 | **0.579±0.022** | 0.559±0.037 | 0.524±0.070 | 0.516±0.066 | 0.576±0.034 | 0.570±0.037 |
| QAb | 0.315±0.029 | **0.322±0.031** | 0.298±0.088 | 0.305±0.033 | 0.306±0.040 | 0.260±0.042 | 0.319±0.020 | 0.305±0.052 |
| RBBB | 0.702±0.073 | 0.721±0.075 | 0.749±0.119 | **0.766±0.044** | 0.753±0.028 | 0.732±0.083 | 0.737±0.031 | 0.755±0.022 |
| SA | 0.214±0.032 | 0.205±0.025 | 0.172±0.043 | 0.179±0.086 | 0.220±0.017 | 0.189±0.050 | 0.266±0.034 | **0.268±0.024** |
| SB | 0.874±0.033 | 0.879±0.036 | **0.902±0.014** | 0.891±0.021 | 0.882±0.033 | 0.865±0.044 | 0.891±0.039 | 0.891±0.020 |
| STach | 0.891±0.018 | 0.894±0.025 | 0.882±0.035 | 0.885±0.025 | 0.898±0.014 | 0.893±0.023 | **0.911±0.011** | 0.896±0.020 |
| TAb | 0.731±0.010 | 0.722±0.020 | 0.719±0.028 | **0.737±0.012** | 0.722±0.017 | 0.720±0.024 | 0.713±0.018 | 0.707±0.023 |
| TInv | 0.288±0.062 | 0.310±0.045 | 0.306±0.063 | 0.318±0.038 | 0.283±0.044 | 0.297±0.057 | **0.352±0.032** | 0.339±0.012 |
| VPB | 0.222±0.138 | 0.334±0.055 | 0.280±0.111 | 0.304±0.079 | 0.343±0.061 | 0.277±0.048 | 0.326±0.041 | **0.369±0.024** |
| Average | 0.507±0.025 | 0.510±0.016 | 0.497±0.035 | 0.504±0.021 | 0.489±0.013 | 0.494±0.024 | **0.553±0.020** | 0.551±0.017 |

Table 10: Detailed model performance for each CVD within the PTB-XL dataset using the base backbone. For each CVD, the averaged $F_{\beta=2}$ and standard deviations are shown across six seeds. The model with the best performance is denoted in **bold**.

| Methods | MixedTeacher | FixMatch | FlexMatch | SoftMatch | Adsh | SAW | CE-SSL$_{r=4}$ | CE-SSL$_{r=32}$ |
|---|---|---|---|---|---|---|---|---|
| AF | 0.882±0.009 | 0.890±0.010 | 0.846±0.042 | 0.880±0.018 | 0.864±0.048 | 0.890±0.019 | **0.908±0.007** | 0.904±0.014 |
| CRBBB | 0.667±0.145 | 0.714±0.068 | 0.697±0.084 | 0.711±0.082 | 0.646±0.121 | 0.696±0.127 | **0.814±0.042** | 0.790±0.045 |
| IAVB | 0.604±0.038 | 0.616±0.026 | 0.577±0.037 | 0.635±0.030 | 0.635±0.050 | 0.646±0.039 | **0.682±0.030** | 0.679±0.019 |
| IRBBB | 0.557±0.061 | 0.535±0.043 | 0.515±0.040 | 0.512±0.049 | 0.551±0.021 | 0.541±0.025 | **0.594±0.032** | 0.561±0.062 |
| LAD | 0.769±0.016 | 0.764±0.020 | 0.758±0.017 | 0.772±0.017 | 0.777±0.009 | 0.754±0.005 | 0.774±0.007 | **0.779±0.004** |
| LAnFB | 0.788±0.019 | **0.800±0.007** | 0.789±0.015 | 0.780±0.018 | 0.776±0.024 | 0.747±0.035 | 0.771±0.018 | 0.784±0.010 |
| LBBB | 0.844±0.046 | 0.789±0.078 | 0.797±0.043 | **0.848±0.043** | 0.820±0.074 | 0.810±0.031 | 0.804±0.037 | 0.761±0.063 |
| NSIVCB | 0.176±0.044 | 0.221±0.028 | **0.244±0.037** | 0.155±0.087 | 0.190±0.061 | 0.225±0.055 | 0.219±0.054 | 0.208±0.068 |
| NSR | 0.968±0.013 | 0.972±0.005 | 0.968±0.006 | 0.972±0.003 | **0.973±0.002** | 0.968±0.004 | 0.970±0.009 | 0.965±0.013 |
| PAC | 0.156±0.037 | 0.107±0.078 | 0.120±0.050 | 0.183±0.028 | 0.148±0.054 | 0.219±0.071 | **0.272±0.039** | 0.262±0.026 |
| PR | 0.588±0.054 | 0.737±0.028 | 0.698±0.049 | 0.638±0.102 | 0.733±0.048 | 0.715±0.059 | 0.728±0.027 | **0.747±0.026** |
| LPR | 0.527±0.035 | 0.525±0.026 | 0.450±0.063 | 0.509±0.025 | 0.458±0.081 | 0.488±0.112 | 0.583±0.042 | **0.600±0.026** |
| QAb | 0.135±0.041 | 0.121±0.054 | 0.152±0.044 | 0.154±0.055 | 0.082±0.065 | 0.128±0.039 | **0.185±0.020** | 0.169±0.037 |
| RAD | 0.428±0.068 | 0.373±0.025 | 0.415±0.057 | 0.361±0.111 | **0.482±0.052** | 0.416±0.068 | 0.408±0.056 | 0.412±0.041 |
| SA | 0.172±0.052 | 0.144±0.041 | 0.150±0.076 | 0.164±0.027 | 0.165±0.046 | 0.175±0.047 | 0.245±0.029 | **0.281±0.042** |
| SB | 0.557±0.026 | 0.549±0.022 | 0.548±0.032 | 0.526±0.042 | 0.554±0.034 | **0.568±0.029** | 0.566±0.049 | 0.558±0.032 |
| STach | 0.817±0.051 | 0.809±0.055 | 0.818±0.049 | 0.770±0.031 | 0.787±0.082 | 0.729±0.054 | 0.853±0.024 | **0.860±0.016** |
| TAb | 0.518±0.050 | 0.497±0.019 | 0.515±0.028 | 0.549±0.026 | 0.519±0.020 | 0.516±0.011 | 0.549±0.035 | **0.561±0.013** |
| TInv | 0.141±0.051 | 0.123±0.014 | 0.124±0.046 | 0.132±0.027 | 0.159±0.035 | **0.182±0.039** | 0.100±0.052 | 0.093±0.044 |
| Average | 0.542±0.014 | 0.541±0.007 | 0.536±0.007 | 0.540±0.011 | 0.543±0.015 | 0.548±0.017 | **0.580±0.006** | 0.578±0.006 |

Table 11: Detailed model performance for each CVD within the Ningbo dataset using the base backbone. For each CVD, the averaged $F_{\beta=2}$ and standard deviations are shown across six seeds. The model with the best performance is denoted in **bold**.

| Methods | MixedTeacher | FixMatch | FlexMatch | SoftMatch | Adsh | SAW | **CE-SSL**$_{r=4}$ | **CE-SSL**$_{r=32}$ |
|---|---|---|---|---|---|---|---|---|
| AFL | 0.959±0.008 | 0.962±0.007 | 0.957±0.007 | **0.966±0.002** | 0.959±0.006 | 0.963±0.005 | 0.963±0.005 | 0.965±0.005 |
| BBB | 0.266±0.160 | 0.291±0.145 | 0.295±0.111 | 0.280±0.120 | 0.287±0.105 | 0.317±0.093 | 0.391±0.040 | **0.397±0.054** |
| CLBBB | 0.713±0.143 | **0.749±0.045** | 0.707±0.135 | 0.708±0.102 | 0.725±0.051 | 0.745±0.050 | 0.719±0.065 | 0.721±0.080 |
| CRBBB | 0.760±0.027 | 0.766±0.017 | 0.722±0.118 | 0.706±0.067 | 0.761±0.020 | 0.712±0.085 | **0.777±0.029** | 0.764±0.036 |
| IAVB | 0.677±0.053 | 0.686±0.030 | 0.675±0.044 | 0.672±0.026 | 0.698±0.047 | 0.690±0.042 | **0.710±0.040** | 0.704±0.040 |
| IRBBB | 0.138±0.092 | 0.094±0.039 | 0.191±0.060 | 0.168±0.039 | 0.167±0.056 | **0.203±0.129** | 0.186±0.044 | 0.153±0.064 |
| LAD | **0.628±0.033** | 0.605±0.046 | 0.596±0.056 | 0.603±0.050 | 0.623±0.022 | 0.585±0.084 | 0.590±0.037 | 0.603±0.039 |
| LAnFB | 0.418±0.081 | 0.426±0.051 | 0.368±0.113 | **0.474±0.050** | 0.419±0.025 | 0.401±0.089 | 0.417±0.059 | 0.435±0.052 |
| LQRSV | 0.221±0.045 | 0.198±0.051 | 0.222±0.047 | 0.208±0.025 | 0.195±0.066 | 0.174±0.054 | 0.245±0.030 | **0.255±0.028** |
| NSIVCB | 0.432±0.056 | 0.388±0.087 | 0.447±0.030 | 0.413±0.057 | 0.436±0.052 | 0.397±0.146 | 0.468±0.076 | **0.476±0.061** |
| NSR | 0.857±0.009 | **0.859±0.013** | 0.853±0.009 | 0.851±0.020 | 0.842±0.017 | 0.841±0.019 | 0.828±0.013 | 0.852±0.011 |
| PAC | 0.413±0.040 | 0.401±0.038 | 0.408±0.037 | 0.428±0.043 | 0.389±0.050 | 0.346±0.061 | **0.512±0.018** | 0.501±0.030 |
| PR | 0.804±0.031 | 0.772±0.079 | 0.793±0.080 | 0.819±0.045 | 0.786±0.063 | 0.818±0.036 | 0.810±0.039 | **0.839±0.022** |
| PRWP | 0.281±0.105 | **0.289±0.059** | 0.214±0.096 | 0.253±0.119 | 0.227±0.094 | 0.232±0.064 | 0.251±0.086 | 0.260±0.072 |
| PVC | 0.613±0.055 | 0.637±0.037 | 0.640±0.040 | **0.652±0.050** | 0.596±0.043 | 0.582±0.083 | 0.638±0.048 | 0.643±0.025 |
| LQT | 0.151±0.045 | **0.197±0.049** | 0.136±0.083 | 0.188±0.063 | 0.134±0.068 | 0.175±0.045 | 0.123±0.071 | 0.161±0.030 |
| QAb | **0.385±0.041** | 0.352±0.046 | 0.350±0.035 | 0.328±0.050 | 0.303±0.084 | 0.333±0.041 | 0.362±0.042 | 0.359±0.063 |
| RAD | 0.362±0.030 | 0.319±0.120 | 0.335±0.033 | **0.389±0.051** | 0.365±0.019 | 0.360±0.064 | 0.366±0.063 | 0.351±0.059 |
| SA | 0.461±0.077 | 0.475±0.058 | 0.497±0.070 | 0.518±0.056 | 0.530±0.043 | 0.417±0.050 | **0.548±0.048** | 0.536±0.060 |
| SB | 0.971±0.004 | 0.975±0.003 | **0.975±0.002** | 0.974±0.003 | 0.970±0.005 | 0.968±0.004 | 0.974±0.002 | 0.974±0.003 |
| STach | 0.919±0.014 | 0.895±0.031 | 0.920±0.016 | 0.912±0.008 | 0.916±0.014 | 0.899±0.046 | **0.934±0.009** | 0.926±0.012 |
| TAb | 0.575±0.039 | 0.591±0.029 | 0.575±0.037 | 0.596±0.019 | 0.598±0.033 | 0.586±0.029 | **0.607±0.025** | 0.597±0.028 |
| TInv | 0.614±0.034 | 0.604±0.033 | **0.627±0.034** | 0.598±0.048 | 0.597±0.034 | 0.590±0.067 | 0.615±0.020 | 0.605±0.050 |
| Average | 0.549±0.028 | 0.545±0.020 | 0.544±0.019 | 0.552±0.020 | 0.545±0.012 | 0.536±0.016 | 0.567±0.011 | **0.569±0.014** |

Table 12: Detailed model performance for each CVD within the Chapman dataset using the base backbone. For each CVD, the averaged $F_{\beta=2}$ and standard deviations are shown across six seeds. The model with the best performance is denoted in **bold**.

| Methods | MixedTeacher | FixMatch | FlexMatch | SoftMatch | Adsh | SAW | **CE-SSL**$_{r=4}$ | **CE-SSL**$_{r=32}$ |
|---|---|---|---|---|---|---|---|---|
| AF | 0.926±0.018 | 0.944±0.008 | 0.917±0.018 | 0.925±0.031 | 0.938±0.007 | 0.935±0.015 | 0.945±0.014 | **0.948±0.010** |
| AFL | 0.482±0.026 | 0.507±0.034 | 0.463±0.060 | **0.523±0.028** | 0.487±0.015 | 0.466±0.051 | 0.473±0.012 | 0.489±0.042 |
| IAVB | 0.356±0.111 | 0.357±0.175 | 0.308±0.170 | 0.418±0.173 | 0.412±0.131 | 0.390±0.156 | **0.524±0.151** | 0.383±0.185 |
| LAD | 0.390±0.173 | 0.397±0.128 | 0.455±0.029 | 0.406±0.098 | 0.410±0.176 | **0.478±0.057** | 0.438±0.059 | 0.445±0.054 |
| LBBB | **0.455±0.122** | 0.295±0.157 | 0.265±0.081 | 0.375±0.123 | 0.420±0.092 | 0.203±0.169 | 0.328±0.127 | 0.339±0.114 |
| LQRSV | 0.081±0.069 | 0.072±0.073 | **0.144±0.026** | 0.091±0.083 | 0.105±0.065 | 0.133±0.077 | 0.105±0.022 | 0.053±0.025 |
| NSIVCB | 0.337±0.129 | 0.329±0.063 | 0.272±0.087 | 0.310±0.044 | 0.313±0.072 | **0.399±0.064** | 0.207±0.071 | 0.370±0.047 |
| NSR | 0.869±0.046 | 0.944±0.004 | 0.930±0.042 | 0.893±0.046 | 0.920±0.030 | 0.937±0.009 | 0.930±0.026 | **0.946±0.015** |
| PAC | 0.111±0.092 | 0.140±0.050 | 0.106±0.062 | 0.147±0.051 | 0.135±0.078 | 0.111±0.056 | **0.211±0.020** | 0.209±0.075 |
| QAb | **0.150±0.114** | 0.114±0.125 | 0.067±0.080 | 0.065±0.098 | 0.137±0.103 | 0.096±0.109 | 0.052±0.083 | 0.064±0.101 |
| RAD | 0.288±0.097 | **0.375±0.051** | 0.276±0.090 | 0.240±0.100 | 0.287±0.092 | 0.285±0.114 | 0.342±0.060 | 0.305±0.051 |
| RBBB | 0.786±0.066 | 0.814±0.069 | 0.729±0.091 | 0.774±0.073 | 0.787±0.064 | 0.790±0.076 | 0.858±0.016 | **0.879±0.033** |
| SB | 0.961±0.028 | 0.974±0.016 | 0.970±0.017 | 0.970±0.014 | 0.963±0.026 | **0.980±0.009** | 0.969±0.012 | 0.978±0.007 |
| STach | 0.943±0.010 | 0.941±0.011 | 0.928±0.022 | 0.939±0.016 | 0.943±0.005 | 0.928±0.041 | 0.950±0.016 | **0.954±0.007** |
| TAb | 0.607±0.036 | 0.646±0.032 | 0.643±0.029 | 0.647±0.026 | 0.656±0.020 | 0.620±0.036 | 0.651±0.018 | **0.667±0.016** |
| VPB | 0.422±0.098 | 0.431±0.147 | 0.443±0.138 | 0.450±0.221 | 0.356±0.192 | 0.407±0.133 | **0.494±0.053** | 0.447±0.039 |
| Average | 0.510±0.024 | 0.518±0.025 | 0.495±0.019 | 0.511±0.021 | 0.517±0.020 | 0.510±0.020 | 0.530±0.012 | **0.530±0.008** |

Table 13: Performance comparisons between CE-SSL and semi-supervised baselines on the base backbone. The average performance on all CVDs within each dataset is shown across six seeds. The standard deviation is also reported for the evaluation metrics.

| Methods | Params ↓ | Mem ↓ | Time/iter ↓ | Ranking Loss ↓ | Coverage ↓ | Macro AUC ↑ | MAP ↑ | Macro $G_{\beta=2}$ ↑ | Macro $F_{\beta=2}$ ↑ |
|---|---|---|---|---|---|---|---|---|---|
| | | | | **G12EC Dataset** | | | | | |
| MixedTeacher | 9.505 M | 3.941 GB | 147 ms | 0.107±0.009 | 4.224±0.236 | 0.835±0.010 | 0.464±0.003 | 0.275±0.016 | 0.507±0.025 |
| FixMatch | 9.505 M | 5.784 GB | 187 ms | 0.107±0.006 | 4.292±0.163 | 0.829±0.004 | 0.468±0.009 | 0.280±0.010 | 0.510±0.016 |
| FlexMatch | 9.505 M | 5.784 GB | 187 ms | 0.113±0.005 | 4.365±0.133 | 0.829±0.009 | 0.450±0.022 | 0.274±0.019 | 0.497±0.035 |
| SoftMatch | 9.505 M | 5.784 GB | 187 ms | 0.110±0.004 | 4.313±0.128 | 0.834±0.004 | 0.457±0.010 | 0.276±0.017 | 0.504±0.021 |
| Adsh | 9.505 M | 3.887 GB | 207 ms | 0.111±0.003 | 4.387±0.129 | 0.827±0.005 | 0.458±0.007 | 0.268±0.009 | 0.489±0.013 |
| SAW | 9.505 M | 5.784 GB | 188 ms | 0.112±0.003 | 4.369±0.105 | 0.827±0.005 | 0.459±0.017 | 0.269±0.018 | 0.494±0.024 |
| **CE-SSL**$_{r=16}$ | **0.510 M** | **2.747 GB** | **98 ms** | **0.092±0.002** | **3.867±0.088** | **0.855±0.005** | **0.476±0.006** | **0.307±0.016** | **0.551±0.017** |
| **CE-SSL**$_{r=4}$ | **0.183 M** | **2.743 GB** | **98 ms** | **0.089±0.003** | **3.804±0.095** | **0.853±0.004** | **0.467±0.006** | **0.304±0.013** | **0.553±0.020** |
| | | | | **PTB-XL Dataset** | | | | | |
| MixedTeacher | 9.505 M | 3.941 GB | 164 ms | 0.037±0.003 | 2.841±0.095 | 0.884±0.008 | 0.509±0.008 | 0.316±0.007 | 0.542±0.014 |
| FixMatch | 9.505 M | 5.784 GB | 208 ms | 0.038±0.001 | 2.905±0.061 | 0.882±0.004 | 0.510±0.006 | 0.322±0.007 | 0.541±0.007 |
| FlexMatch | 9.505 M | 5.784 GB | 209 ms | 0.039±0.001 | 2.937±0.048 | 0.887±0.005 | 0.505±0.005 | 0.316±0.008 | 0.536±0.007 |
| SoftMatch | 9.505 M | 5.784 GB | 209 ms | 0.039±0.003 | 2.919±0.097 | 0.885±0.006 | 0.508±0.007 | 0.317±0.009 | 0.540±0.011 |
| Adsh | 9.505 M | 3.887 GB | 316 ms | 0.038±0.002 | 2.879±0.054 | 0.886±0.004 | 0.511±0.005 | 0.322±0.008 | 0.543±0.015 |
| SAW | 9.505 M | 5.784 GB | 208 ms | 0.037±0.003 | 2.855±0.093 | 0.889±0.005 | 0.520±0.007 | 0.323±0.019 | 0.548±0.017 |
| **CE-SSL**$_{r=16}$ | **0.582 M** | **2.748 GB** | **110 ms** | **0.031±0.000** | **2.641±0.020** | **0.901±0.003** | **0.530±0.005** | **0.346±0.006** | **0.578±0.006** |
| **CE-SSL**$_{r=4}$ | **0.159 M** | **2.744 GB** | **109 ms** | **0.030±0.001** | **2.626±0.026** | **0.899±0.004** | **0.526±0.005** | **0.346±0.005** | **0.580±0.006** |
| | | | | **Ningbo Dataset** | | | | | |
| MixedTeacher | 9.506 M | 3.941 GB | 173 ms | 0.035±0.002 | 2.982±0.077 | 0.925±0.006 | 0.496±0.020 | 0.324±0.018 | 0.549±0.028 |
| FixMatch | 9.506 M | 5.784 GB | 217 ms | 0.035±0.003 | 3.025±0.121 | 0.922±0.009 | 0.493±0.023 | 0.321±0.014 | 0.545±0.020 |
| FlexMatch | 9.506 M | 5.784 GB | 217 ms | 0.037±0.002 | 3.078±0.090 | 0.921±0.007 | 0.489±0.024 | 0.318±0.012 | 0.544±0.019 |
| SoftMatch | 9.506 M | 5.784 GB | 217 ms | 0.035±0.001 | 3.018±0.049 | 0.923±0.005 | 0.496±0.024 | 0.321±0.014 | 0.552±0.020 |
| Adsh | 9.506 M | 3.887 GB | 423 ms | 0.035±0.002 | 3.007±0.090 | 0.921±0.004 | 0.492±0.023 | 0.318±0.010 | 0.545±0.012 |
| SAW | 9.506 M | 5.784 GB | 215 ms | 0.037±0.001 | 3.064±0.036 | 0.924±0.004 | 0.492±0.024 | 0.314±0.010 | 0.536±0.016 |
| **CE-SSL**$_{r=16}$ | **0.550 M** | **2.748 GB** | **115 ms** | **0.030±0.001** | **2.805±0.063** | **0.928±0.002** | **0.505±0.019** | **0.334±0.011** | **0.569±0.014** |
| **CE-SSL**$_{r=4}$ | **0.168 M** | **2.744 GB** | **114 ms** | **0.030±0.001** | **2.776±0.028** | **0.929±0.001** | **0.500±0.017** | **0.327±0.010** | **0.567±0.011** |
| | | | | **Chapman Dataset** | | | | | |
| MixedTeacher | 9.504 M | 3.941 GB | 148 ms | 0.047±0.002 | 2.615±0.068 | 0.889±0.012 | 0.519±0.018 | 0.327±0.019 | 0.510±0.024 |
| FixMatch | 9.504 M | 5.784 GB | 186 ms | 0.046±0.004 | 2.626±0.096 | 0.897±0.006 | 0.520±0.009 | 0.339±0.012 | 0.518±0.025 |
| FlexMatch | 9.504 M | 5.784 GB | 185 ms | 0.047±0.004 | 2.659±0.103 | 0.895±0.006 | 0.518±0.008 | 0.325±0.010 | 0.495±0.019 |
| SoftMatch | 9.504 M | 5.784 GB | 187 ms | 0.047±0.004 | 2.649±0.079 | 0.898±0.006 | 0.525±0.012 | 0.335±0.011 | 0.511±0.021 |
| Adsh | 9.504 M | 3.887 GB | 207 ms | 0.046±0.004 | 2.621±0.117 | 0.896±0.005 | 0.528±0.008 | 0.335±0.013 | 0.517±0.020 |
| SAW | 9.504 M | 5.784 GB | 185 ms | 0.049±0.003 | 2.699±0.072 | 0.897±0.007 | 0.524±0.009 | 0.333±0.012 | 0.510±0.020 |
| **CE-SSL**$_{r=16}$ | **0.581 M** | **2.748 GB** | **97 ms** | **0.040±0.002** | **2.483±0.055** | **0.896±0.006** | **0.536±0.004** | **0.355±0.005** | **0.530±0.008** |
| **CE-SSL**$_{r=4}$ | **0.180 M** | **2.743 GB** | **97 ms** | **0.038±0.002** | **2.418±0.049** | **0.898±0.005** | **0.526±0.006** | **0.352±0.009** | **0.530±0.012** |

Table 14: Performance comparisons between CE-SSL and semi-supervised baselines on the medium backbone. The average performance on all CVDs within each dataset is shown across six seeds. The standard deviation is also reported for the evaluation metrics.

| Methods | Params ↓ | Mem ↓ | Time/iter ↓ | Ranking Loss ↓ | Coverage ↓ | Macro AUC ↑ | MAP ↑ | Macro $G_{\beta=2}$ ↑ | Macro $F_{\beta=2}$ ↑ |
|---|---|---|---|---|---|---|---|---|---|
| **G12EC Dataset** | | | | | | | | | |
| MixedTeacher | 50.493 M | 9.461 GB | 396 ms | 0.096±0.003 | 4.016±0.060 | 0.846±0.008 | 0.499±0.009 | 0.303±0.014 | 0.537±0.018 |
| FixMatch | 50.493 M | 13.589 GB | 499 ms | 0.096±0.006 | 4.027±0.109 | 0.850±0.009 | 0.499±0.014 | 0.299±0.016 | 0.529±0.016 |
| FlexMatch | 50.493 M | 13.589 GB | 498 ms | 0.104±0.003 | 4.216±0.070 | 0.848±0.008 | 0.499±0.009 | 0.294±0.019 | 0.521±0.020 |
| SoftMatch | 50.493 M | 13.589 GB | 498 ms | 0.097±0.003 | 4.096±0.093 | 0.853±0.007 | 0.505±0.008 | 0.309±0.010 | 0.536±0.013 |
| Adsh | 50.493 M | 9.251 GB | 524 ms | 0.098±0.003 | 4.107±0.090 | 0.845±0.008 | 0.493±0.011 | 0.298±0.014 | 0.531±0.020 |
| SAW | 50.493 M | 13.589 GB | 499 ms | 0.100±0.003 | 4.129±0.083 | 0.847±0.004 | 0.490±0.007 | 0.293±0.014 | 0.526±0.012 |
| **CE-SSL**$_{r=16}$ | **1.568 M** | **6.158 GB** | **243 ms** | **0.086±0.004** | **3.740±0.134** | **0.862±0.006** | **0.507±0.007** | **0.317±0.022** | **0.561±0.024** |
| **CE-SSL**$_{r=4}$ | **0.458 M** | **6.146 GB** | **241 ms** | **0.085±0.002** | **3.741±0.068** | **0.862±0.007** | **0.503±0.006** | **0.316±0.013** | **0.560±0.015** |
| **PTB-XL Dataset** | | | | | | | | | |
| MixedTeacher | 50.494 M | 9.459 GB | 440 ms | 0.032±0.001 | 2.706±0.049 | 0.898±0.004 | 0.539±0.005 | 0.340±0.013 | 0.559±0.012 |
| FixMatch | 50.494 M | 13.589 GB | 553 ms | 0.034±0.002 | 2.767±0.053 | 0.898±0.003 | 0.536±0.006 | 0.340±0.006 | 0.556±0.010 |
| FlexMatch | 50.494 M | 13.589 GB | 553 ms | 0.034±0.001 | 2.747±0.047 | 0.901±0.004 | 0.529±0.004 | 0.348±0.013 | 0.559±0.008 |
| SoftMatch | 50.494 M | 13.589 GB | 553 ms | 0.034±0.001 | 2.790±0.026 | 0.898±0.003 | 0.533±0.004 | 0.341±0.007 | 0.553±0.009 |
| Adsh | 50.494 M | 9.251 GB | 796 ms | 0.033±0.002 | 2.757±0.079 | 0.901±0.003 | 0.537±0.007 | 0.339±0.008 | 0.557±0.014 |
| SAW | 50.494 M | 13.589 GB | 554 ms | 0.034±0.001 | 2.778±0.050 | 0.899±0.001 | 0.531±0.010 | 0.344±0.011 | 0.562±0.009 |
| **CE-SSL**$_{r=16}$ | **1.485 M** | **6.161 GB** | **271 ms** | **0.027±0.001** | **2.539±0.033** | **0.913±0.003** | **0.550±0.004** | **0.369±0.005** | **0.588±0.003** |
| **CE-SSL**$_{r=4}$ | **0.505 M** | **6.150 GB** | **270 ms** | **0.027±0.001** | **2.529±0.019** | **0.914±0.003** | **0.547±0.003** | **0.372±0.006** | **0.599±0.010** |
| **Ningbo Dataset** | | | | | | | | | |
| MixedTeacher | 50.496 M | 9.459 GB | 457 ms | 0.031±0.002 | 2.856±0.078 | 0.926±0.009 | 0.525±0.023 | 0.342±0.016 | 0.571±0.023 |
| FixMatch | 50.496 M | 13.589 GB | 572 ms | 0.031±0.002 | 2.869±0.081 | 0.931±0.003 | 0.531±0.021 | 0.349±0.014 | 0.575±0.015 |
| FlexMatch | 50.496 M | 13.589 GB | 573 ms | 0.031±0.002 | 2.853±0.081 | 0.930±0.002 | 0.524±0.012 | 0.347±0.013 | 0.575±0.018 |
| SoftMatch | 50.496 M | 13.589 GB | 574 ms | 0.031±0.002 | 2.877±0.094 | 0.927±0.002 | 0.525±0.019 | 0.344±0.014 | 0.573±0.017 |
| Adsh | 50.496 M | 9.251 GB | 1061 ms | 0.031±0.002 | 2.868±0.061 | 0.927±0.004 | 0.523±0.013 | 0.342±0.012 | 0.571±0.017 |
| SAW | 50.496 M | 13.589 GB | 572 ms | 0.032±0.002 | 2.911±0.105 | 0.930±0.003 | 0.525±0.017 | 0.342±0.013 | 0.578±0.016 |
| **CE-SSL**$_{r=16}$ | **1.705 M** | **6.172 GB** | **282 ms** | **0.027±0.001** | **2.701±0.051** | **0.933±0.003** | **0.531±0.018** | **0.356±0.013** | **0.588±0.021** |
| **CE-SSL**$_{r=4}$ | **0.507 M** | **6.160 GB** | **282 ms** | **0.026±0.001** | **2.661±0.058** | **0.934±0.004** | **0.525±0.018** | **0.352±0.013** | **0.587±0.020** |
| **Chapman Dataset** | | | | | | | | | |
| MixedTeacher | 50.492 M | 9.461 GB | 394 ms | 0.037±0.002 | 2.420±0.071 | 0.909±0.010 | 0.539±0.007 | 0.348±0.016 | 0.513±0.026 |
| FixMatch | 50.492 M | 13.589 GB | 495 ms | 0.038±0.004 | 2.439±0.092 | 0.905±0.010 | 0.538±0.011 | 0.357±0.009 | 0.522±0.020 |
| FlexMatch | 50.492 M | 13.589 GB | 495 ms | 0.041±0.003 | 2.519±0.077 | 0.901±0.004 | 0.531±0.011 | 0.345±0.016 | 0.512±0.030 |
| SoftMatch | 50.492 M | 13.589 GB | 495 ms | 0.043±0.004 | 2.546±0.101 | 0.902±0.009 | 0.535±0.008 | 0.355±0.015 | 0.526±0.026 |
| Adsh | 50.492 M | 9.251 GB | 527 ms | 0.039±0.004 | 2.440±0.073 | 0.909±0.006 | 0.546±0.007 | 0.356±0.007 | 0.530±0.013 |
| SAW | 50.492 M | 13.589 GB | 493 ms | 0.043±0.003 | 2.549±0.073 | 0.901±0.006 | 0.531±0.008 | 0.357±0.013 | 0.532±0.027 |
| **CE-SSL**$_{r=16}$ | **1.601 M** | **6.159 GB** | **241 ms** | **0.035±0.002** | **2.362±0.049** | **0.909±0.007** | **0.553±0.013** | **0.367±0.008** | **0.540±0.019** |
| **CE-SSL**$_{r=4}$ | **0.402 M** | **6.145 GB** | **240 ms** | **0.034±0.001** | **2.334±0.033** | **0.908±0.008** | **0.538±0.014** | **0.361±0.009** | **0.531±0.019** |

Table 15: Performance comparisons between CE-SSL and semi-supervised baselines on the large backbone. The average performance on all CVDs within each dataset is shown across six seeds. The standard deviation is also reported for the evaluation metrics.

| Methods | Params ↓ | Mem ↓ | Time/iter ↓ | Ranking Loss ↓ | Coverage ↓ | Macro AUC ↑ | MAP ↑ | Macro $G_{\beta=2}$ ↑ | Macro $F_{\beta=2}$ ↑ |
|---|---|---|---|---|---|---|---|---|---|
| | | | | **G12EC Dataset** | | | | | |
| MixedTeacher | 113.489 M | 14.257 GB | 728 ms | 0.111±0.022 | 4.365±0.447 | 0.835±0.026 | 0.489±0.018 | 0.285±0.017 | 0.517±0.029 |
| FixMatch | 113.489 M | 20.061 GB | 966 ms | 0.100±0.005 | 4.147±0.113 | 0.843±0.007 | 0.493±0.008 | 0.293±0.011 | 0.518±0.015 |
| FlexMatch | 113.489 M | 20.061 GB | 966 ms | 0.099±0.006 | 4.088±0.149 | 0.847±0.003 | 0.489±0.005 | 0.299±0.011 | 0.534±0.015 |
| SoftMatch | 113.489 M | 20.061 GB | 943 ms | 0.100±0.007 | 4.138±0.194 | 0.847±0.004 | 0.498±0.005 | 0.297±0.004 | 0.532±0.013 |
| Adsh | 113.489 M | 13.815 GB | 951 ms | 0.103±0.003 | 4.240±0.073 | 0.843±0.008 | 0.496±0.007 | 0.294±0.010 | 0.521±0.023 |
| SAW | 113.489 M | 20.061 GB | 939 ms | 0.102±0.002 | 4.189±0.070 | 0.842±0.003 | 0.490±0.005 | 0.300±0.007 | 0.534±0.019 |
| **CE-SSL**$_{r=16}$ | **2.658 M** | **9.217 GB** | **472 ms** | **0.085±0.005** | **3.778±0.140** | **0.857±0.004** | **0.509±0.007** | **0.322±0.009** | **0.565±0.010** |
| **CE-SSL**$_{r=4}$ | **0.761 M** | **9.206 GB** | **453 ms** | **0.084±0.003** | **3.742±0.117** | **0.859±0.004** | **0.506±0.007** | **0.323±0.004** | **0.561±0.002** |
| | | | | **PTB-XL Dataset** | | | | | |
| MixedTeacher | 113.490 M | 14.257 GB | 809 ms | 0.035±0.001 | 2.831±0.032 | 0.895±0.006 | 0.522±0.004 | 0.334±0.006 | 0.556±0.008 |
| FixMatch | 113.490 M | 20.061 GB | 1072 ms | 0.035±0.003 | 2.805±0.102 | 0.894±0.004 | 0.521±0.006 | 0.342±0.007 | 0.560±0.012 |
| FlexMatch | 113.490 M | 20.061 GB | 1071 ms | 0.041±0.004 | 3.016±0.124 | 0.893±0.004 | 0.519±0.006 | 0.342±0.010 | 0.557±0.010 |
| SoftMatch | 113.490 M | 20.061 GB | 1047 ms | 0.038±0.003 | 2.886±0.094 | 0.893±0.004 | 0.523±0.007 | 0.334±0.007 | 0.542±0.011 |
| Adsh | 113.490 M | 13.815 GB | 1432 ms | 0.036±0.003 | 2.848±0.114 | 0.892±0.002 | 0.527±0.005 | 0.343±0.009 | 0.563±0.010 |
| SAW | 113.490 M | 20.061 GB | 1039 ms | 0.035±0.002 | 2.825±0.068 | 0.899±0.006 | 0.532±0.006 | 0.347±0.007 | 0.560±0.010 |
| **CE-SSL**$_{r=16}$ | **2.235 M** | **9.220 GB** | **508 ms** | **0.030±0.002** | **2.618±0.061** | **0.909±0.004** | **0.537±0.004** | **0.358±0.005** | **0.587±0.008** |
| **CE-SSL**$_{r=4}$ | **0.712 M** | **9.211 GB** | **506 ms** | **0.029±0.001** | **2.602±0.028** | **0.908±0.003** | **0.535±0.004** | **0.356±0.006** | **0.582±0.008** |
| | | | | **Ningbo Dataset** | | | | | |
| MixedTeacher | 113.493 M | 14.257 GB | 840 ms | 0.033±0.002 | 2.934±0.079 | 0.929±0.003 | 0.518±0.021 | 0.341±0.018 | 0.572±0.026 |
| FixMatch | 113.493 M | 20.061 GB | 1111 ms | 0.033±0.002 | 2.962±0.070 | 0.926±0.004 | 0.513±0.024 | 0.337±0.018 | 0.563±0.027 |
| FlexMatch | 113.493 M | 20.061 GB | 1083 ms | 0.035±0.002 | 3.038±0.076 | 0.926±0.004 | 0.511±0.023 | 0.332±0.012 | 0.562±0.017 |
| SoftMatch | 113.493 M | 20.061 GB | 1080 ms | 0.034±0.002 | 2.999±0.081 | 0.924±0.005 | 0.513±0.023 | 0.333±0.014 | 0.561±0.022 |
| Adsh | 113.493 M | 13.815 GB | 1896 ms | 0.035±0.003 | 3.003±0.111 | 0.927±0.003 | 0.511±0.024 | 0.342±0.011 | 0.570±0.014 |
| SAW | 113.493 M | 20.061 GB | 1077 ms | 0.035±0.002 | 3.009±0.083 | 0.925±0.005 | 0.509±0.025 | 0.337±0.014 | 0.565±0.025 |
| **CE-SSL**$_{r=16}$ | **2.234 M** | **9.235 GB** | **530 ms** | **0.029±0.001** | **2.779±0.027** | **0.931±0.002** | **0.523±0.027** | **0.344±0.010** | **0.578±0.013** |
| **CE-SSL**$_{r=4}$ | **0.740 M** | **9.223 GB** | **528 ms** | **0.028±0.001** | **2.741±0.039** | **0.930±0.002** | **0.513±0.018** | **0.346±0.007** | **0.584±0.009** |
| | | | | **Chapman Dataset** | | | | | |
| MixedTeacher | 113.487 M | 14.257 GB | 724 ms | 0.040±0.003 | 2.493±0.077 | 0.904±0.009 | 0.544±0.011 | 0.341±0.007 | 0.516±0.022 |
| FixMatch | 113.487 M | 20.061 GB | 960 ms | 0.042±0.002 | 2.545±0.048 | 0.900±0.008 | 0.534±0.014 | 0.350±0.013 | 0.518±0.026 |
| FlexMatch | 113.487 M | 20.061 GB | 937 ms | 0.045±0.003 | 2.620±0.079 | 0.895±0.014 | 0.523±0.024 | 0.333±0.018 | 0.495±0.025 |
| SoftMatch | 113.487 M | 20.061 GB | 931 ms | 0.043±0.003 | 2.555±0.068 | 0.894±0.008 | 0.536±0.011 | 0.345±0.010 | 0.518±0.021 |
| Adsh | 113.487 M | 13.815 GB | 957 ms | 0.046±0.003 | 2.649±0.075 | 0.893±0.009 | 0.533±0.010 | 0.341±0.010 | 0.511±0.014 |
| SAW | 113.487 M | 20.061 GB | 933 ms | 0.044±0.004 | 2.599±0.093 | 0.900±0.007 | 0.533±0.011 | 0.344±0.015 | 0.518±0.029 |
| **CE-SSL**$_{r=16}$ | **2.205 M** | **9.223 GB** | **451 ms** | **0.037±0.001** | **2.417±0.035** | **0.904±0.004** | **0.556±0.006** | **0.371±0.010** | **0.552±0.018** |
| **CE-SSL**$_{r=4}$ | **0.716 M** | **9.206 GB** | **451 ms** | **0.036±0.001** | **2.404±0.041** | **0.902±0.006** | **0.550±0.008** | **0.365±0.006** | **0.548±0.010** |

Table 16: Performance comparisons between CE-SSL and parameter-efficient semi-supervised baselines on the base backbone. The average performance on all CVDs within each dataset is shown across six seeds. The standard deviation is also reported for the evaluation metrics.

| Methods | Params ↓ | Time/iter ↓ | Ranking Loss ↓ | Coverage ↓ | Macro AUC ↑ | MAP ↑ | Macro $G_{\beta=2}$ ↑ | Macro $F_{\beta=2}$ ↑ |
|---|---|---|---|---|---|---|---|---|
| **G12EC Dataset** | | | | | | | | |
| FixMatch | 9.505 M | 187 ms | 0.107±0.006 | 4.292±0.163 | 0.829±0.004 | 0.468±0.009 | 0.280±0.010 | 0.510±0.016 |
| + LoRA$_{r=16}$ | 0.795 M | 204 ms | 0.098±0.003 | 4.003±0.114 | 0.841±0.009 | 0.460±0.017 | 0.279±0.022 | 0.518±0.031 |
| + DyLoRA$_{r=16}$ | 0.795 M | 204 ms | 0.098±0.004 | 3.981±0.084 | 0.841±0.009 | 0.456±0.010 | 0.282±0.017 | 0.515±0.022 |
| + AdaLoRA$_{r=16}$ | 0.796 M | 237 ms | 0.096±0.003 | 3.986±0.110 | 0.844±0.007 | 0.461±0.008 | 0.284±0.015 | 0.520±0.015 |
| + IncreLoRA$_{r=16}$ | 0.824 M | 430 ms | 0.088±0.003 | 3.770±0.056 | 0.850±0.005 | 0.460±0.008 | 0.289±0.011 | 0.532±0.013 |
| + LoRA$_{r=4}$ | 0.222 M | 202 ms | 0.092±0.004 | 3.859±0.124 | 0.850±0.007 | 0.467±0.004 | 0.289±0.014 | 0.529±0.024 |
| + DyLoRA$_{r=4}$ | 0.222 M | 203 ms | 0.095±0.002 | 3.915±0.106 | 0.843±0.005 | 0.460±0.009 | 0.278±0.017 | 0.518±0.016 |
| + AdaLoRA$_{r=4}$ | 0.222 M | 236 ms | 0.093±0.003 | 3.871±0.079 | 0.849±0.005 | 0.463±0.008 | 0.288±0.011 | 0.528±0.016 |
| + IncreLoRA$_{r=4}$ | 0.246 M | 292 ms | 0.090±0.001 | 3.817±0.043 | 0.847±0.005 | 0.454±0.006 | 0.281±0.015 | 0.521±0.022 |
| **CE-SSL$_{r=16}$** | **0.510 M** | **98 ms** | **0.092±0.002** | **3.867±0.088** | **0.855±0.005** | **0.476±0.006** | **0.307±0.016** | **0.551±0.017** |
| **CE-SSL$_{r=4}$** | **0.183 M** | **98 ms** | **0.089±0.003** | **3.804±0.095** | **0.853±0.004** | **0.467±0.006** | **0.304±0.013** | **0.553±0.020** |
| **PTB-XL Dataset** | | | | | | | | |
| FixMatch | 9.505 M | 208 ms | 0.038±0.001 | 2.905±0.061 | 0.882±0.004 | 0.510±0.006 | 0.322±0.007 | 0.541±0.007 |
| + LoRA$_{r=16}$ | 0.795 M | 225 ms | 0.033±0.001 | 2.733±0.034 | 0.892±0.002 | 0.520±0.006 | 0.331±0.005 | 0.557±0.004 |
| + DyLoRA$_{r=16}$ | 0.795 M | 226 ms | 0.033±0.001 | 2.716±0.057 | 0.894±0.003 | 0.524±0.003 | 0.321±0.010 | 0.553±0.010 |
| + AdaLoRA$_{r=16}$ | 0.796 M | 262 ms | 0.032±0.001 | 2.687±0.025 | 0.896±0.003 | 0.508±0.009 | 0.326±0.012 | 0.552±0.015 |
| + IncreLoRA$_{r=16}$ | 0.825 M | 469 ms | 0.031±0.001 | 2.620±0.020 | 0.903±0.002 | 0.520±0.004 | 0.342±0.008 | 0.573±0.008 |
| + LoRA$_{r=4}$ | 0.222 M | 225 ms | 0.032±0.001 | 2.673±0.035 | 0.898±0.004 | 0.522±0.006 | 0.329±0.012 | 0.554±0.009 |
| + DyLoRA$_{r=4}$ | 0.222 M | 225 ms | 0.032±0.001 | 2.668±0.036 | 0.896±0.003 | 0.521±0.005 | 0.328±0.008 | 0.554±0.008 |
| + AdaLoRA$_{r=4}$ | 0.223 M | 263 ms | 0.032±0.000 | 2.696±0.010 | 0.896±0.002 | 0.510±0.003 | 0.323±0.008 | 0.550±0.012 |
| + IncreLoRA$_{r=4}$ | 0.246 M | 322 ms | 0.031±0.001 | 2.630±0.034 | 0.899±0.004 | 0.518±0.006 | 0.338±0.009 | 0.570±0.010 |
| **CE-SSL$_{r=16}$** | **0.582 M** | **110 ms** | **0.031±0.000** | **2.641±0.020** | **0.901±0.003** | **0.530±0.005** | **0.346±0.006** | **0.578±0.006** |
| **CE-SSL$_{r=4}$** | **0.159 M** | **109 ms** | **0.030±0.001** | **2.626±0.026** | **0.899±0.004** | **0.526±0.005** | **0.346±0.005** | **0.580±0.006** |
| **Ningbo Dataset** | | | | | | | | |
| FixMatch | 9.506 M | 217 ms | 0.035±0.003 | 3.025±0.121 | 0.922±0.009 | 0.493±0.023 | 0.321±0.014 | 0.545±0.020 |
| + LoRA$_{r=16}$ | 0.796 M | 234 ms | 0.032±0.001 | 2.864±0.045 | 0.926±0.002 | 0.497±0.018 | 0.326±0.007 | 0.561±0.008 |
| + DyLoRA$_{r=16}$ | 0.796 M | 235 ms | 0.032±0.002 | 2.874±0.083 | 0.927±0.003 | 0.498±0.017 | 0.321±0.011 | 0.553±0.016 |
| + AdaLoRA$_{r=16}$ | 0.797 M | 272 ms | 0.032±0.002 | 2.851±0.054 | 0.925±0.003 | 0.487±0.021 | 0.317±0.017 | 0.546±0.028 |
| + IncreLoRA$_{r=16}$ | 0.827 M | 491 ms | 0.030±0.001 | 2.772±0.045 | 0.929±0.003 | 0.499±0.023 | 0.328±0.011 | 0.564±0.016 |
| + LoRA$_{r=4}$ | 0.223 M | 234 ms | 0.031±0.001 | 2.842±0.046 | 0.926±0.003 | 0.489±0.026 | 0.319±0.013 | 0.551±0.019 |
| + DyLoRA$_{r=4}$ | 0.223 M | 234 ms | 0.031±0.001 | 2.841±0.034 | 0.924±0.003 | 0.489±0.020 | 0.323±0.016 | 0.556±0.026 |
| + AdaLoRA$_{r=4}$ | 0.224 M | 272 ms | 0.033±0.001 | 2.896±0.037 | 0.923±0.004 | 0.480±0.018 | 0.312±0.006 | 0.543±0.017 |
| + IncreLoRA$_{r=4}$ | 0.247 M | 332 ms | 0.030±0.001 | 2.794±0.046 | 0.927±0.002 | 0.490±0.025 | 0.314±0.014 | 0.551±0.022 |
| **CE-SSL$_{r=16}$** | **0.550 M** | **115 ms** | **0.030±0.001** | **2.805±0.063** | **0.928±0.002** | **0.505±0.019** | **0.334±0.011** | **0.569±0.014** |
| **CE-SSL$_{r=4}$** | **0.168 M** | **114 ms** | **0.030±0.001** | **2.776±0.028** | **0.929±0.001** | **0.500±0.017** | **0.327±0.010** | **0.567±0.011** |
| **Chapman Dataset** | | | | | | | | |
| FixMatch | 9.504 M | 186 ms | 0.046±0.004 | 2.626±0.096 | 0.897±0.006 | 0.520±0.009 | 0.339±0.012 | 0.518±0.025 |
| + LoRA$_{r=16}$ | 0.795 M | 201 ms | 0.041±0.002 | 2.493±0.058 | 0.899±0.005 | 0.521±0.014 | 0.338±0.011 | 0.515±0.015 |
| + DyLoRA$_{r=16}$ | 0.795 M | 202 ms | 0.042±0.004 | 2.512±0.091 | 0.899±0.003 | 0.524±0.011 | 0.336±0.009 | 0.511±0.015 |
| + AdaLoRA$_{r=16}$ | 0.795 M | 234 ms | 0.042±0.001 | 2.520±0.039 | 0.883±0.011 | 0.503±0.020 | 0.338±0.018 | 0.498±0.019 |
| + IncreLoRA$_{r=16}$ | 0.822 M | 426 ms | 0.041±0.003 | 2.484±0.072 | 0.884±0.017 | 0.495±0.022 | 0.334±0.019 | 0.504±0.029 |
| + LoRA$_{r=4}$ | 0.222 M | 201 ms | 0.038±0.001 | 2.427±0.039 | 0.902±0.006 | 0.522±0.010 | 0.338±0.011 | 0.523±0.012 |
| + DyLoRA$_{r=4}$ | 0.222 M | 200 ms | 0.039±0.002 | 2.445±0.057 | 0.898±0.010 | 0.518±0.013 | 0.331±0.008 | 0.506±0.016 |
| + AdaLoRA$_{r=4}$ | 0.222 M | 233 ms | 0.039±0.002 | 2.457±0.044 | 0.891±0.010 | 0.512±0.014 | 0.345±0.014 | 0.521±0.018 |
| + IncreLoRA$_{r=4}$ | 0.246 M | 288 ms | 0.039±0.001 | 2.446±0.039 | 0.888±0.008 | 0.502±0.015 | 0.339±0.014 | 0.505±0.022 |
| **CE-SSL$_{r=16}$** | **0.581 M** | **97 ms** | **0.040±0.002** | **2.483±0.055** | **0.896±0.006** | **0.536±0.004** | **0.355±0.005** | **0.530±0.008** |
| **CE-SSL$_{r=4}$** | **0.180 M** | **97 ms** | **0.038±0.002** | **2.418±0.049** | **0.898±0.005** | **0.526±0.006** | **0.352±0.009** | **0.530±0.012** |

Table 17: Performance comparisons between CE-SSL and parameter-efficient semi-supervised baselines on the medium backbone. The average performance on all CVDs within each dataset is shown across six seeds. The standard deviation is also reported for the evaluation metrics.

| Methods | Params ↓ | Time/iter ↓ | Ranking Loss ↓ | Coverage ↓ | Macro AUC ↑ | MAP ↑ | Macro $G_{\beta=2}$ ↑ | Macro $F_{\beta=2}$ ↑ |
|---|---|---|---|---|---|---|---|---|
| **G12EC Dataset** | | | | | | | | |
| FixMatch | 50.493 M | 499 ms | 0.096±0.006 | 4.027±0.109 | 0.850±0.009 | 0.499±0.014 | 0.299±0.016 | 0.529±0.016 |
| + LoRA$_{r=16}$ | 2.135 M | 545 ms | 0.093±0.003 | 3.943±0.094 | 0.854±0.005 | 0.494±0.006 | 0.300±0.016 | 0.534±0.024 |
| + DyLoRA$_{r=16}$ | 2.135 M | 542 ms | 0.092±0.003 | 3.913±0.117 | 0.851±0.007 | 0.494±0.012 | 0.296±0.015 | 0.533±0.021 |
| + AdaLoRA$_{r=16}$ | 2.136 M | 585 ms | 0.096±0.004 | 4.013±0.095 | 0.847±0.008 | 0.478±0.014 | 0.296±0.009 | 0.533±0.008 |
| + IncreLoRA$_{r=16}$ | 2.164 M | 977 ms | 0.085±0.003 | 3.683±0.100 | 0.859±0.007 | 0.482±0.005 | 0.299±0.011 | 0.553±0.014 |
| + LoRA$_{r=4}$ | 0.597 M | 543 ms | 0.092±0.003 | 3.895±0.075 | 0.850±0.007 | 0.485±0.006 | 0.292±0.021 | 0.522±0.021 |
| + DyLoRA$_{r=4}$ | 0.597 M | 542 ms | 0.093±0.006 | 3.910±0.159 | 0.851±0.006 | 0.483±0.008 | 0.292±0.017 | 0.532±0.021 |
| + AdaLoRA$_{r=4}$ | 0.598 M | 584 ms | 0.093±0.005 | 3.933±0.135 | 0.850±0.005 | 0.486±0.005 | 0.295±0.008 | 0.533±0.012 |
| + IncreLoRA$_{r=4}$ | 0.621 M | 749 ms | 0.084±0.003 | 3.660±0.114 | 0.861±0.007 | 0.486±0.008 | 0.301±0.007 | 0.552±0.013 |
| **CE-SSL$_{r=16}$** | **1.568 M** | **243 ms** | **0.086±0.004** | **3.740±0.134** | **0.862±0.006** | **0.507±0.007** | **0.317±0.022** | **0.561±0.024** |
| **CE-SSL$_{r=4}$** | **0.458 M** | **241 ms** | **0.085±0.002** | **3.741±0.068** | **0.862±0.007** | **0.503±0.006** | **0.316±0.013** | **0.560±0.015** |
| **PTB-XL Dataset** | | | | | | | | |
| FixMatch | 50.494 M | 553 ms | 0.034±0.002 | 2.767±0.053 | 0.898±0.003 | 0.536±0.006 | 0.340±0.006 | 0.556±0.010 |
| + LoRA$_{r=16}$ | 2.135 M | 603 ms | 0.030±0.001 | 2.632±0.050 | 0.906±0.005 | 0.532±0.005 | 0.352±0.008 | 0.571±0.012 |
| + DyLoRA$_{r=16}$ | 2.135 M | 603 ms | 0.031±0.001 | 2.683±0.060 | 0.903±0.006 | 0.533±0.008 | 0.344±0.017 | 0.567±0.017 |
| + AdaLoRA$_{r=16}$ | 2.137 M | 652 ms | 0.031±0.001 | 2.636±0.022 | 0.905±0.004 | 0.529±0.006 | 0.350±0.006 | 0.571±0.005 |
| + IncreLoRA$_{r=16}$ | 2.005 M | 1090 ms | 0.029±0.001 | 2.567±0.029 | 0.908±0.004 | 0.540±0.007 | 0.364±0.007 | 0.586±0.013 |
| + LoRA$_{r=4}$ | 0.598 M | 602 ms | 0.030±0.001 | 2.609±0.056 | 0.908±0.005 | 0.530±0.006 | 0.345±0.008 | 0.571±0.013 |
| + DyLoRA$_{r=4}$ | 0.598 M | 600 ms | 0.030±0.001 | 2.607±0.038 | 0.907±0.003 | 0.530±0.005 | 0.342±0.022 | 0.564±0.016 |
| + AdaLoRA$_{r=4}$ | 0.598 M | 650 ms | 0.030±0.001 | 2.610±0.024 | 0.907±0.003 | 0.534±0.005 | 0.354±0.003 | 0.578±0.006 |
| + IncreLoRA$_{r=4}$ | 0.623 M | 830 ms | 0.028±0.000 | 2.548±0.009 | 0.912±0.002 | 0.542±0.005 | 0.362±0.013 | 0.586±0.012 |
| **CE-SSL$_{r=16}$** | **1.485 M** | **271 ms** | **0.027±0.001** | **2.539±0.033** | **0.913±0.003** | **0.550±0.004** | **0.369±0.005** | **0.588±0.003** |
| **CE-SSL$_{r=4}$** | **0.505 M** | **270 ms** | **0.027±0.001** | **2.529±0.019** | **0.914±0.003** | **0.547±0.003** | **0.372±0.006** | **0.599±0.010** |
| **Ningbo Dataset** | | | | | | | | |
| FixMatch | 50.496 M | 572 ms | 0.031±0.002 | 2.869±0.081 | 0.931±0.003 | 0.531±0.021 | 0.349±0.014 | 0.575±0.015 |
| + LoRA$_{r=16}$ | 2.137 M | 625 ms | 0.028±0.001 | 2.759±0.044 | 0.927±0.003 | 0.518±0.017 | 0.345±0.008 | 0.580±0.012 |
| + DyLoRA$_{r=16}$ | 2.137 M | 625 ms | 0.028±0.002 | 2.735±0.061 | 0.928±0.004 | 0.502±0.022 | 0.331±0.009 | 0.564±0.014 |
| + AdaLoRA$_{r=16}$ | 2.139 M | 674 ms | 0.030±0.001 | 2.799±0.084 | 0.927±0.002 | 0.507±0.020 | 0.330±0.010 | 0.565±0.018 |
| + IncreLoRA$_{r=16}$ | 2.145 M | 1124 ms | 0.027±0.001 | 2.679±0.044 | 0.932±0.002 | 0.521±0.014 | 0.337±0.008 | 0.569±0.015 |
| + LoRA$_{r=4}$ | 0.600 M | 624 ms | 0.028±0.001 | 2.722±0.058 | 0.929±0.002 | 0.516±0.014 | 0.338±0.015 | 0.565±0.016 |
| + DyLoRA$_{r=4}$ | 0.600 M | 621 ms | 0.028±0.001 | 2.717±0.039 | 0.929±0.002 | 0.510±0.018 | 0.335±0.011 | 0.569±0.017 |
| + AdaLoRA$_{r=4}$ | 0.600 M | 672 ms | 0.030±0.003 | 2.790±0.083 | 0.927±0.004 | 0.505±0.019 | 0.325±0.006 | 0.558±0.017 |
| + IncreLoRA$_{r=4}$ | 0.622 M | 858 ms | 0.027±0.001 | 2.667±0.018 | 0.931±0.001 | 0.519±0.013 | 0.338±0.010 | 0.569±0.018 |
| **CE-SSL$_{r=16}$** | **1.705 M** | **282 ms** | **0.027±0.001** | **2.701±0.051** | **0.933±0.003** | **0.531±0.018** | **0.356±0.013** | **0.588±0.021** |
| **CE-SSL$_{r=4}$** | **0.507 M** | **282 ms** | **0.026±0.001** | **2.661±0.058** | **0.934±0.004** | **0.525±0.018** | **0.352±0.013** | **0.587±0.020** |
| **Chapman Dataset** | | | | | | | | |
| FixMatch | 50.492 M | 495 ms | 0.038±0.004 | 2.439±0.092 | 0.905±0.010 | 0.538±0.011 | 0.357±0.009 | 0.522±0.020 |
| + LoRA$_{r=16}$ | 2.134 M | 540 ms | 0.038±0.002 | 2.424±0.053 | 0.899±0.009 | 0.532±0.021 | 0.345±0.009 | 0.514±0.024 |
| + DyLoRA$_{r=16}$ | 2.134 M | 540 ms | 0.037±0.004 | 2.401±0.095 | 0.903±0.008 | 0.531±0.013 | 0.345±0.013 | 0.518±0.027 |
| + AdaLoRA$_{r=16}$ | 2.135 M | 583 ms | 0.037±0.002 | 2.394±0.066 | 0.894±0.009 | 0.511±0.020 | 0.343±0.004 | 0.493±0.013 |
| + IncreLoRA$_{r=16}$ | 2.159 M | 962 ms | 0.035±0.001 | 2.337±0.034 | 0.889±0.010 | 0.515±0.013 | 0.342±0.011 | 0.496±0.017 |
| + LoRA$_{r=4}$ | 0.596 M | 539 ms | 0.036±0.002 | 2.372±0.051 | 0.901±0.005 | 0.535±0.007 | 0.357±0.010 | 0.521±0.022 |
| + DyLoRA$_{r=4}$ | 0.596 M | 537 ms | 0.036±0.002 | 2.371±0.034 | 0.903±0.005 | 0.528±0.015 | 0.350±0.011 | 0.515±0.020 |
| + AdaLoRA$_{r=4}$ | 0.597 M | 580 ms | 0.036±0.002 | 2.362±0.029 | 0.901±0.008 | 0.521±0.018 | 0.344±0.006 | 0.508±0.006 |
| + IncreLoRA$_{r=4}$ | 0.618 M | 746 ms | 0.035±0.002 | 2.348±0.046 | 0.888±0.010 | 0.510±0.016 | 0.344±0.008 | 0.502±0.020 |
| **CE-SSL$_{r=16}$** | **1.601 M** | **241 ms** | **0.035±0.002** | **2.362±0.049** | **0.909±0.007** | **0.553±0.013** | **0.367±0.008** | **0.540±0.019** |
| **CE-SSL$_{r=4}$** | **0.402 M** | **240 ms** | **0.034±0.001** | **2.334±0.033** | **0.908±0.008** | **0.538±0.014** | **0.361±0.009** | **0.531±0.019** |

Table 18: Performance comparisons between CE-SSL and parameter-efficient semi-supervised baselines on the large backbone. The average performance on all CVDs within each dataset is shown across six seeds. The standard deviation is also reported for the evaluation metrics.

| Methods | Params ↓ | Time/iter ↓ | Ranking Loss ↓ | Coverage ↓ | Macro AUC ↑ | MAP ↑ | Macro $G_{\beta=2}$ ↑ | Macro $F_{\beta=2}$ ↑ |
|---|---|---|---|---|---|---|---|---|
| **G12EC Dataset** | | | | | | | | |
| FixMatch | 113.489 M | 966 ms | 0.100±0.005 | 4.147±0.113 | 0.843±0.007 | 0.493±0.008 | 0.293±0.011 | 0.518±0.015 |
| + LoRA$_{r=16}$ | 3.201 M | 1023 ms | 0.094±0.003 | 3.983±0.134 | 0.849±0.004 | 0.492±0.006 | 0.294±0.010 | 0.530±0.020 |
| + DyLoRA$_{r=16}$ | 3.201 M | 1024 ms | 0.091±0.003 | 3.911±0.070 | 0.849±0.005 | 0.492±0.008 | 0.297±0.017 | 0.534±0.022 |
| + AdaLoRA$_{r=16}$ | 3.203 M | 1025 ms | 0.093±0.003 | 3.972±0.119 | 0.842±0.003 | 0.482±0.008 | 0.296±0.008 | 0.532±0.010 |
| + IncreLoRA$_{r=16}$ | 3.245 M | 1575 ms | 0.084±0.002 | 3.708±0.075 | 0.851±0.003 | 0.493±0.008 | 0.309±0.013 | 0.543±0.021 |
| + LoRA$_{r=4}$ | 0.896 M | 1021 ms | 0.092±0.005 | 3.895±0.130 | 0.851±0.005 | 0.493±0.007 | 0.296±0.010 | 0.530±0.012 |
| + DyLoRA$_{r=4}$ | 0.896 M | 1021 ms | 0.090±0.004 | 3.864±0.133 | 0.849±0.006 | 0.495±0.008 | 0.301±0.010 | 0.535±0.008 |
| + AdaLoRA$_{r=4}$ | 0.896 M | 1021 ms | 0.089±0.002 | 3.852±0.069 | 0.847±0.004 | 0.487±0.003 | 0.297±0.018 | 0.527±0.024 |
| + IncreLoRA$_{r=4}$ | 0.921 M | 1348 ms | 0.082±0.003 | 3.666±0.086 | 0.856±0.006 | 0.497±0.007 | 0.308±0.010 | 0.542±0.016 |
| **CE-SSL$_{r=16}$** | **2.658 M** | **453 ms** | **0.085±0.005** | **3.778±0.140** | **0.857±0.004** | **0.509±0.007** | **0.322±0.009** | **0.565±0.010** |
| **CE-SSL$_{r=4}$** | **0.761 M** | **453 ms** | **0.084±0.003** | **3.742±0.117** | **0.859±0.004** | **0.506±0.007** | **0.323±0.004** | **0.561±0.002** |
| **PTB-XL Dataset** | | | | | | | | |
| FixMatch | 113.490 M | 1072 ms | 0.035±0.003 | 2.805±0.102 | 0.894±0.004 | 0.521±0.006 | 0.342±0.007 | 0.560±0.012 |
| + LoRA$_{r=16}$ | 3.202 M | 1135 ms | 0.030±0.001 | 2.635±0.023 | 0.903±0.002 | 0.522±0.006 | 0.332±0.010 | 0.550±0.014 |
| + DyLoRA$_{r=16}$ | 3.202 M | 1135 ms | 0.031±0.001 | 2.674±0.030 | 0.906±0.002 | 0.528±0.002 | 0.346±0.010 | 0.566±0.008 |
| + AdaLoRA$_{r=16}$ | 3.203 M | 1138 ms | 0.033±0.000 | 2.716±0.019 | 0.894±0.003 | 0.517±0.006 | 0.345±0.008 | 0.558±0.006 |
| + IncreLoRA$_{r=16}$ | 3.167 M | 1758 ms | 0.031±0.001 | 2.660±0.030 | 0.898±0.004 | 0.519±0.004 | 0.348±0.012 | 0.566±0.012 |
| + LoRA$_{r=4}$ | 0.897 M | 1133 ms | 0.030±0.001 | 2.621±0.026 | 0.904±0.003 | 0.523±0.006 | 0.342±0.012 | 0.564±0.014 |
| + DyLoRA$_{r=4}$ | 0.897 M | 1133 ms | 0.030±0.001 | 2.632±0.030 | 0.903±0.004 | 0.525±0.004 | 0.339±0.011 | 0.570±0.010 |
| + AdaLoRA$_{r=4}$ | 0.897 M | 1134 ms | 0.032±0.001 | 2.699±0.052 | 0.897±0.004 | 0.516±0.004 | 0.339±0.007 | 0.567±0.007 |
| + IncreLoRA$_{r=4}$ | 0.920 M | 1493 ms | 0.030±0.000 | 2.631±0.016 | 0.901±0.002 | 0.524±0.005 | 0.351±0.005 | 0.573±0.010 |
| **CE-SSL$_{r=16}$** | **2.235 M** | **508 ms** | **0.030±0.002** | **2.618±0.061** | **0.909±0.004** | **0.537±0.004** | **0.358±0.005** | **0.587±0.008** |
| **CE-SSL$_{r=4}$** | **0.712 M** | **506 ms** | **0.029±0.001** | **2.602±0.028** | **0.908±0.003** | **0.535±0.004** | **0.356±0.006** | **0.582±0.008** |
| **Ningbo Dataset** | | | | | | | | |
| FixMatch | 113.493 M | 1111 ms | 0.033±0.002 | 2.962±0.070 | 0.926±0.004 | 0.513±0.024 | 0.337±0.018 | 0.563±0.027 |
| + LoRA$_{r=16}$ | 3.205 M | 1177 ms | 0.030±0.002 | 2.845±0.073 | 0.925±0.003 | 0.508±0.023 | 0.336±0.009 | 0.566±0.017 |
| + DyLoRA$_{r=16}$ | 3.205 M | 1176 ms | 0.030±0.002 | 2.834±0.093 | 0.927±0.003 | 0.504±0.024 | 0.326±0.013 | 0.553±0.019 |
| + AdaLoRA$_{r=16}$ | 3.206 M | 1180 ms | 0.031±0.001 | 2.855±0.074 | 0.920±0.002 | 0.491±0.017 | 0.315±0.011 | 0.545±0.018 |
| + IncreLoRA$_{r=16}$ | 3.247 M | 1766 ms | 0.028±0.001 | 2.750±0.030 | 0.926±0.004 | 0.502±0.024 | 0.330±0.009 | 0.562±0.017 |
| + LoRA$_{r=4}$ | 0.900 M | 1173 ms | 0.029±0.001 | 2.802±0.053 | 0.925±0.002 | 0.510±0.023 | 0.332±0.010 | 0.563±0.018 |
| + DyLoRA$_{r=4}$ | 0.900 M | 1174 ms | 0.030±0.001 | 2.803±0.035 | 0.926±0.003 | 0.505±0.028 | 0.330±0.010 | 0.564±0.020 |
| + AdaLoRA$_{r=4}$ | 0.900 M | 1175 ms | 0.031±0.003 | 2.846±0.128 | 0.922±0.002 | 0.495±0.023 | 0.326±0.012 | 0.555±0.023 |
| + IncreLoRA$_{r=4}$ | 0.923 M | 1545 ms | 0.028±0.001 | 2.736±0.028 | 0.927±0.003 | 0.499±0.016 | 0.329±0.011 | 0.561±0.017 |
| **CE-SSL$_{r=16}$** | **2.234 M** | **530 ms** | **0.029±0.001** | **2.779±0.027** | **0.931±0.002** | **0.523±0.027** | **0.344±0.010** | **0.578±0.013** |
| **CE-SSL$_{r=4}$** | **0.740 M** | **528 ms** | **0.028±0.001** | **2.741±0.039** | **0.930±0.002** | **0.513±0.018** | **0.346±0.007** | **0.584±0.009** |
| **Chapman Dataset** | | | | | | | | |
| FixMatch | 113.487 M | 960 ms | 0.042±0.002 | 2.545±0.048 | 0.900±0.008 | 0.534±0.014 | 0.350±0.013 | 0.518±0.026 |
| + LoRA$_{r=16}$ | 3.200 M | 1016 ms | 0.040±0.003 | 2.501±0.079 | 0.896±0.004 | 0.541±0.005 | 0.343±0.015 | 0.509±0.024 |
| + DyLoRA$_{r=16}$ | 3.200 M | 1016 ms | 0.040±0.004 | 2.484±0.080 | 0.897±0.006 | 0.537±0.009 | 0.352±0.006 | 0.524±0.021 |
| + AdaLoRA$_{r=16}$ | 3.201 M | 1018 ms | 0.037±0.002 | 2.416±0.039 | 0.894±0.003 | 0.531±0.011 | 0.343±0.011 | 0.509±0.021 |
| + IncreLoRA$_{r=16}$ | 3.231 M | 1541 ms | 0.033±0.002 | 2.309±0.037 | 0.895±0.008 | 0.539±0.014 | 0.355±0.010 | 0.517±0.017 |
| + LoRA$_{r=4}$ | 0.894 M | 1016 ms | 0.038±0.001 | 2.452±0.039 | 0.901±0.006 | 0.542±0.008 | 0.338±0.015 | 0.512±0.028 |
| + DyLoRA$_{r=4}$ | 0.894 M | 1015 ms | 0.037±0.003 | 2.415±0.068 | 0.899±0.005 | 0.533±0.015 | 0.341±0.013 | 0.505±0.021 |
| + AdaLoRA$_{r=4}$ | 0.895 M | 1015 ms | 0.036±0.001 | 2.412±0.039 | 0.898±0.005 | 0.543±0.007 | 0.341±0.007 | 0.517±0.019 |
| + IncreLoRA$_{r=4}$ | 0.920 M | 1340 ms | 0.033±0.001 | 2.316±0.028 | 0.897±0.007 | 0.539±0.025 | 0.360±0.017 | 0.528±0.019 |
| **CE-SSL$_{r=16}$** | **2.205 M** | **451 ms** | **0.037±0.001** | **2.417±0.035** | **0.904±0.004** | **0.556±0.006** | **0.371±0.010** | **0.552±0.018** |
| **CE-SSL$_{r=4}$** | **0.716 M** | **451 ms** | **0.036±0.001** | **2.404±0.041** | **0.902±0.006** | **0.550±0.008** | **0.365±0.006** | **0.548±0.010** |

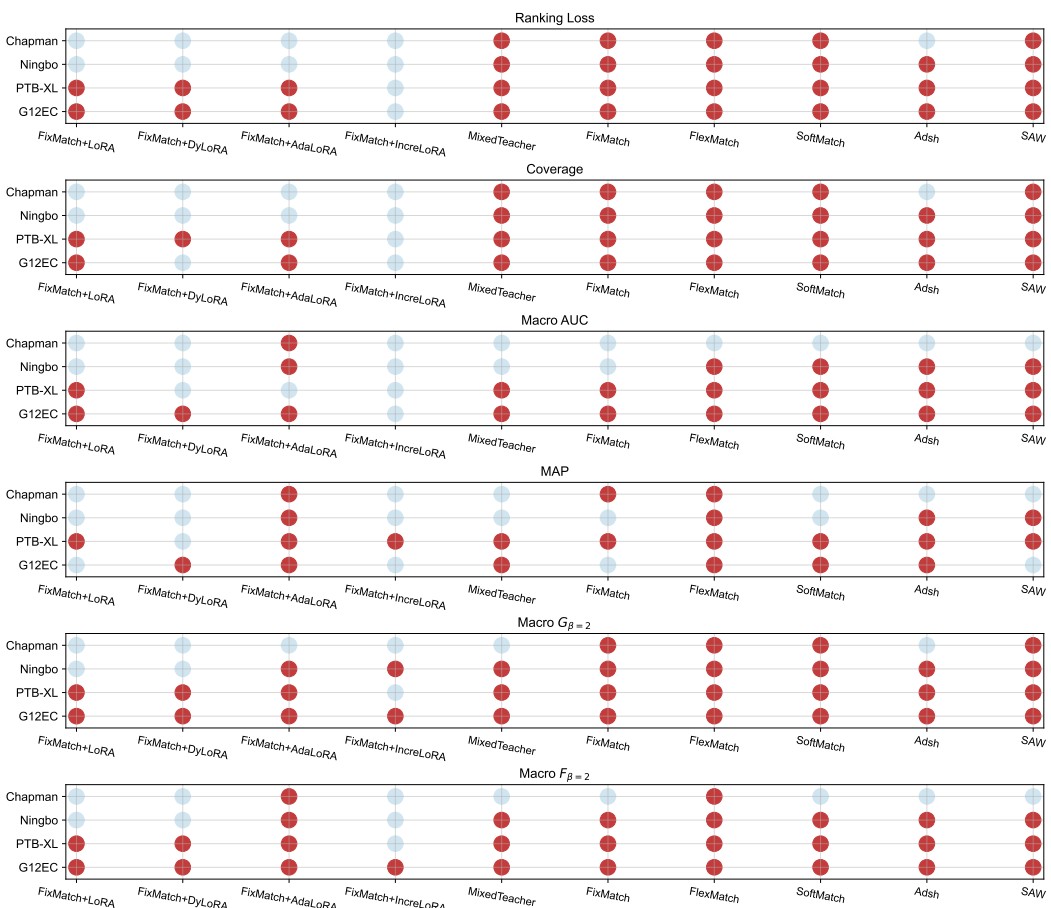

Figure 7: Paired t-test results for model performance on the base backbone. Specifically, we use the paired t-test to check if the proposed CE-SSL significantly outperforms other baseline models on four datasets and six evaluation metrics. Each circle represents a paired t-test result between CE-SSL and a baseline model. The colors of the circles denote the significance levels (two-sided $p$-value) of the test results after false discovery rate (FDR) correction for multiple testing. The red circle indicates that the corresponding two-sided $p$-value is less than 0.05.

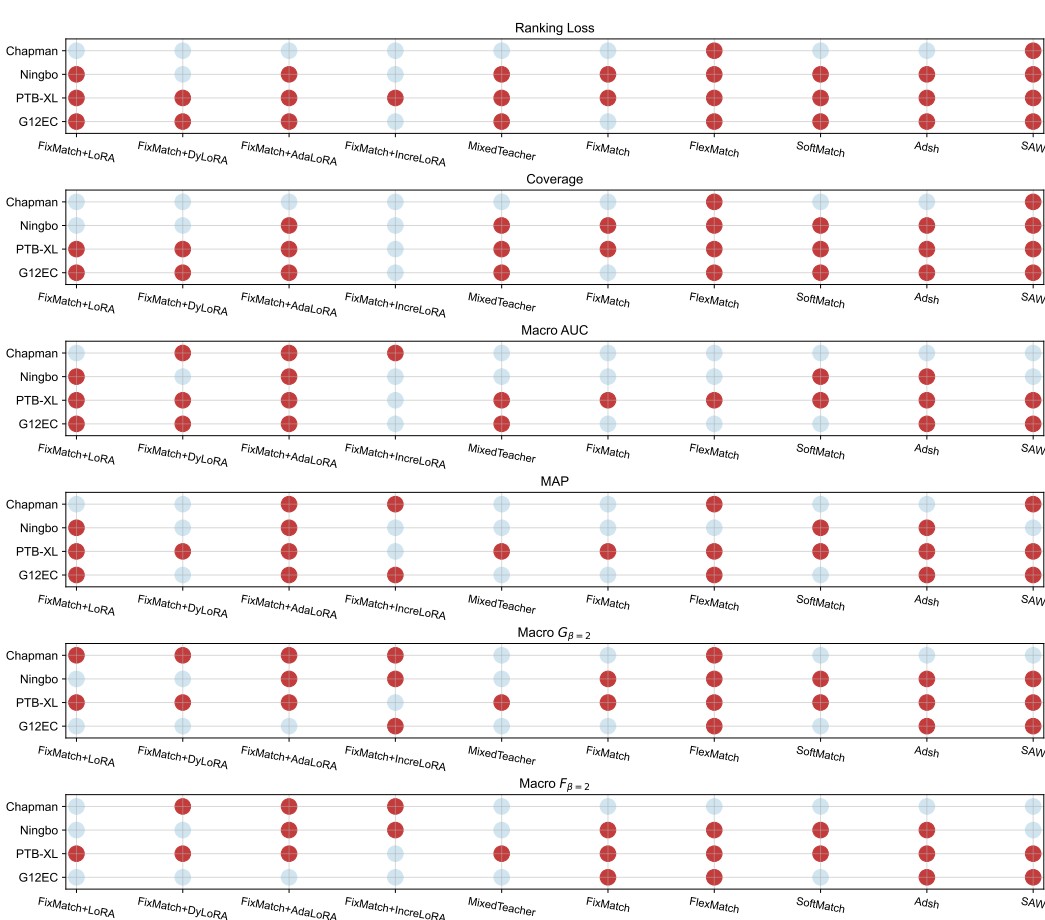

Figure 8: Paired t-test results for model performance on the medium backbone. Specifically, we use the paired t-test to check if the proposed CE-SSL significantly outperforms other baseline models on four datasets and six evaluation metrics. Each circle represents a paired t-test result between CE-SSL and a baseline model. The colors of the circles denote the significance levels (two-sided $p$-value) of the test results after false discovery rate (FDR) correction for multiple testing. The red circle indicates that the corresponding two-sided $p$-value is less than 0.05.

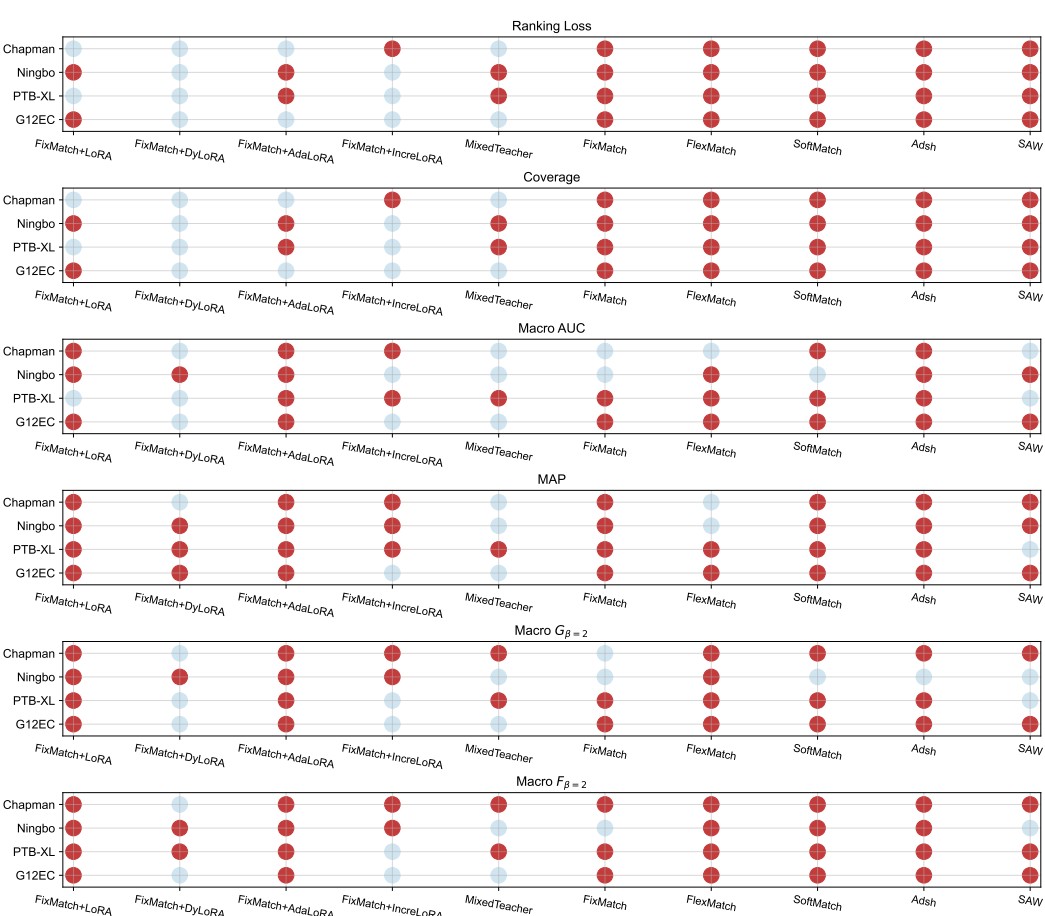

Figure 9: Paired t-test results for model performance on the large backbone. Specifically, we use the paired t-test to check if the proposed CE-SSL significantly outperforms other baseline models on four datasets and six evaluation metrics. Each circle represents a paired t-test result between CE-SSL and a baseline model. The colors of the circles denote the significance levels (two-sided $p$-value) of the test results after false discovery rate (FDR) correction for multiple testing. The red circle indicates that the corresponding two-sided $p$-value is less than 0.05.

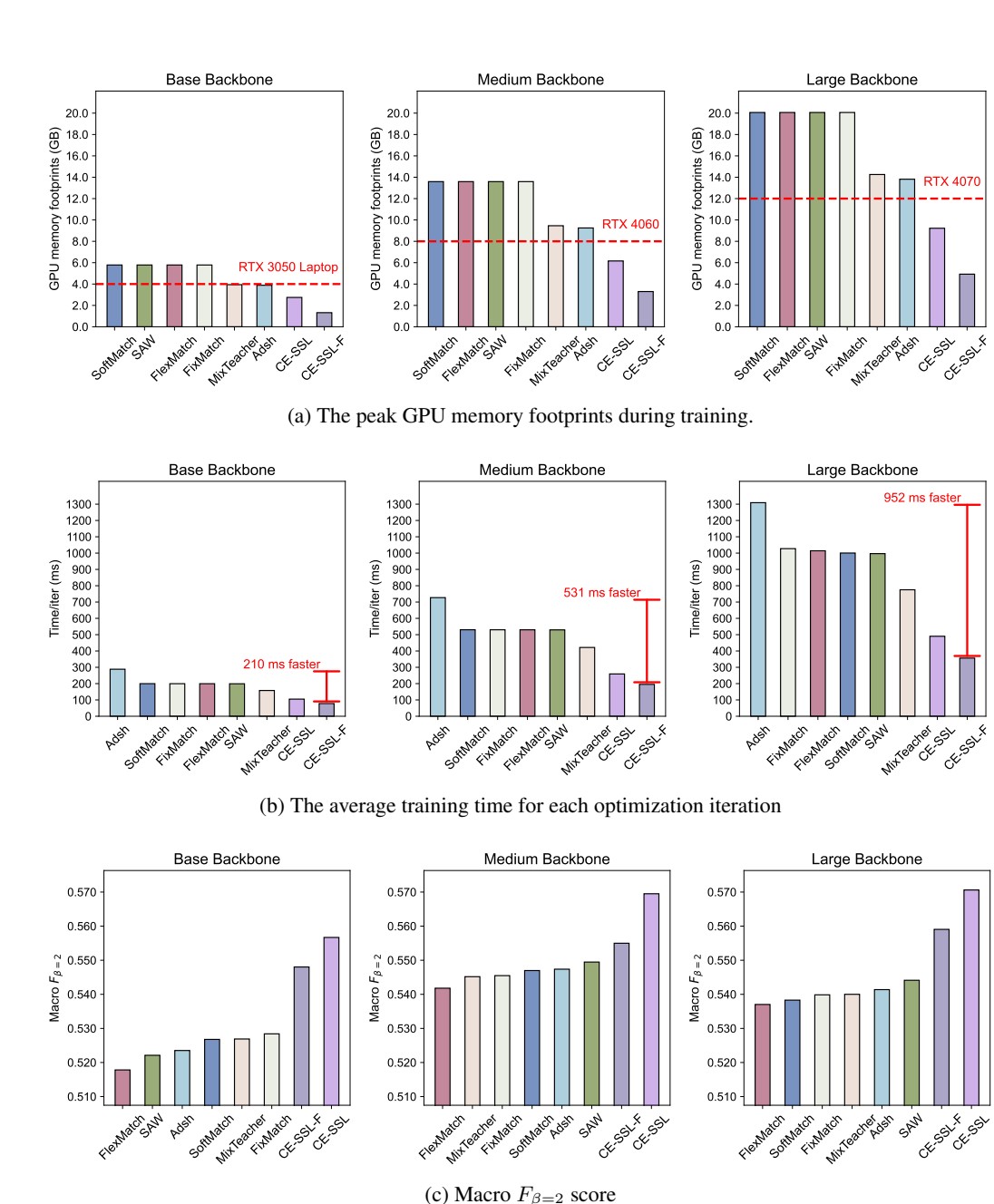

(a) The peak GPU memory footprints during training.

(b) The average training time for each optimization iteration

(c) Macro $F_{\beta=2}$ score

Figure 10: Evaluation of the average detection performance and computational efficiency of various semi-supervised methods across four downstream datasets. Specifically, the macro $F_{\beta=2}$ scores are used to evaluate their CVDs detection performance. Additionally, the average training time per iteration and the maximum GPU memory usage are presented to evaluate computational efficiency. Methods exceeding the GPU memory thresholds, indicated by the red dashed lines, are not deployable on the corresponding NVIDIA GPU cards.

Table 19: Ablation study of the proposed CE-SSL on the base backbone. 'w/o random deactivation' represents the CE-SSL without the random deactivation technique, and the deactivation probability $p$ is set to zero. 'w/o rank allocation' represents the CE-SSL without the one-shot rank allocation, and all pre-trained weights are updated with the initial rank $r$. 'w/o semi-supervised BN' denotes the CE-SSL without the semi-supervised batch normalization for lightweight semi-supervised learning.

| Methods | Time/iter ↓ | Ranking Loss ↓ | Coverage ↓ | Macro AUC ↑ | MAP ↑ | Macro $G_{\beta=2}$ ↑ | Macro $F_{\beta=2}$ ↑ |
|---|---|---|---|---|---|---|---|
| **G12EC Dataset** | | | | | | | |
| w/o random deactivation | 104ms | 0.095±0.004 | 3.954±0.163 | 0.848±0.007 | 0.470±0.007 | 0.294±0.015 | 0.536±0.021 |
| w/o rank allocation | 97ms | 0.092±0.002 | 3.848±0.049 | 0.849±0.007 | 0.467±0.009 | 0.294±0.016 | 0.537±0.019 |
| w/o semi-supervised BN | 78ms | 0.092±0.002 | 3.895±0.104 | 0.854±0.004 | 0.475±0.011 | 0.297±0.021 | 0.536±0.029 |
| **CE-SSL** | **98ms** | **0.092±0.002** | **3.867±0.088** | **0.855±0.005** | **0.476±0.006** | **0.307±0.016** | **0.551±0.017** |
| **PTB-XL Dataset** | | | | | | | |
| w/o random deactivation | 115ms | 0.034±0.002 | 2.741±0.062 | 0.890±0.005 | 0.516±0.009 | 0.328±0.012 | 0.554±0.011 |
| w/o rank allocation | 108ms | 0.032±0.001 | 2.692±0.046 | 0.895±0.003 | 0.530±0.005 | 0.332±0.011 | 0.560±0.014 |
| w/o semi-supervised BN | 87ms | 0.031±0.002 | 2.670±0.064 | 0.899±0.004 | 0.532±0.006 | 0.332±0.010 | 0.565±0.007 |
| **CE-SSL** | **110ms** | **0.031±0.000** | **2.641±0.020** | **0.901±0.003** | **0.530±0.005** | **0.346±0.006** | **0.578±0.006** |
| **Ningbo Dataset** | | | | | | | |
| w/o random deactivation | 121ms | 0.032±0.003 | 2.887±0.085 | 0.925±0.005 | 0.497±0.015 | 0.321±0.013 | 0.553±0.017 |
| w/o rank allocation | 114ms | 0.030±0.001 | 2.801±0.023 | 0.928±0.002 | 0.497±0.021 | 0.325±0.010 | 0.563±0.014 |
| w/o semi-supervised BN | 92ms | 0.031±0.001 | 2.821±0.058 | 0.929±0.003 | 0.499±0.017 | 0.325±0.012 | 0.559±0.018 |
| **CE-SSL** | **115ms** | **0.030±0.001** | **2.805±0.063** | **0.928±0.002** | **0.505±0.019** | **0.334±0.011** | **0.569±0.014** |
| **Chapman Dataset** | | | | | | | |
| w/o random deactivation | 102ms | 0.041±0.003 | 2.505±0.080 | 0.895±0.010 | 0.526±0.005 | 0.335±0.012 | 0.514±0.015 |
| w/o rank allocation | 96ms | 0.041±0.001 | 2.503±0.040 | 0.892±0.008 | 0.527±0.012 | 0.346±0.007 | 0.514±0.018 |
| w/o semi-supervised BN | 77ms | 0.040±0.002 | 2.468±0.050 | 0.896±0.010 | 0.533±0.010 | 0.350±0.020 | 0.527±0.026 |
| **CE-SSL** | **97ms** | **0.040±0.002** | **2.483±0.055** | **0.896±0.006** | **0.536±0.004** | **0.355±0.005** | **0.530±0.008** |

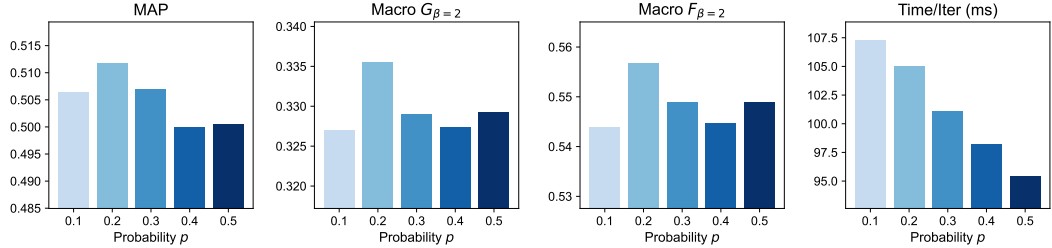

Figure 11: Effect of the deactivation probability. The averaged performance and training time of the CE-SSL across four datasets and six random seeds under different deactivation probabilities $p$ are presented.

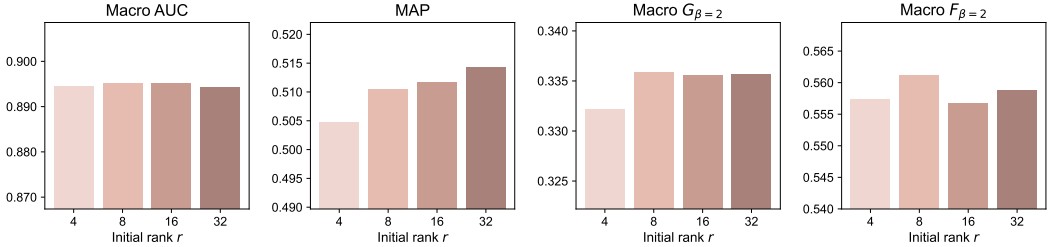

Figure 12: Effect of the rank initialization. Averaged performance of the CE-SSL across four datasets and six random seeds under different initial ranks $r$.

Table 20: Ablation study of the proposed CE-SSL on the medium backbone. 'w/o random deactivation' represents the CE-SSL without the random deactivation technique, and the deactivation probability $p$ is set to zero. 'w/o rank allocation' represents the CE-SSL without the one-shot rank allocation, and all pre-trained weights are updated with the initial rank $r$. 'w/o semi-supervised BN' denotes the CE-SSL without the semi-supervised batch normalization for lightweight semi-supervised learning.

| Methods | Time/iter ↓ | Ranking Loss ↓ | Coverage ↓ | Macro AUC ↑ | MAP ↑ | Macro $G_{\beta=2}$ ↑ | Macro $F_{\beta=2}$ ↑ |
|---|---|---|---|---|---|---|---|
| **G12EC Dataset** | | | | | | | |
| w/o random deactivation | 259ms | 0.091±0.004 | 3.887±0.138 | 0.855±0.005 | 0.497±0.010 | 0.304±0.015 | 0.540±0.022 |
| w/o rank allocation | 241ms | 0.087±0.003 | 3.795±0.110 | 0.861±0.005 | 0.506±0.004 | 0.306±0.017 | 0.551±0.022 |
| w/o semi-supervised BN | 189ms | 0.085±0.005 | 3.750±0.159 | 0.864±0.007 | 0.506±0.008 | 0.308±0.021 | 0.548±0.024 |
| **CE-SSL** | **243ms** | **0.086±0.004** | **3.740±0.134** | **0.862±0.006** | **0.507±0.007** | **0.317±0.022** | **0.561±0.024** |
| **PTB-XL Dataset** | | | | | | | |
| w/o random deactivation | 289ms | 0.030±0.002 | 2.630±0.064 | 0.905±0.003 | 0.534±0.006 | 0.351±0.006 | 0.577±0.013 |
| w/o rank allocation | 269ms | 0.028±0.001 | 2.563±0.028 | 0.912±0.005 | 0.540±0.006 | 0.351±0.012 | 0.575±0.016 |
| w/o semi-supervised BN | 213ms | 0.028±0.001 | 2.563±0.035 | 0.911±0.003 | 0.547±0.005 | 0.358±0.010 | 0.582±0.016 |
| **CE-SSL** | **271ms** | **0.027±0.001** | **2.539±0.033** | **0.913±0.003** | **0.550±0.004** | **0.369±0.005** | **0.588±0.003** |
| **Ningbo Dataset** | | | | | | | |
| w/o random deactivation | 301ms | 0.028±0.001 | 2.744±0.046 | 0.930±0.003 | 0.516±0.021 | 0.336±0.017 | 0.558±0.029 |
| w/o rank allocation | 281ms | 0.028±0.001 | 2.736±0.055 | 0.932±0.003 | 0.518±0.022 | 0.343±0.017 | 0.574±0.023 |
| w/o semi-supervised BN | 224ms | 0.027±0.000 | 2.671±0.028 | 0.934±0.002 | 0.525±0.020 | 0.346±0.018 | 0.576±0.024 |
| **CE-SSL** | **282ms** | **0.027±0.001** | **2.701±0.051** | **0.933±0.003** | **0.531±0.018** | **0.356±0.013** | **0.588±0.021** |
| **Chapman Dataset** | | | | | | | |
| w/o random deactivation | 256ms | 0.036±0.002 | 2.388±0.043 | 0.911±0.006 | 0.549±0.016 | 0.353±0.009 | 0.530±0.013 |
| w/o rank allocation | 240ms | 0.037±0.002 | 2.397±0.055 | 0.906±0.007 | 0.537±0.010 | 0.349±0.014 | 0.515±0.022 |
| w/o semi-supervised BN | 188ms | 0.035±0.001 | 2.349±0.019 | 0.912±0.006 | 0.555±0.016 | 0.356±0.007 | 0.525±0.018 |
| **CE-SSL** | **241ms** | **0.035±0.002** | **2.362±0.049** | **0.909±0.007** | **0.553±0.013** | **0.367±0.008** | **0.540±0.019** |

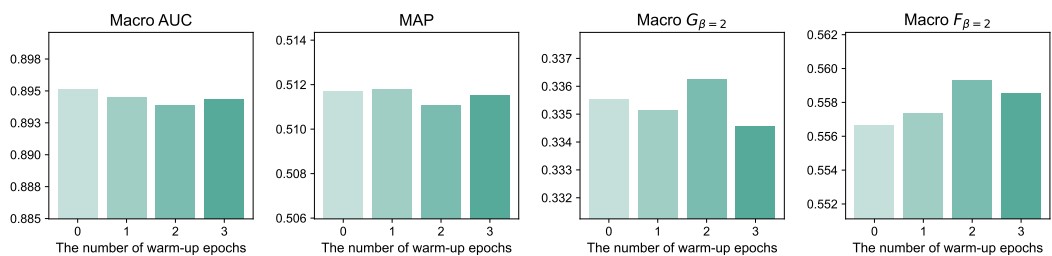

Figure 13: Effect of warmup epochs for rank allocation. Averaged performance of the CE-SSL across four datasets and six random seeds under different numbers of warmup epochs for rank allocation.

Table 21: Ablation study of the proposed CE-SSL on the large backbone. 'w/o random deactivation' represents the CE-SSL without the random deactivation technique, and the deactivation probability $p$ is set to zero. 'w/o rank allocation' represents the CE-SSL without the one-shot rank allocation, and all pre-trained weights are updated with the initial rank $r$. 'w/o semi-supervised BN' denotes the CE-SSL without the semi-supervised batch normalization for lightweight semi-supervised learning.

| Methods | Time/iter ↓ | Ranking Loss ↓ | Coverage ↓ | Macro AUC ↑ | MAP ↑ | Macro $G_{\beta=2}$ ↑ | Macro $F_{\beta=2}$ ↑ |
|---|---|---|---|---|---|---|---|
| **G12EC Dataset** | | | | | | | |
| w/o random deactivation | 483ms | 0.092±0.005 | 3.948±0.164 | 0.850±0.005 | 0.498±0.006 | 0.309±0.008 | 0.547±0.013 |
| w/o rank allocation | 450ms | 0.088±0.003 | 3.830±0.100 | 0.855±0.002 | 0.499±0.005 | 0.312±0.009 | 0.551±0.016 |
| w/o semi-supervised BN | 332ms | 0.088±0.005 | 3.839±0.129 | 0.855±0.005 | 0.506±0.008 | 0.314±0.014 | 0.552±0.018 |
| **CE-SSL** | **453ms** | **0.085±0.005** | **3.778±0.140** | **0.857±0.004** | **0.509±0.007** | **0.322±0.009** | **0.565±0.010** |
| **PTB-XL Dataset** | | | | | | | |
| w/o random deactivation | 542ms | 0.030±0.001 | 2.612±0.026 | 0.907±0.004 | 0.531±0.004 | 0.349±0.008 | 0.572±0.012 |
| w/o rank allocation | 501ms | 0.030±0.001 | 2.642±0.038 | 0.909±0.005 | 0.534±0.003 | 0.340±0.016 | 0.562±0.019 |
| w/o semi-supervised BN | 373ms | 0.030±0.001 | 2.630±0.046 | 0.910±0.003 | 0.540±0.006 | 0.360±0.010 | 0.592±0.008 |
| **CE-SSL** | **508ms** | **0.030±0.002** | **2.618±0.061** | **0.909±0.004** | **0.537±0.004** | **0.358±0.005** | **0.587±0.008** |
| **Ningbo Dataset** | | | | | | | |
| w/o random deactivation | 563ms | 0.031±0.002 | 2.860±0.094 | 0.927±0.004 | 0.513±0.026 | 0.333±0.013 | 0.567±0.022 |
| w/o rank allocation | 523ms | 0.029±0.001 | 2.757±0.043 | 0.930±0.001 | 0.514±0.026 | 0.335±0.013 | 0.568±0.022 |
| w/o semi-supervised BN | 392ms | 0.028±0.001 | 2.759±0.039 | 0.931±0.002 | 0.519±0.024 | 0.343±0.013 | 0.576±0.021 |
| **CE-SSL** | **529ms** | **0.029±0.001** | **2.779±0.027** | **0.931±0.002** | **0.523±0.027** | **0.344±0.010** | **0.578±0.013** |
| **Chapman Dataset** | | | | | | | |
| w/o random deactivation | 480ms | 0.038±0.001 | 2.438±0.041 | 0.905±0.004 | 0.549±0.006 | 0.348±0.013 | 0.529±0.021 |
| w/o rank allocation | 447ms | 0.038±0.002 | 2.448±0.060 | 0.904±0.008 | 0.551±0.006 | 0.352±0.009 | 0.520±0.016 |
| w/o semi-supervised BN | 329ms | 0.037±0.002 | 2.411±0.051 | 0.903±0.006 | 0.554±0.009 | 0.366±0.005 | 0.546±0.011 |
| **CE-SSL** | **450ms** | **0.037±0.001** | **2.417±0.035** | **0.904±0.004** | **0.556±0.006** | **0.371±0.010** | **0.552±0.018** |

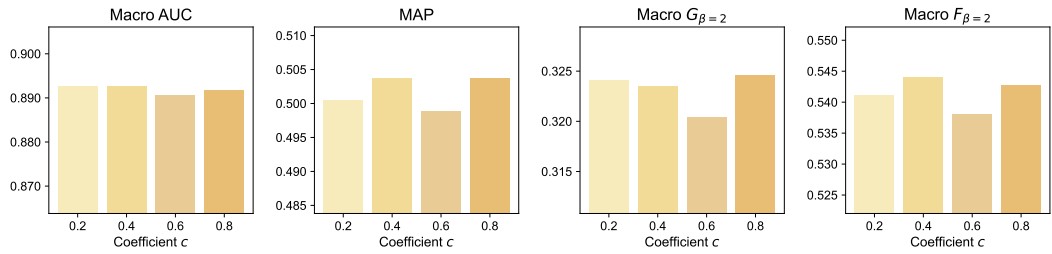

Figure 14: Effect of the ratio of important weight matrices. We adjust the ratio of the important weight matrices to the total number of weight matrices and report the averaged performance across four datasets and six random seeds. Important weights are adapted with rank $r$ while the remaining weights are adapted with rank $\frac{1}{2}r$.

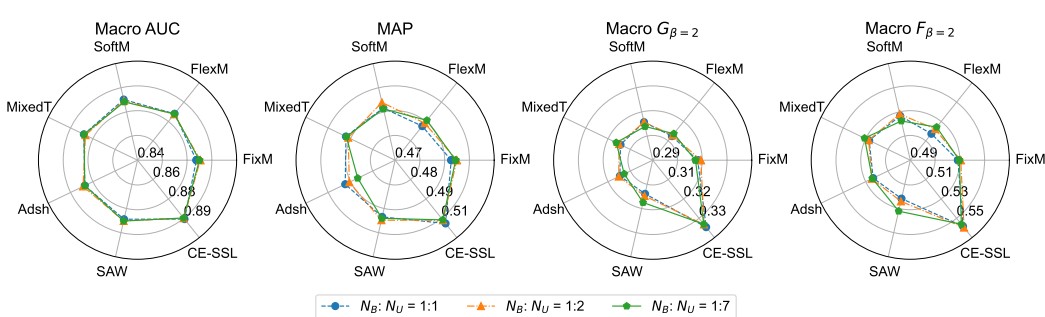

Figure 15: Effect of the batch size of unlabeled data. $N_B : N_U$ denotes the ratio between the batch size of labeled data and unlabeled data during model training. The averaged performance of different semi-supervised methods across four datasets and six random seeds is presented. For simplicity, 'MixedT,' 'SoftM,' 'FixM,' and 'FlexM' denote the MixedTeacher, SoftMatch, FixMatch, and FlexMatch methods, respectively.

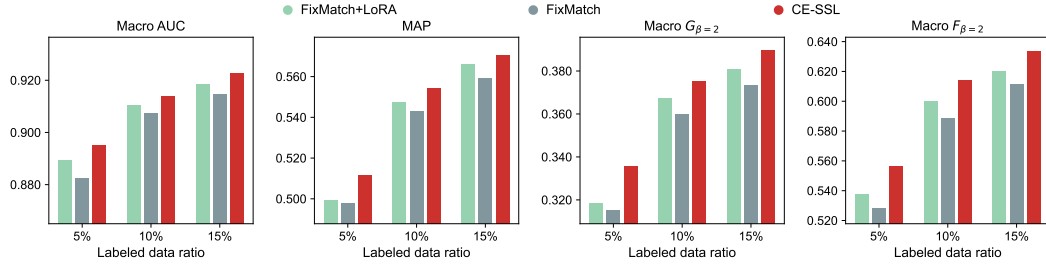

Figure 16: Effect of the ratio of labeled samples for model training. We adjust the ratio of the labeled samples in the dataset from 0.05 to 0.15 and report the averaged performance of different models across four datasets and six random seeds.

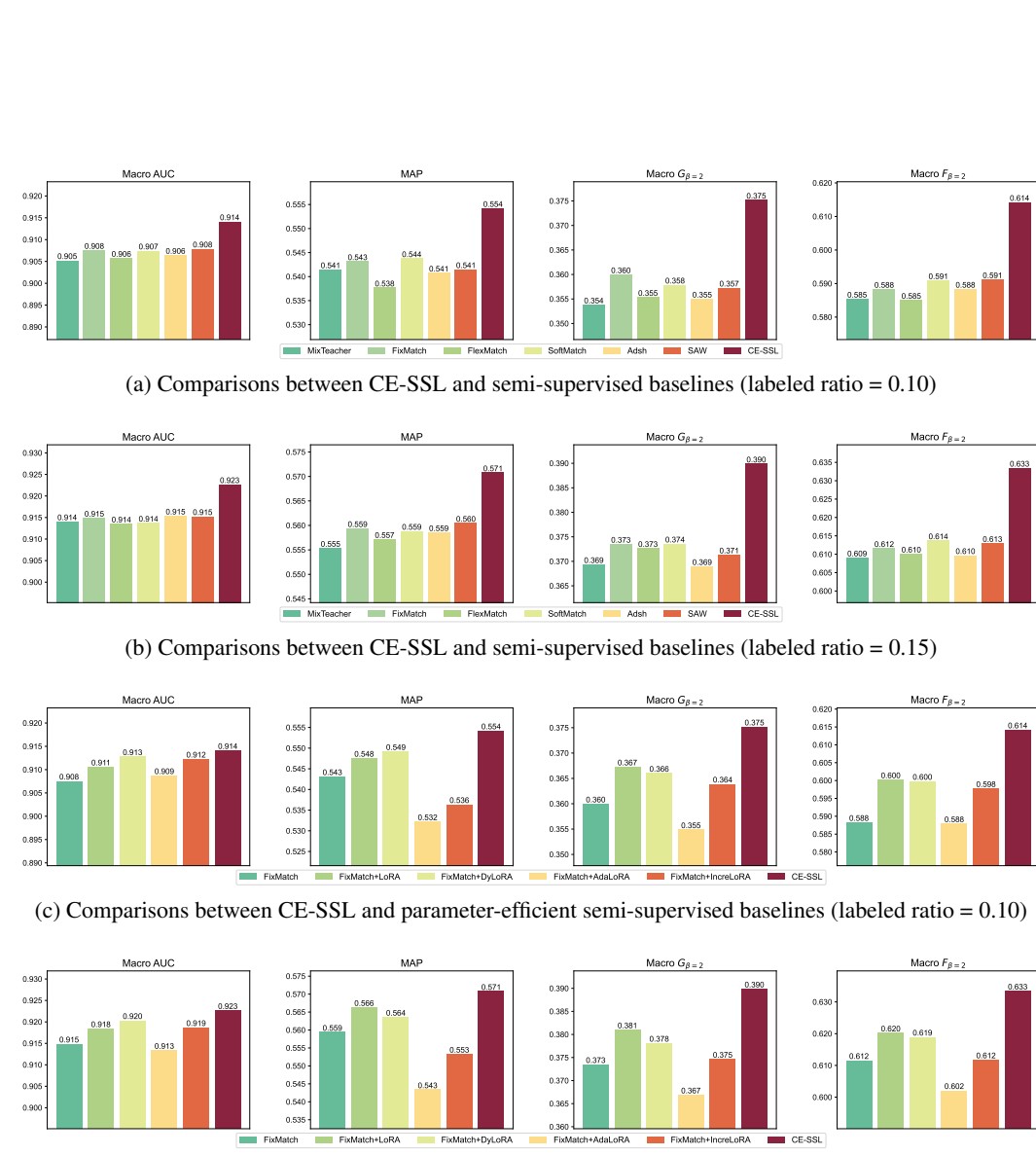

(a) Comparisons between CE-SSL and semi-supervised baselines (labeled ratio = 0.10)

(b) Comparisons between CE-SSL and semi-supervised baselines (labeled ratio = 0.15)

(c) Comparisons between CE-SSL and parameter-efficient semi-supervised baselines (labeled ratio = 0.10)

(d) Comparisons between CE-SSL and parameter-efficient semi-supervised baselines (labeled ratio = 0.15)

Figure 17: Performance comparisons between CE-SSL and the baseline models under various labeled ratios using the base backbone.

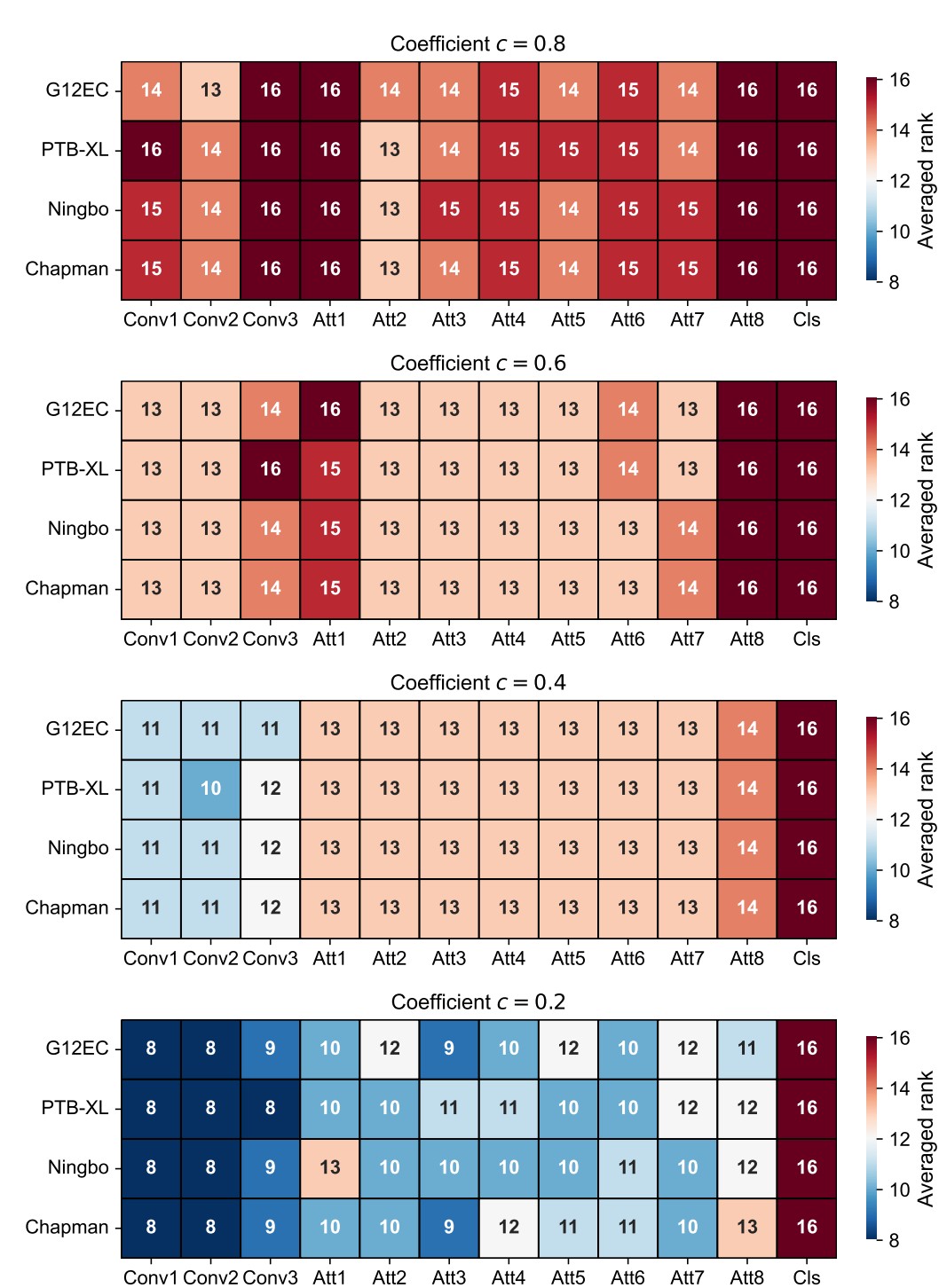

Figure 18: The rank distributions generated by the proposed one-shot rank allocation method on four datasets using the base backbone. Specifically, we visualize the allocated rank of each block in the backbone network, which is the average rank of the incremental matrices within the block. For simplicity, we present the abbreviations of different blocks.('Conv1': the 1-st convolution block; 'Att1': the 1-st self-attention block ; 'Cls': classification block.

