# OpenReview forum: "CE-SSL: Computation-Efficient Semi-Supervised Learning for ECG-based Cardiovascular Diseases Detection"
_ICLR.cc/2026/Conference — Submitted to ICLR 2026_

### Official Review · Reviewer_xbwK · 2025-10-28

**Soundness:** 2
**Presentation:** 2
**Contribution:** 2
**Rating:** 6
**Confidence:** 1

**Summary:**

The paper proposes CE-SSL, a computation-efficient semi-supervised learning framework for CVD detection from ECG under limited labeled data. It combines random-deactivation, one-shot rank allocation, and a lightweight semi-supervised pipeline to adapt pre-trained models efficiently. Experiments on four datasets show that CE-SSL achieves better detection performance than state-of-the-art methods while using less computation and storage, making it practical for clinical applications.

**Strengths:**

* The paper tackles the practical trade-off between detection performance and computational efficiency in ECG-based CVD detection under limited labels.

* Integrates low-rank adaptation, one-shot rank allocation, and semi-supervised BN into a single framework, which could reduce training costs compared to traditional SSL methods.

* Each component (RD-LoRA, one-shot rank allocation, lightweight SSL) is described with equations and reasoning. The logic of reducing computation while trying to maintain performance is consistent.

* Experiments on multiple downstream datasets show reductions in GPU usage, training time, and parameter size, which aligns with the goal of computational efficiency.

**Weaknesses:**

* Complexity vs. gain: The random-deactivation and one-shot rank allocation add methodological complexity. It’s not entirely clear if the performance gain justifies the added complexity, especially for smaller backbones.

* Semi-supervised BN limits: Using unlabeled data only in BN layers may not exploit all information in the unlabeled data. The claim of comparable performance to SOTA might depend heavily on dataset characteristics.

**Questions:**

1. How sensitive is the one-shot rank allocation to the initial gradient estimation? Could this approximation fail in practice?

2. Is there any trade-off in inference speed due to merging low-rank matrices in the testing stage?

3. Could the semi-supervised BN approach handle situations where labeled and unlabeled distributions differ significantly?

---

> ### Author Response · Authors · 2025-11-20
> **Authors' rebuttal to the weakness raised by the reviewer.**
>
> **Weakness 1. Complexity vs. gain: The random-deactivation and one-shot rank allocation add methodological complexity. It’s not entirely clear if the performance gain justifies the added complexity, especially for smaller backbones.**
>
> In our manuscript (Section 4.2 and Appendix D.5), ablation studies using backbones with various sizes (base, medium, large) have been provided to justify the contribution of the proposed three techniques: one-shot rank allocation (RA), random deactivation (RD), and semi-supervised BN (SSBN). The results clearly demonstrate that they both significantly increase the CVDs detection performance of the fine-tuned models, without introducing heavy computational burdens. Consequently, we believe that adding them to the proposed CE-SSL framework is essential to achieve a good balance between fine-tuning performance and computational efficiency. For simplicity, we present the brief results on the PTB-XL and Chapman databases here. Full results can be found in the main paper.
>
> *Table R1. Brief Ablation Study of CE-SSL on the PTB-XL Dataset. 'RA: One-Shot Rank Allocation', 'RD: Random Deactivation', 'SSBN: Semi-Supervised BN'*
> |Methods | Params | Time/iter | Macro $F_{β=2}$ |
> |---|---|---|---|
> | LoRA | 0.795M | 87ms | 0.537±0.008|
> | +RA | 0.582M | 91ms | 0.554±0.008 |
> | +RD | 0.795M | 83ms | 0.558±0.010 |
> | +SSBN | 0.795M | 111ms | 0.558±0.012 |
> | +RA+RD | 0.582M | 87ms | 0.565±0.007 |
> | CE-SSL (RA+RD+SSBN) | 0.582M | 110ms | 0.578±0.006|
>
> **Weakness 2. Semi-supervised BN limitations: Using unlabeled data only in BN layers may not fully utilize all the information in the unlabeled data. The claim of comparable performance to SOTA might depend heavily on dataset characteristics.**
>
> We agree that semi-supervised BN cannot exploit all the information within the unlabeled data, which is designed to achieve high computation efficiency. In our study, its performance and stability are validated by various independent and real-world ECG datasets, in the task of CVDs detection using pre-trained models. Although it cannot fully exploit the information, it can achieve comparable performance to “heavy” SSL methods and demonstrate much higher computational efficiency. Additionally, its effectiveness across diverse datasets indicates that the claim might not depend heavily on dataset characteristics. In the updated manuscript, we rephrase our claim related to the performance of semi-supervised BN as in Section 2.3:
>
>  *However, the results demonstrate that it achieves comparable CVDS detection performance to the SOTA methods on four downstream ECG datasets, while achieving less memory consumption and faster training speed.*
>
> On the other hand, we also agree that whether this claim holds in other areas deserves further investigation, which will be included in our future studies.

---

> ### Author Response · Authors · 2025-11-20
> **Authors' rebuttal to the questions raised by the reviewer.**
>
> **Question 1: How sensitive is the one-shot rank allocation to the initial gradient estimation? Could this approximation fail in practice?**
>
> We sincerely thank for reviewer for this question. In our manuscript, we discuss the sensitivity of the one-shot rank allocation to the initial gradient estimation, which is presented in Appendix D.9: Effect of Warm-Up Epochs for Rank Allocation, due to page limits. The experiment results demonstrate that the one-shot rank allocation is insensitive to the initial gradient estimation, because the pre-trained models carry useful information for the downstream tasks. However, if the pre-trained models are not well-suited for the downstream tasks, the approximation may not achieve good performance. This condition is unusual because researchers typically fine-tune pre-trained models that can generalize well to downstream tasks in practice.
>
> **Question 2: Is there any trade-off in inference speed due to merging low-rank matrices in the testing stage?**
>
> There is no trade-off in inference speed due to merging low-rank matrices in the testing stage. Because after merging low-rank matrices in the testing stage, the inference process of the fine-tuned model remains the same as the pre-trained model.
>
> **Question 3: Could the semi-supervised BN approach handle situations where labeled and unlabeled distributions differ significantly?**
>
> In the semi-supervised BN, unlabeled data is used to provide an accurate estimation of the BN parameters, which prevents the model from over-fitting to the limited labeled samples. In practice, if the unlabeled and labeled data are sampled from multiple datasets, their distributions may differ; however, this does not render the semi-supervised BN ineffective.  In our manuscript, we conduct a cross-dataset external validation to demonstrate it, which is presented in Section 4.3. Specifically, we sample unlabeled and labeled data from four independent downstream datasets for fine-tuning, and use an external dataset for evaluation. This experiment can quantify the cross-distribution robustness of different methods, where the unlabeled training data, labeled training data, and the test data might follow different distributions. The results demonstrate that adding the semi-supervised BN to our system can increase macro $F_{\beta=2}$ score by 3.67%, without introducing heavy computational burdens like SOTA semi-supervised methods.
>
> *Table R2. External Validation of CE-SSL on the held-out dataset. 'RA: One-Shot Rank Allocation', 'RD: Random Deactivation', 'SSBN: Semi-Supervised BN'*
> |Methods |Macro AUC|MAP|Macro $G\_{β=2}$| Macro $F\_{β=2}$ |
> |---|---|---|---|---|
> |FT|0.866|0.588|0.315|0.558|
> |LoRA|0.868|0.571|0.298|0.550|
> |RA + RD|0.879|0.606|0.310|0.563|
> |CE-SSL (RA+RD+SSBN)|0.886|0.620|0.343|0.600|
>
> **At last, thank you again for your valuable feedback and insights.**

---

### Official Review · Reviewer_aD9M · 2025-11-01

**Soundness:** 3
**Presentation:** 3
**Contribution:** 2
**Rating:** 6
**Confidence:** 4

**Summary:**

This paper proposes CE-SSL, a computation-efficient semi-supervised learning framework for ECG-based cardiovascular disease detection. The method combines random-deactivation LoRA (RD-LoRA), one-shot rank allocation, and a lightweight semi-supervised BatchNorm module to achieve efficient fine-tuning of large ECG models with limited labeled data. CE-SSL aims to reduce the computational cost of semi-supervised learning while maintaining strong performance. Experiments on four ECG datasets show consistent gains in Fβ=2 with 30–70% lower training cost compared to standard SSL approaches. The framework is conceptually clear, empirically validated, and practically relevant for resource-constrained medical AI applications.

**Strengths:**

1. The paper tackles a well-defined and realistic challenge—efficiently adapting large ECG models under limited labeled data and computation—which is an important problem for real-world clinical applications.
2. The proposed framework integrates stochastic LoRA activation, one-shot rank allocation, and semi-supervised BN in a logically consistent way. The design is simple, interpretable, and effectively reduces computational overhead without compromising accuracy.
3. Evaluations across four ECG datasets demonstrate solid and reproducible gains in Fβ=2 with substantial reductions in computation and memory cost. The results are consistent and adequately support the efficiency claims.
4. The framework is easy to implement, uses public datasets, and aligns well with deployment needs in constrained biomedical environments, making it practically useful beyond academic settings.

**Weaknesses:**

1. The derivation of RD-LoRA assumes independence between the random gate and the low-rank parameters, which is not strictly valid during training. While this approximation is common in stochastic-depth literature and works empirically, it should be explicitly acknowledged rather than presented as a rigorous proof.
2. The paper only removes the semi-supervised BN in ablations, leaving unclear how much each individual component (RD-LoRA, rank allocation, SSL-BN) contributes to the final performance or whether their synergy is essential.
3. The reported efficiency gain partly arises from a simpler semi-supervised design compared with heavier methods like FixMatch or FlexMatch. A more balanced discussion of this difference would make the efficiency claim more convincing.

**Questions:**

1. Could the authors clarify how the random gating variable δ interacts with the low-rank matrices during training? Since δ affects which adapters receive gradients, A and B are not fully independent of δ. Explaining this as an approximation (similar to stochastic depth) would help align the theory with practical implementation.
2. It would strengthen the paper to report results for variants that retain only one of the three modules (e.g., RD-LoRA only, SSL-BN only). Such analysis could help confirm whether the performance–efficiency trade-off truly relies on their joint design.
3. The improvement from SSL-BN appears modest. Can the authors provide direct comparisons between standard BN (using only labeled data) and SSL-BN (using both labeled and unlabeled data) to quantify the real contribution of unlabeled samples?
4. Have the authors tested whether CE-SSL generalizes across centers or device domains (e.g., training on PTB-XL and testing on Chapman)? Demonstrating cross-distribution robustness would further support the framework’s practical value.

---

> ### Author Response · Authors · 2025-11-20
> **Authors' rebuttal to the weakness raised by the reviewer.**
>
> **Weakness 1. Independence assumption during training.**
>
> We sincerely thank the reviewer for the comment. We agree that the random gate and the low-rank parameters are not strictly independent in during fine-tuning. Hence, in Section 2.1, we add some descriptions to clarify that Eq. 5 can only approximate the expectation of the hidden layer outputs $h$ during fine-tuning.
>
> $$
> \mathbb{E}\_{\delta \sim Ber(\delta, 1-p)}\left[h\right]\approx(W\_0+\mathbb{E}\_{\delta \sim Ber(\delta, 1-p)}\left[\delta\right]BA)X=(W\_0+(1-p)BA)X. (Eq. 5)
> $$
>
> However, we would like to clarify that our explanation regarding the effectiveness of the proposed RD-LoRA is still insightful (Appendix C.1). Considering that non-linear deep networks are difficult to analyze, we simply considers a linear network $M$ with $n$ fully-connected layers, defined as $M(X)=\prod\_{i=1}^{n} W^{i}\_0 X$, where $X$ is the input data and $W^{i}\_0\in\mathbb{R}^{c\_{out}\times c\_{in}}$ is the pre-trained weight matrix at the $i$-th layer.
>
> After being fine-tuned and before being merged into the pre-trained weights, the low-rank matrices $\{A, B\}$ are fixed, and only $\delta$ is a random variable in the testing stage. Hence, given the testing data $X_{test}$ and the ground truth $Y_{test}$, the testing loss can be computed as,
>
> $$
> \mathcal{L}(Y\_{test},\mathbb{E}\_{\delta \sim Ber(\delta, 1-p)}[M (X\_{test})])=\mathcal{L}(Y\_{test},\prod\_{i=1}^{n}(W^{i\}_0+(1-p)B^iA^i)X\_{test}) \leq \mathbb{E}\_{\delta \sim Ber(\delta, 1-p)}\left[\mathcal{L}(Y\_{test},M(X\_{test}))\right].  (Eq. 14)
> $$
>
> The inequality holds due to the convexity of the multi-label cross-entropy loss function and the Jensen inequality. It indicates that merging the optimized low-rank parameters with the pre-trained weights generates an ensemble network. Its error on the test data is less than or equal to the expected error of the sub-networks.  This analysis provides a brief explanation of the effectiveness of RD-LoRA in increasing the performance of the fine-tuned model.
>
> **Weakness 2. How much each individual component contributes to the final performance**
>
> We thank the reviewer for this comment. We would like to clarify that we not only removed semi-supervised BN but also RD-LoRA and one-shot rank allocation from CE-SSL, as shown in our previous ablation study. In the updated manuscript, we present a more comprehensive ablation study to thoroughly evaluate the contribution of each module and demonstrate that their synergy is crucial. Specifically, we add the proposed one-shot rank allocation (RA), random deactivation (RD), and semi-supervised BN (SSBN) to the supervised baseline LoRA and record the corresponding model performance on the four datasets. Here, we present the results on the PTB-XL database briefly. Full results can be found in Section 4.2 Ablation Study.
>
> *Table R1. Brief Ablation Study of CE-SSL on the PTB-XL Dataset. 'RA: One-Shot Rank Allocation', 'RD: Random Deactivation', 'SSBN: Semi-Supervised BN'*
> |Methods | Params | Time/iter | Macro $F_{β=2}$ |
> |---|---|---|---|
> | LoRA | 0.795M | 87ms | 0.537±0.008|
> | +RA | 0.582M | 91ms | 0.554±0.008 |
> | +RD | 0.795M | 83ms | 0.558±0.010 |
> | +SSBN | 0.795M | 111ms | 0.558±0.012 |
> | +RA+RD | 0.582M | 87ms | 0.565±0.007 |
> | CE-SSL (RA+RD+SSBN) | 0.582M | 110ms | 0.578±0.006|
>
>
> The results demonstrate that the proposed three modules all contribute to increasing the CVDs detection performance of the fine-tuned models, and their synergy is essential
>
> **Weakness 3. The reported efficiency gain partly arises from a simpler semi-supervised design compared with heavier methods like FixMatch or FlexMatch. A more balanced discussion of this difference would make the efficiency claim more convincing.**
>
> We thank the reviewer for this comment. We agree that the efficiency gain achieved by CE-SSL partly arises from our simpler semi-supervised design (semi-supervised BN). However, we want to clarify that the RD-LoRA and the one-shot rank allocation also contribute to the efficiency gain. As shown in **Table R1**, they enhance the CVDs detection performance of the fine-tuned model, contributing to a good balance between computational efficiency and diagnostic performance.
>
> Specifically, adding the RD-LoRA module to the supervised baseline significantly increases CVDs detection performance and slightly improves training speed (+RD vs LoRA), when the deactivation probability $p$ is set to 0.2.  As shown in Appendix D.7, the Time/iter can be significantly reduced by increasing $p$ from 0.2 to 0.5. As illustrated in Appendix D.5, the improvements of in training speed become more obvious on larger backbones. Secondly, the introduction of RA cannot increase the training speed but further reduce the number of trainable parameters (+RA+RD vs +RD).  In summary, our discussion demonstrates that all the modules and designs within CE-SSL are critical to achieving high computational efficiency and robust CVDs detection performance simultaneously.

---

> ### Author Response · Authors · 2025-11-20
> **Authors' rebuttal to the questions 1-2 raised by the reviewer.**
>
> **Question 1. Could the authors clarify how the random gating variable δ interacts with the low-rank matrices during training?**
>
> We thank the reviewer for this valuable question. We would like to clarify both the implementation and the theoretical aspects.
>
> *Implementation aspects:*
>
> As illustrated in the pseudo-code provided in Appendix Algorithm 1, at the i-th layer within the backbone, the random gating variable $\delta\_i$ controls whether low-rank parameters $A_i$ and $B_i$ participate in the forward process and receive gradients after the backward process. We agree that they are not fully independent of $\delta\_i$ in the training stage, according to the practical implementation.  Hence, we add necessary descriptions to our updated manuscript to remind readers that Eq. 5 can only approximate the expectation of the hidden layer outputs $h$ during the training stage.
>
> $$
> \mathbb{E}\_{\delta \sim Ber(\delta, 1-p)}\left[h\right]\approx(W\_0+\mathbb{E}\_{\delta \sim Ber(\delta, 1-p)}\left[\delta\right]BA)X=(W\_0+(1-p)BA)X. (Eq. 5)
> $$
>
> For the linear networks considered in our brief analysis, we would like to clarify that the expectation of $h$ can be computed during the testing stage, as $A_i$ and $B_i$ are fixed and not variable. This property helps us to explain the ensemble property of RD-LoRA during the testing stage.
>
> *Theoretical aspects:*
>
> In the training stage, for known inputs $X$ and ground truths $Y$, the expectation of the training loss function $\mathbb{E}\_{\delta\sim Ber(\delta, 1-p)}\left[\mathcal{L}(Y,M(X))\right]$ at the iteration $t$ can be given as,
>
> $$
> \mathbb{E}^{t}\_{\delta\sim B(\delta, 1-p)}\left[\mathcal{L}(Y,M(X))\right]=
> (1-p)^n\mathcal{L}(Y,\prod\_{i=1}^{n}(W_0^{i}+B^i\_tA^i\_t)X)+\sum_{j=1}^{n}\left[p(1-p)^{n-1}\mathcal{L}(Y,\prod\_{i=1,i\neq j}^{n}(W\_0^{i}+B^i\_tA^i\_t)W\_0^{j})X\right]+\cdots + p^n\mathcal{L}(Y,\prod\_{i=1}^{n} W^{i}\_0X). (Eq.13)
> $$
>
> This expectation is derived by simply listing all the possible sub-networks produced by the random gate $\delta\sim B(\delta, 1-p)$ and calculating the corresponding probability using $p$, which does not assume independence between the gate and the low-rank parameters. Eq.13 can be regarded as a weighted mean of the losses of $2^n$ sub-networks, which are minimized during iterative model training. In other words, with RD-LoRA, we can implicitly train various sub-networks with different activated low-rank parameters. It can increase the fine-tuning computation efficiency because deactivating some low-rank matrices eliminates the need to compute update matrices in certain layers.
>
> **Question 2. It would strengthen the paper to report results for variants that retain only one of the three modules (e.g., RD-LoRA only, SSL-BN only). Such analysis could help confirm whether the performance–efficiency trade-off truly relies on their joint design.**
>
> We thank the reviewer for this comment. In the updated manuscript (Section 4.2), we present a more comprehensive ablation study to thoroughly evaluate the contribution of each module and demonstrate that their synergy is crucial. Details see our responses to Weakness 2.

---

> ### Author Response · Authors · 2025-11-20
> **Authors' rebuttal to the questions 3-4 raised by the reviewer.**
>
> **Question 3. The improvement from SSL-BN appears modest. Can the authors provide direct comparisons between standard BN (using only labeled data) and SSL-BN (using both labeled and unlabeled data) to quantify the real contribution of unlabeled samples?**
>
> We thank the reviewer for this comment. In our updated manuscript, we provide a direct comparison between standard BN and SSL-BN in the ablation study and the external validation.
>
> *Ablation study (Paper Section 4.2)*
>
> As shown in Table R1, we can observe that the SSL-BN-only models demonstrate better CVDs detection performance than the BN-only models.  A main advantage of semi-supervised learning is enhancing the model’s generalization performance on unseen samples, particularly when labeled data is expensive to collect. The improvement from SSL-BN appears modest in the ablation because the training data and the testing data are sampled from the same dataset, which cannot fully assess the generalization performance of the fine-tuned models. Consequently, an external validation is conducted to fully evaluate the real contribution of unlabeled samples, which is introduced below.
>
> *External Validation (Paper Section 4.3)*
>
> In our previous manuscript, the external validation was introduced in the Appendix due to page limits. In the updated manuscript, we move it to the main manuscript and add more experiments for model comparison. Specifically, four downstream datasets (G12EC, Chapman, PTB-XL, Ningbo) are used for fine-tuning, and one held-out dataset provided by [1] is used for evaluation. We integrate the proposed one-shot rank allocation (RA) and random-deactivation low-rank adaptation modules (RD) as a powerful supervised baseline (RA+RD). We then compare its generalization performance with that of CE-SSL using the external dataset, as shown in **Table R2** in the updated paper. **Note that the difference between RA+RD and CE-SSL is whether to use semi-supervised BN or not.** In other words, the performance gap between them quantifies the contribution of unlabeled samples to the generalization performance. The results demonstrate that CE-SSL outperforms the powerful supervised baseline (RA+RD) by **3.67% on macro $F\_{\\beta=2}$ score**. It indicates that leveraging the information within the unlabeled data using semi-supervised BN can improve the model’s generalization performance in CVDs detection.
>
> *Table R2. External Validation of CE-SSL on the held-out dataset. 'RA: One-Shot Rank Allocation', 'RD: Random Deactivation', 'SSBN: Semi-Supervised BN'*
> |Methods |Macro AUC|MAP|Macro $G\_{β=2}$| Macro $F\_{β=2}$ |
> |---|---|---|---|---|
> |FT|0.866|0.588|0.315|0.558|
> |LoRA|0.868|0.571|0.298|0.550|
> |RA + RD|0.879|0.606|0.310|0.563|
> |CE-SSL (RA+RD+SSBN)|0.886|0.620|0.343|0.600|
>
> [1] Lai, J., Tan, H., Wang, J., Ji, L., Guo, J., Han, B., ... & Yang, W. (2023). Practical intelligent diagnostic algorithm for wearable 12-lead ECG via self-supervised learning on large-scale dataset. Nature Communications, 14(1), 3741.
>
> **Question 4. Have the authors tested whether CE-SSL generalizes across centers or device domains (e.g., training on PTB-XL and testing on Chapman)? Demonstrating cross-distribution robustness would further support the framework’s practical value.**
>
> We relocate the external validation from the Appendix to the main manuscript, demonstrating the cross-distribution robustness of CE-SSL. A brief summary of the results and analysis can be found in our responses to Question 3 and Section 4.3 in our paper.
>
> **At last, thank you again for your valuable feedback and insights.**

---

> ### Comment · Reviewer_aD9M · 2025-11-28
> **Revised Review After Authors’ Response**
>
> I thank the authors for the detailed and clear rebuttal. The additional clarifications and extended experiments have addressed my main concerns.
>
> The explanation of the independence assumption in RD-LoRA is now consistent with practical stochastic-depth approximations, and the ensemble interpretation helps make the mechanism intuitive. The expanded ablation study (RA only, RD only, SSL-BN only, and their combinations) provides much stronger evidence that all three components contribute meaningfully and that their combination is indeed synergistic. The external validation experiments added to the main paper effectively demonstrate the value of SSL-BN and support the cross-distribution robustness of CE-SSL.
>
> Overall, the revision improves both clarity and empirical support. The proposed framework remains simple, practical, and relevant for efficient ECG model adaptation in constrained settings. I consider the paper marginally above the acceptance threshold, and I update my assessment accordingly.

---

### Official Review · Reviewer_ZnbY · 2025-11-01

**Soundness:** 3
**Presentation:** 3
**Contribution:** 3
**Rating:** 4
**Confidence:** 4

**Summary:**

This paper introduces CE-SSL, a computation efficient semi-supervised method for fine-tuning pre-trained models to address label scarcity and computational limitation in downstream ECG datasets for cardiovascular disease (CVD) detection. CE-SSL combines three main methods: (I) Random deactivation of layers' low-rank matrices which is very similar to classic dropout, but applied to full layers compared to neurons in layer (see Eq.2); (II) one-shot rank allocation, considering only two possible ranks (see Eqn.10) and (III) A lightweight semi-supervised learning using both labeled and unlabeled data to update BN layers in a semi-supervised manner. These improve generalization and stability under label scarcity while reducing memory and computation costs. Experiments on 4 public ECG datasets show that CE-SSL outperforms some SOTA semi-supervised and parameter-efficient baselines in terms of multi-label classification accuracy, while reducing training time, GPU memory footprint, and parameter storage needs.

Overall, the work is quite solid in providing extensive empirical evaluation with some ablation (e.g., on $p$, dropout rate or rank $r$), yet theoretical and methodological contributions are limited. All components of the method have been done before, including random deactivation (not much different than classic dropout); one-shot rank allocation, and semi-supervised tuning; some specifics could be different but ideas are there. Also, the straightforward argument in App C.1 and its conclusion (final network is an ensemble of all possible sub-networks during model training) are not too surprising given the sampling technique. I believe this work is useful for the "Health informatics" community, but I am afraid it could be of limited interest to ICLR community.

**Strengths:**

- Extensive validation with large backbone models against 6 baselines, and on 4 downstream datasets, while also include some ablation.
- Flexibility with initial rank $r$ and top-k important layers, allowing for accuracy-computation tradeoff.
- Avoids pseudo-labeling & consistency training to reduce complexity and potential error-propagation from unlabeled data.

**Weaknesses:**

- I believe the main weakness of this work is its limited scope and interest to ICLR community. I enumerate some more suggestions below:
- I suggest including direct comparisons with fully supervised models trained on the same downstream datasets with limited labels. This will highlight how much performance gain is attributed to the semi-supervised pre-trained method.
- Better justification/analysis for the use of a single gradient pass for rank estimation could benefit the paper (Is Taylor approximation sufficiently accurate? Could you ablate the approximation to deduce its effect?
- Other ablations could include comparisons between "no rank allocation", "random allocation", and "one-shot allocation" to isolate the effect of the latter.
- Benchmarking against newer ECG foundation models (e.g., large-scale pretrained ECG transformers or contrastive frameworks) would position CE-SSL more clearly within the current state of the field.
-  To quantify the quality of pre-trained representations by CE-SSL, one can use linear probing. Also, it It could be helpful to evaluate CE-SSL using various backbone models (e.g., transformer-based or hybrid ECG encoders).

**Questions:**

See weakness section.

---

> ### Author Response · Authors · 2025-11-20
> **Authors' rebuttal to the weakness 1 raised by the reviewer.**
>
> **Weakness 1. I believe the main weakness of this work is its limited scope and interest to ICLR community.**
>
> We thank the reviewer for the concern.  However, we would like to clarify that the novelty and contribution of CE-SSL lie in achieving robust fine-tuning performance without a significant sacrifice of computational efficiency. It is a crucial issue in the domain of computation-efficient deep learning and aligns with the scopes of the ICLR community. To the best of our knowledge, no prior study has designed and evaluated a framework to achieve this before. The effectiveness of CE-SSL is empirically validated on ECG datasets for CVDs detection, which is an important application for medical AI and is also explored by other ICLR papers [1,2]. More importantly, its methodology and overall design are general and not tailored for ECG applications. Consequently, we believe it has the potential to be generalized and applied to other fields in resource-constrained medical AI, as well as to broader domains in efficient deep learning.
>
> At the same time, we agree that the components of CE-SSL are motivated by some classic theories (Taylor Expansion, etc). But we want to kindly clarify that they are different from the existing techniques and have not been done before. First, random-deactivation low-rank adaptation (RD-LoRA) is applied to the low-rank adapters. RD-LoRA deactivates them during fine-tuning in a stochastic manner, enabling an ensemble of diverse adapter combinations during fine-tuning. However, classic dropout is applied to the neurons while all adapters are activated. Consequently, it cannot accelerate the gradient forward process and cannot explore diverse adapter combinations like RD-LoRA. Additionally, one-shot rank allocation determines the optimal rank distribution using only one gradient pass, under the guidance of a simple yet effective estimation provided in our study (Eq. 10). On the contrary, previous works like AdaLoRA [3] and IncreLoRA [4] applied a computation-inefficient and multi-step estimation process, which relies on singular value decomposition and thus introduces extra hyper-parameters for orthogonality regularization. At last, our lightweight semi-supervised learning module integrates normalization and parameter optimization into one forward-backward step to avoid extra computational costs, which is more efficient than the two-step process proposed in [5]. In our paper, extensive experiments on various pre-trained models and downstream datasets demonstrate that CE-SSL outperforms existing methods in CVDs detection with much lower computational costs. Moreover, a comprehensive ablation study indicates that the proposed three techniques are crucial to achieve a good balance between computational efficiency and detection performance under limited supervision.
>
> [1] Na, Y., Park, M., Tae, Y., & Joo, S. (2024). Guiding masked representation learning to capture spatio-temporal relationship of electrocardiogram. The Thirteenth International Conference on Learning Representations.
>
>
> [2] Jin, J., Wang, H., Li, H., Li, J., Pan, J., & Hong, S. (2025). Reading your heart: Learning ecg words and sentences via pre-training ecg language model. The Twelfth International Conference on Learning Representations.
>
> [3] Zhang, Q., Chen, M., Bukharin, A., Karampatziakis, N., He, P., Cheng, Y., ... & Zhao, T. (2023). Adalora: Adaptive budget allocation for parameter-efficient fine-tuning. arXiv preprint arXiv:2303.10512.
>
> [4] Zhang, F., Li, L., Chen, J., Jiang, Z., Wang, B., & Qian, Y. (2023). Increlora: Incremental parameter allocation method for parameter-efficient fine-tuning. arXiv preprint arXiv:2308.12043.
>
> [5] Koçyigit, M. T., Sevilla-Lara, L., Hospedales, T. M., & Bilen, H. (2020). Unsupervised batch normalization. In Proceedings of the IEEE/CVF Conference on Computer Vision and Pattern Recognition Workshops (pp. 918-919).

---

> ### Author Response · Authors · 2025-11-20
> **Authors' rebuttal to the weakness 2-4 raised by the reviewer.**
>
> **Weakness 2. I suggest including direct comparisons with fully supervised models trained on the same downstream datasets with limited labels. This will highlight how much performance gain is attributed to the semi-supervised pre-trained method.**
>
> We thank the reviewer for this valuable suggestion. In our previous paper, a direct comparison between supervised baseline and semi-supervised methods was presented in the Appendix due to page limits. In the updated manuscript, we have relocated the relevant discussions to Section 4.3 (Ablation Study) and Section 4.4 (External Validation). Specifically, as shown in **Table R1**, adding semi-supervised BN to LoRA (a fully supervised fine-tuning method) increases the $F\_{\beta=2}$ score from 0.537±0.008 to 0.558±0.012 on the PTB-XL database.
>
> *Table R1. Brief Ablation Study of CE-SSL on the PTB-XL Dataset. 'RA: One-Shot Rank Allocation', 'RD: Random Deactivation', 'SSBN: Semi-Supervised BN'*
> |Methods | Params | Time/iter | Macro $F_{β=2}$ |
> |---|---|---|---|
> | LoRA | 0.795M | 87ms | 0.537±0.008|
> | +RA | 0.582M | 91ms | 0.554±0.008 |
> | +RD | 0.795M | 83ms | 0.558±0.010 |
> | +SSBN | 0.795M | 111ms | 0.558±0.012 |
> | +RA+RD | 0.582M | 87ms | 0.565±0.007 |
> | CE-SSL (RA+RD+SSBN) | 0.582M | 110ms | 0.578±0.006|
>
> *Table R2. External Validation of CE-SSL on the held-out dataset. 'RA: One-Shot Rank Allocation', 'RD: Random Deactivation', 'SSBN: Semi-Supervised BN'*
> |Methods |Macro AUC|MAP|Macro $G\_{β=2}$| Macro $F\_{β=2}$ |
> |---|---|---|---|---|
> |FT|0.866|0.588|0.315|0.558|
> |LoRA|0.868|0.571|0.298|0.550|
> |RA + RD|0.879|0.606|0.310|0.563|
> |CE-SSL (RA+RD+SSBN)|0.886|0.620|0.343|0.600|
>
> Additionally, as shown in **Table R2** (External Validation), we integrate the proposed one-shot rank allocation (RA) and random-deactivation low-rank adaptation modules (RD) as a powerful supervised baseline (RA+RD) for comprehensive comparisons. Two classic fine-tuning methods are also used for benchmarking, including Full Fine-Tuning (FT) and LoRA. The results demonstrate that CE-SSL outperforms the powerful supervised baseline (RA+RD) by 3.67% on macro $F_{\beta=2}$ score. Additionally, CE-SSL achieves a macro AUC of 0.886 on the external dataset and demonstrates better cross-distribution robustness than LoRA (0.866) and FT (0.868). In summary, we believe the ablation study and the external validation can quantify and highlight the performance gain of introducing semi-supervised methods.
>
> **Weakness 3. Better justification/analysis for the use of a single gradient pass for rank estimation could benefit the paper (Is Taylor approximation sufficiently accurate? Could you ablate the approximation to deduce its effect?**
>
> We thank the reviewer for this question. K-order Taylor approximation is sufficiently accurate to estimate the importance of different low-rank parameters if the loss function is K-times differentiable at the point. In practice, the first-order approximation is a common strategy for importance estimation due to its low computational complexity and empirical effectiveness [3]. In our manuscript, we also present the ablation of the approximation in **Table R1**. The results demonstrate that applying the Taylor approximation for rank allocation can increase macro $F_{\beta=2}$ from 0.537±0.008 (LoRA) to 0.554±0.008 (+RA) in the PTB-XL dataset.
>
> Additionally, in Appendix D.9, we also demonstrate that determining the optimal ranks using a single gradient pass before fine-tuning does not hinder the model's performance.
>
> [3] Molchanov, P., Mallya, A., Tyree, S., Frosio, I., & Kautz, J. (2019). Importance estimation for neural network pruning. In Proceedings of the IEEE/CVF conference on computer vision and pattern recognition (pp. 11264-11272).
>
> **Weakness 4. Other ablations could include comparisons between "no rank allocation", "random allocation", and "one-shot allocation" to isolate the effect of the latter.**
>
> We thank the reviewer for this suggestion. In our updated ablation (**Table R1**), “no rank allocation” (also known as LoRA) and “one-shot rank allocation” (RA) are implemented to evaluate the contribution of the latter. “Random allocation” is not included for discussion because it is hard to define how many random rank distributions should be used for comparisons.

---

> ### Author Response · Authors · 2025-12-02
> **Authors' rebuttal to the weakness 5-6 raised by the reviewer.**
>
> **Weakness 5. Benchmarking against newer ECG foundation models (e.g., large-scale pretrained ECG transformers or contrastive frameworks) would position CE-SSL more clearly within the current state of the field.**
>
> We thank the reviewer for this valuable suggestion. In Section 4.3 (External Validation), we fine-tune different pretrained backbones using full fine-tuning (FT) and CE-SSL on four downstream datasets and evaluate them on the external dataset. We include five backbones for benchmarking, including the backbones (base, medium, large) provided in our study and two newer ECG foundation models: ECG-FM [4] and ECGFounder [5]. As shown in **Table R3**, the results demonstrate that ECGFounder and our medium backbone demonstrate the best and the second-best generalization performance on the external dataset. More importantly, we can observe that CE-SSL can consistently improve their performance, which indicates its effectiveness across various backbones.
>
> *Table R3. External Validation of CE-SSL and FT using different pre-trained backbones.*
> |Methods |Macro AUC|MAP|Macro $G\_{β=2}$| Macro $F\_{β=2}$ |
> |---|---|---|---|---|
> |ECGFM + FT                 |0.814|0.445|0.184|0.379|
> |Base + FT                      |0.852|0.516|0.265|0.503|
> |Medium + FT                  |0.866|0.588|0.315|0.558|
> |Large + FT                     |0.857|0.539|0.280|0.518|
> |ECGFounder + FT         |0.877|0.628|0.302|0.575|
> |Medium + CE-SSL         |0.886|0.620|0.343|0.600|
> |ECGFounder + CE-SSL|0.893|0.671|0.314|0.598|
>
>
> **Weakness 6. To quantify the quality of pre-trained representations by CE-SSL, one can use linear probing. Also, it It could be helpful to evaluate CE-SSL using various backbone models (e.g., transformer-based or hybrid ECG encoders).**
>
> We thank the reviewer for this valuable suggestion, but we would like to clarify that CE-SSL is not a pre-training method, but rather a semi-supervised fine-tuning method, which is clearly introduced in our paper. We agree that we should evaluate CE-SSL using various backbones. Specifically, we include ECG-FM [4] (a transformer-based model) and ECGFounder [5] (a CNN-based model) for the experiments in Section 4.3 (External Validation). As shown in **Table R3**, the results demonstrate the effectiveness of CE-SSL across various backbones.
>
> [4] McKeen, K., Masood, S., Toma, A., Rubin, B., & Wang, B. (2025). Ecg-fm: An open electrocardiogram foundation model. JAMIA open, 8(5), ooaf122.
>
> [5] Li, J., Aguirre, A. D., Junior, V. M., Jin, J., Liu, C., Zhong, L., ... & Hong, S. (2025). An Electrocardiogram Foundation Model Built on over 10 Million Recordings. NEJM AI, 2(7), AIoa2401033.
>
> **At last, thank you again for your valuable feedback and insights.**

---

### Author Response · Authors · 2025-11-25
**Summary of Rebuttal and Revisions**

First of all, we sincerely thank the reviewers for their time in reviewing our paper and providing insightful comments.

We thanks the reviewers for recognizing the strengths of CE-SSL. We list the main strengths here:

***(1) CE-SSL tackles a realistic and important challenge in medical AI (Reviewers aD9M, xbwK)*** -- efficiently fine-tuning large pre-trained models under limited supervision and computation costs. In other words, CE-SSL's novelty and contribution lie in achieving robust fine-tuning performance under limited supervision, without a significant sacrifice of computational efficiency.

***(2) CE-SSL is well-designed (Reviewers aD9M, xbwK).*** CE-SSL is easy to deploy but effective, interpretable and well-motivated. Moreover, it is conceptually clear, flexible, and practically useful beyond academic settings.

***(3) Experiments are solid and extensive (Reviewers ZnbY, aD9M, xbwK).*** The experiments conducted on public datasets are consistent, convincing and adequately support our claims. Without compromising fine-tuning performance, CE-SSL achieves substantial reductions in computation costs compared with previous methods. Consequently, it demonstrates effectiveness and significant superiority in addressing the challenge introduced above.

We also thank the reviewer for pointing out their concerns and we have submitted a point-by-point rebuttal to address them. At the same time, we have revised the paper to improve its quality and made our claim more convincing, according to their valuable suggestions. A revised version of the paper has been submitted and the changes are highlighted in ***blue***. Key updates are summarized as follows:

***(i) Theoretical analysis of the proposed RD-LoRA is refined.*** We have added relevant literature to justify the approximation used in the training process and improved the clarity of our analysis regarding the ensemble property of RD-LoRA in the testing stage.

***(ii) A more detailed ablation study is provided.*** We report the performance of variants that retain only one of the three techniques proposed in our paper. Additionally, a fully supervised baseline is used for comprehensively analyzing the contribution of each technique. The ablation study clearly indicates that the synergy of the techniques is essential to achieve a good balance between **computational efficiency** and **detection performance** under **limited supervision**, which is the challenging goal defined in our paper.

***(iii) A more detailed external validation is provided.*** In the updated manuscript, we move the external validation of CE-SSL from Appendix to the main text and provide more results to demonstrate its cross-distribution robustness. The experiments also indicate that semi-supervised BN can significantly increase the model's generalization performance. Additionally, we demonstrate that the effectiveness of semi-supervised BN remains when the inputs are sampled from diverse distributions.

***(iv) Evaluating CE-SSL using newer ECG-based foundation models.*** In the external validation, we evaluate CE-SSL using ECGFM [1] and ECGFounder [2], as well as the backbones (base, medium, large) provided in our study. The results demonstrate that our pre-trained backbones can achieve comparable performance to ECGFounder, which is a SOTA foundation model for ECG analysis. More importantly, the effectiveness of CE-SSL and its superior performance over full fine-tuning are well demonstrated.

[1] McKeen, K., Masood, S., Toma, A., Rubin, B., & Wang, B. (2025). Ecg-fm: An open electrocardiogram foundation model. JAMIA open, 8(5), ooaf122.

[2] Li, J., Aguirre, A. D., Junior, V. M., Jin, J., Liu, C., Zhong, L., ... & Hong, S. (2025). An Electrocardiogram Foundation Model Built on over 10 Million Recordings. NEJM AI, 2(7), AIoa2401033.

---

### Meta-Review · Area_Chair_sh2P · 2026-01-06

**Summary:**

This submission proposes CE-SSL, a computation-efficient semi-supervised fine-tuning paradigm for ECG-based CVD detection that combines: (i) random-deactivation LoRA (RD-LoRA), (ii) one-shot rank allocation (RA), and (iii) a lightweight semi-supervised BN (SSBN). Across reviews, the paper was viewed as practically motivated with strong empirical validation on multiple public ECG datasets and clear efficiency-oriented metrics (training time / memory / parameter storage), but faced concerns along three axes:

- Novelty / ICLR-level interest. Reviewer ZnbY’s core concern was that each component resembles prior ideas, making the methodological contribution feel incremental even if the engineering integration is useful.

- Methodological clarity and theoretical soundness for RD-LoRA. Reviewer aD9M pointed out that the original derivation implicitly relied on an independence assumption between the random gate and low-rank parameters during training, and requested clearer alignment between theory and implementation.

- Attribution and robustness evidence. Multiple reviewers requested stronger ablations to isolate each component’s effect and stronger evidence of cross-distribution generalization.

After rebuttal/revision, the paper now provides stronger ablation and external-validation evidence, and clarifies the approximation nature of the RD-LoRA analysis. The remaining debate is primarily about novelty/ICLR breadth rather than correctness or empirical support.

While the rebuttal improves clarity and adds experiments, it does not fundamentally change the novelty assessment: the paper remains an incremental combination tailored to an ECG setting, and the general ML insight appears limited relative to ICLR’s bar.
Given these concerns, I am sorry to recommend Reject.

**Reviewer Concerns:**

## Addressed in rebuttal / revision

- More complete ablation to isolate module contributions. The authors added a more complete ablation decomposition with compute metrics and performance.

- Clarification of RD-LoRA training-vs-test assumptions. The authors now acknowledge that the independence-style reasoning is an approximation during training and position the analysis closer to stochastic-depth intuition. This resolves aD9M’s concern that the theory was previously framed too rigorously.

- External validation and backbone benchmarking. The authors moved external validation to the main text and added results on held-out data, plus evaluation on newer ECG foundation models. This improves empirical positioning and partially answers ZnbY’s request for stronger field context.

## Concerns still outstanding (and decisive for the recommendation)

- ICLR-level novelty / general-interest contribution remains weak. Even after revision, the core method remains a composition of known ideas. The rebuttal strengthens the empirical case but does not establish a new principle, a clearly novel algorithmic component, or a compelling generalization beyond the ECG domain. This was ZnbY’s main objection and is not fully overcome by additional experiments.

- Efficiency claims remain entangled with “lighter SSL” rather than a fundamentally better compute–accuracy frontier. Reviewers noted that some efficiency gains come from choosing a lightweight BN-only semi-supervised strategy. The rebuttal adds discussion and ablation evidence, but the paper still does not cleanly establish that the framework advances the state of computation-efficient SSL broadly, rather than presenting a practical, domain-specific trade-off.

Overall, while the rebuttal improves clarity and empirical support, the remaining issues concern scope and contribution, which are difficult to fix via rebuttal alone and are central to the acceptance decision at ICLR.

**Reviewer Scores:**

`Reviewer ZnbY`(original rating: 4)
- Expected change after full discussion: 4 → 4 (no change)
- The authors addressed many actionable requests, which may improve perceived rigor. However, ZnbY’s primary concern is that the work is of limited interest to the ICLR community due to incremental methodological contribution. With full discussion, I expect ZnbY to maintain a below-threshold score.

`Reviewer aD9M`(original rating: 6 )
- Expected change after full discussion: 6 → 6 (no change)
- aD9M’s technical concerns were addressed. In discussion, they would likely stand by that position.

`Reviewer xbwK`(original rating: 6)
- Expected change after full discussion: 6 → 4
- While the rebuttal answers xbwK’s practical questions and strengthens evidence, a fuller discussion would likely focus on ICLR’s novelty bar and general ML contribution. Given xbwK already expressed uncertainty and “would not mind if rejected,” I expect discussion to pull the score slightly downward toward borderline reject(4) once novelty/community-fit is emphasized.

---

### Decision · Program_Chairs · 2026-01-26

Reject